# Enzymatic synthesis and nanopore sequencing of 12-letter supernumerary DNA

Hinako Kawabe [1], Christopher A. Thomas [2], Shuichi Hoshika [3,4], Myong-Jung Kim[3,4], Myong-Sang Kim[4], Logan Miessner[5], Nicholas Kaplan[1], Jonathan M. Craig [2], Jens H. Gundlach [2], Andrew H. Laszlo [2], Steven A. Benner[3,4] & Jorge A. Marchand [1,6] ✉

The 4-letter DNA alphabet (A, T, G, C) as found in Nature is an elegant, yet non-exhaustive solution to the problem of storage, transfer, and evolution of biological information. Here, we report on strategies for both writing and reading DNA with expanded alphabets composed of up to 12 letters (A, T, G, C, B, S, P, Z, X, K, J, V). For writing, we devise an enzymatic strategy for inserting a singular, orthogonal xenonucleic acid (XNA) base pair into standard DNA sequences using 2′-deoxy-xenonucleoside triphosphates as substrates. Integrating this strategy with combinatorial oligos generated on a chip, we construct libraries containing single XNA bases for parameterizing kmer basecalling models for commercially available nanopore sequencing. These elementary steps are combined to synthesize and sequence DNA containing 12 letters – the upper limit of what is accessible within the electroneutral, canonical base pairing framework. By introducing low-barrier synthesis and sequencing strategies, this work overcomes previous obstacles paving the way for making expanded alphabets widely accessible.

The 4-letter standard genetic alphabet of DNA (A, T, G, C) is ubiquitous and one of the defining biomolecular signatures of life on Earth. Nature's ability to read, write, and translate this information forms the basis for life as an emergent property of nucleic acid heteropolymers[1]. Like Nature, humanity has also learned how to manipulate the 4 letters of DNA, spurring major advances in biotechnology, information storage, and healthcare. As archetypal examples, the standard nucleic acids are key components for diagnostic tests to screen for disease[2,3] or detect toxins[4], therapeutics that create immune responses[5], and even as a molecular system for long-term storage of digital information[6,7].

However, theory makes clear that Nature's choice of four building blocks for DNA is far from exhausting the two rules of complementarity that govern canonical, hydrogen-bonding base pairing— (a) size complementarity, where the larger purines (A, G) pair with the smaller pyrimidines (C, T) and (b) hydrogen bonding complementarity, where hydrogen bond donors pair with hydrogen bond acceptors. These rules allow for up to 12 different nucleotides, forming six orthogonal pairs (Fig. 1, Supplementary Table 1). Within these rules, multiple heterocyclic systems are also available to support each of the various hydrogen bonding combinations (e.g., $S^n$ and $S^c$). It is therefore possible to envision various 'supernumerary' (def. in excess of the normal) DNA codes as a fusion of the natural nucleobases (A, T, G, C) with a set of the synthetic hydrogen bonding xenonucleobases (B, S, P, Z, X, K, J, V). As a hallmark example, a subset of authors from this work previously reported on chemical synthesis and characterization of one such 8-letter code, hachimoji DNA, comprised of natural (A, T, G, C) and synthetic (B, $S^c$, P, Z) nucleobases[8].

To a minor extent, the biomolecular compatibility of expanded non-canonical hydrogen bonding base pairings has previously been

[1]Department of Chemical Engineering, University of Washington, Seattle, WA 98195, USA. [2]Department of Physics, University of Washington, Seattle, WA 98195, USA. [3]Foundation for Applied Molecular Evolution, Alachua, FL 32615, USA. [4]Firebird Biomolecular Sciences LLC, Alachua, FL 32615, USA. [5]Department of Biochemistry, University of Washington, Seattle, WA 98195, USA. [6]Molecular Engineering & Sciences Institute, University of Washington, Seattle, WA 98195, USA. ✉e-mail: jmarcha@uw.edu

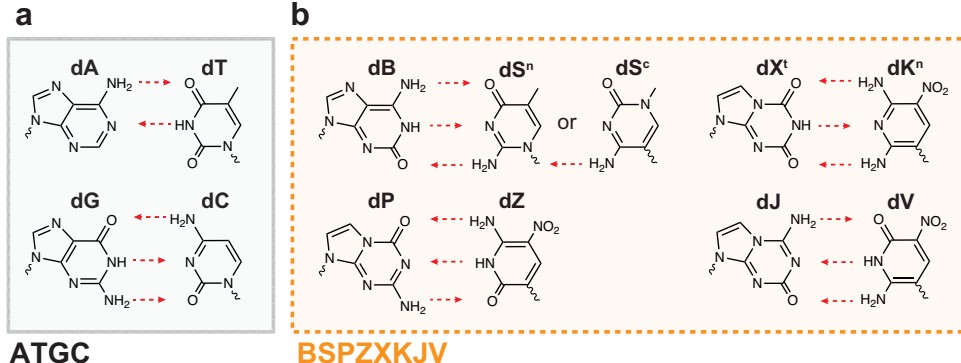

**Fig. 1 | Nucleobases for an expanded 12-letter supernumerary DNA alphabet.**
**a** Structures of standard purine and pyrimidine nucleobases found in life.
**b** Structures of mutually orthogonal synthetic xenonucleobases that could form
the basis of a 12-letter supernumerary DNA. Single letter abbreviations of each base
indicated above nucleobase structure. Arrows indicate hydrogen bonding between
base pairs, drawn in the direction of donor-to-acceptor. S nucleobase has two
possible structures which both base pair with B: the N-nucleoside ($S^n$) and
C-nucleoside ($S^c$).

studied, including stability in the DNA double helix[8], the ability to be
replicated by DNA polymerases[9], transcribed by RNA polymerases[8,10],
reverse transcribed by reverse transcriptases[10], and even translated by
the ribosome[11]. These xenonucleotides are at the forefront of nucleic
acid research since they significantly expand DNA's chemical, struc-
tural, and binding repertoire. Efforts at appropriating expanded DNA
codes have already resulted in more sensitive diagnostics tests[12,13],
highly specific aptamer-based therapeutics that are cheaper and more
soluble than antibodies[14,15], semi-synthetic organisms capable of bio-
manufacturing new molecules[16], catalytic nucleic acids (XNAzymes)
with enzyme-like activity[17], and even denser forms of digital informa-
tion storage[18].

Though the chemistry and biophysics of these additional xeno-
nucleotide pairs have been examined in isolated form, the biomole-
cular tools and commercial infrastructure for sequencing alphabets
comprised of either more than 4 letters or alternative sets of letters are
critically lacking. Notably, methods for routine sequencing of xeno-
nucleic acids (XNAs) are decades behind that of DNA and RNA and rely
on low-throughput, non-multiplexed measurements, such as gel-shift
assays[19,20], mass spectrometry[21], and selective conversion of XNAs to
standard bases followed by Sanger sequencing[22]. This stands in stark
contrast to the state of sequencing for the standard nucleobases (A, T,
G, C), which has a multitude of high throughput, multiplexable, and
low-cost options[23,24]. To put the disparity of sequencing technology in
perspective, XNA sequencing technology is lower throughput, less
sensitive, and less generalizable than the methods Sanger and Coulson
developed in the 1970s and has no service-oriented solution. Con-
versely, ATGC-sequencing technology is in its 'third generation.'

Currently, research and development in the field of XNA face fun-
damental barriers to entry in the form of sequencing, which generally
requires highly specialized equipment and analytical expertise. One
possible solution is to adapt existing first-, second-, or third-generation
DNA sequencing technology to work with more DNA letters. However,
modern sequencing infrastructure is inherently inflexible and highly
specialized for ATGC sequencing. Adapting fluorescence-based DNA
sequencing techniques for XNA sequencing, such as Illumina sequen-
cing, would require a plethora of innovations including new reagents
(e.g., XNA nucleotides with unique fluorophores), engineered poly-
merases capable of replicating XNAs, modification of instrumentation
to handle more cycles, and creation of new data collection/analysis
pipelines. Any fluorescence-based XNA next-generation sequencing
strategy is not realistically attainable at present. As an alternative
approach, other next-generation sequencing methods built for DNA
might be more amenable to serving as XNA sequencing solutions.

Nanopore sequencing has the ability to sequence non-canonical
bases such as epigenetic and epitranscriptomic modifications[25–27]. In
recently published work, co-authors assessed the practicality of
nanopore sequencing for 8-letter hachimoji DNA (A, T, G, C, B, $S^c$, P, Z)
using the Hel308 motor protein with an MspA pore[8,28]. This work cri-
tically established that third-generation (high throughput, multi-
plexable, single molecule, real-time) sequencing of supernumerary
DNA is theoretically possible despite the "k-mer explosion" in possible
current signals induced by an expanded DNA alphabet. As a limitation,
this previous work did not attempt to build models for decoding the
nanopore current signals to nucleic acid sequences. In addition, it was
performed on a non-commercial research platform consisting of a
single nanopore run by a technician (low throughput, non-multi-
plexable). While useful for the development of sequencing technology,
this previous nanopore setup cannot be easily adopted by those in
other fields.

Beyond these efforts on non-commercial devices, others have
approached the classification of non-standard bases using commercial
nanopores (GridION, ONT)[29]. Importantly this previous work showed
that commercial nanopore sequencing platforms are indeed capable
of sequencing chemically modified nucleobases including 2,4-dia-
mino-purine, 5-nitro-indole, and 5-octadiynyldeoxyuracil. However,
orthogonal base-pairing nucleotides were never tested and only a
small sequence space was explored, both of which exclude their
applicability to expanded and evolvable genetic alphabets. For com-
mercial nanopore sequencing to be applicable to 4$^+$-letter genetic
alphabet systems that contain orthogonal XNA base pairs, bespoke
nanopore sequencing models will be required.

Similarly, our ability to synthesize nucleic acids with xenonu-
cleotide base pairs is at least a generation behind modern ATGC-
synthesis technology. Presently, de novo synthesis of DNA with non-
standard base pairs is only possible through phosphoramidite synth-
esis—commercial access is both limited and costly, standing as a major
barrier to entry. For example, standard phosphoramidite synthesis
costs for non-standard bases average around $100–400 USD/nt—or
over 1000 times more expensive than A, T, G, and C synthesis
($0.04–0.40 USD/nt). Furthermore, the next-generation synthesis
methods that have transformed our ability to explore sequence space
(pooled synthesis, synthesis-on-a-chip, enzymatic synthesis) are not
commercially available for orthogonal base pairs. Lowering barriers to
entry for routine synthesis and sequencing of XNAs with orthogonal
base pairs will be required to bring expanded genetic alphabets to the
next-generation eras of synthetic biology, information storage, ther-
apeutic discovery, sequencing, and synthesis.

Here we report progress on both synthesis and sequencing that makes supernumerary DNA sequences containing 6-letter, 8-letter, 10-letter, or 12-letter alphabets easily accessible. In the area of XNA synthesis, we introduce an enzyme-assisted strategy that can be used to incorporate single orthogonal XNA base pairs ($B \equiv S^n$ or $B \equiv S^c$; $P \equiv Z$; $X^t \equiv K^n$; and $J \equiv V$) into synthetic 4-letter DNA. For XNA sequencing, we put theory to practice and develop commercial nanopore base-calling models capable of sequencing single XNA bases ($B$, $S^n$, $S^c$, $P$, $Z$, $X^t$, $K^n$, $J$, and $V$) embedded in a standard DNA (i.e., A, T, G, C only) context.

## Results

### Non-templated XNA tailing by DNA polymerases

Under the supernumerary DNA framework, the two standard base pairs (A = T, G ≡ C) can be combined with any of the four mutually orthogonal base pairs ($B \equiv S^n$ or $B \equiv S^c$; $P \equiv Z$; $X^t \equiv K^n$; and $J \equiv V$) shown in Fig. 1 and Supplementary Table 1. Though phosphoramidite synthesis might seem like a general approach for de novo synthesis of supernumerary DNA, the chemical instability of xenonucleobases $J$, $S^n$, and $X^t$ in organic synthesis has limited sequences to only 8 letters (the hachimoji set: $B$, $S^c$, $P$, $Z$).

To meet the challenge of generalizing the synthesis of supernumerary DNA to include any possible base pair, we were inspired by recent trends in enzymatic synthesis of nucleic acids. Enzymes like terminal deoxynucleotidyl transferase (TdT) have been shown to catalyze non-templated addition of a wide range of modified nucleotide building blocks on ssDNA and can do so at neutral pH[6,30–32]. However, the processive nature of TdT-like enzymes precludes them from being used for sequence-defined addition of dNTPs. More so, TdT-based enzymatic synthesis of nucleic acids would require specially protected building blocks or polymerase-nucleotide conjugates that are not commercially available.

Lacking a suitable alternative, we, therefore, needed to develop an enzymatic synthesis strategy that would be flexible enough to handle all desired xenonucleobases using 2'-deoxynucleoside triphosphates as the universal building block and be specific enough to catalyze a non-processing $N + 1$ addition. We found our solution by exploiting a poorly understood side reaction catalyzed by many DNA polymerases. Over 35 years ago, scientists studying DNA polymerization first reported on the non-templated blunt-end $N + 1$ addition of a nucleotide to the 3'-end of dsDNA by the small fragment of DNA Pol I (small Klenow Fragment or KF exo-)[33,34]. In this reaction, KF catalyzes the addition of a dNTP to the free 3'-OH end of blunt-end dsDNA resulting in a 3' $N + 1$ DNA product. We imagined that if 2'-deoxy-xenonucleoside triphosphates (dxNTPs) could serve as tailing substrates for a synthetic DNA hairpin, the non-processive nature of this reaction would provide a means for a controlled, non-templated, enzymatic semi-synthesis of 6-, 8-, 10-, and 12-letter DNA (Supplementary Fig. 1; hairpins used in this work are listed in Supplementary Tables 2–7). Furthermore, since this strategy avoids the use of environmentally harmful phosphoramidites, it is inherently a green solution to a synthesis problem.

After a campaign of screening, two enzymes were identified as being able to tail both standard DNA purines and pyrimidines to the blunt end of our dsDNA hairpins (Supplementary Fig. 2, Supplementary Table 8). These two enzymes included the originally reported small Klenow Fragment (KF exo-) polymerase and an engineered polymerase from hyperthermophilic marine archaea (engineered 9°N DNA polymerase-Therminator)[35]. Next, we tested the activity on the expanded set of XNA letters. 2'-Deoxy-xenonucleoside triphosphate building blocks for an 8-letter code are readily available from various commercial sources (dBTP, dS$^n$TP/dS$^c$TP, dPTP, and dZTP). To reach the full extent of the 12-letter alphabet, we chemically synthesized the 2'-deoxy-xenonucleoside triphosphates of the remaining bases: dX$^t$TP, dK$^n$TP, dJTP, dVTP (Supplementary Figs. 3–6). Next, we developed a sensitive liquid chromatography/mass spectrometry

(UPLC/QTOF) assay for detecting tailing activity. In this assay, a DNA polymerase and 3' → 5' exonuclease are simultaneously used to perform $N + 1$ tailing and $N + 1$ removal of a dxNTP substrate on an exo-resistant, blunt-end DNA hairpin (Fig. 2a). The net reaction results in formation of dxNMP + PPi, and requires the presence of a 3'-OH blunt-end DNA, exonuclease, DNA polymerase, and dxNTP. From this UPLC/QTOF assay, it was evident that both KF (exo-) and Therminator polymerase were fully capable of non-templated $N + 1$ addition of all four standard dNTPs and all nine dxNTPs tested, including both the N-nucleoside ($S^n$) and C-nucleoside ($S^c$) of S (Fig. 2b, c, Supplementary Figs. 7 and 8).

We next turned our attention to optimizing XNA tailing reaction components and conditions. Of particular concern were competing side reactions such as PPi-mediated pyrophosphorolysis ($N-1$) and consecutive tailing ($N + 2$) (Supplementary Figs. 1 and 9–11). Since $N + 1$-tailed DNA is unable to undergo self or blunt-end ligation, agarose gel-based assays were used to characterize the amount of remaining starting material. Reaction conditions (reaction time, dxNTP concentration, temperature, choice of polymerase, and reaction additives) were optimized around maximizing $N + 1$ tailing and consuming unreacted blunt-end DNA, the latter of which would be a major source of non-specific product formation in subsequent steps. Proving that this strategy would indeed be non-processive, high-resolution UPLC/QTOF assays were then used to show the formation of the $N + 1$ DNA as the main tailing product (Supplementary Fig. 12) under optimized conditions. XNA tailing reaction characterization for each dxNTP required for a supernumerary 12-letter DNA alphabet (dBTP, dPTP, dS$^n$TP or dS$^c$TP, dZTP, dX$^t$TP, dK$^n$TP, dJTP, and dVTP) is shown in Fig. 2d, e, with conditions listed in Supplementary Table 8. Under these optimized conditions for all dxNTPs tested, the extent of reaction is estimated to be >95% (Supplementary Table 9).

### Ligation of XNA overhangs with complementary XNA base pairs

While $N + 1$ tailing can be used for non-templated extension of the 3' blunt-end of DNA, we ultimately desired a *base pair* embedded in a dsDNA sequence. We next developed a strategy for joining two DNA hairpins with complementary $N + 1$ base overhangs to generate xenonucleotide base pairs. Here, we envisioned using dsDNA ligases to catalyze end-to-end joining of two $N + 1$-tailed DNA hairpins with complementary xenonucleotide overhangs (Fig. 2f). The hairpin design of the substrates generates a desired dsDNA ligation product that lacks a free 5' or 3' end, making it fully resistant to exonucleases. Subsequent treatment of the ligation reaction with exonucleases therefore allows us to remove unreacted starting material and partially ligated products.

The ideal dsDNA ligase should be able to ligate DNA strands with only single nucleotide overhangs and have relaxed specificity for both the overhanging nucleotide and its adjacent sequence context. Previous work has shown general promiscuity of phage ligases (T3 DNA ligase, T4 DNA ligase, and T7 DNA ligase) including their ability to ligate modified and non-standard nucleotide substrates (Supplementary Fig. 13)[36–38]. To ensure any ligation product we observe comes from complementary overhang ligation, we performed a negative control in which we incubated hairpins individually in the presence of the respective ligases (Supplementary Fig. 14). In these single hairpin reactions, any ligation product would indicate either blunt-end ligation, from incomplete XNA tailing, or formation of a self-ligation (mismatch ligation) product. Mismatch ligation has previously been observed to arise in conditions where crowding agents in ligation buffer are present, ligase concentration is high, reaction time is long, and is dependent on both the overhanging base and choice of DNA ligase[39]. Taking these constraints into consideration, we screened XNA ligation reaction conditions that would generate the ligation product only when two hairpins with complementary $N + 1$ overhangs are present in the same reaction. Though the sequence context for all ligation

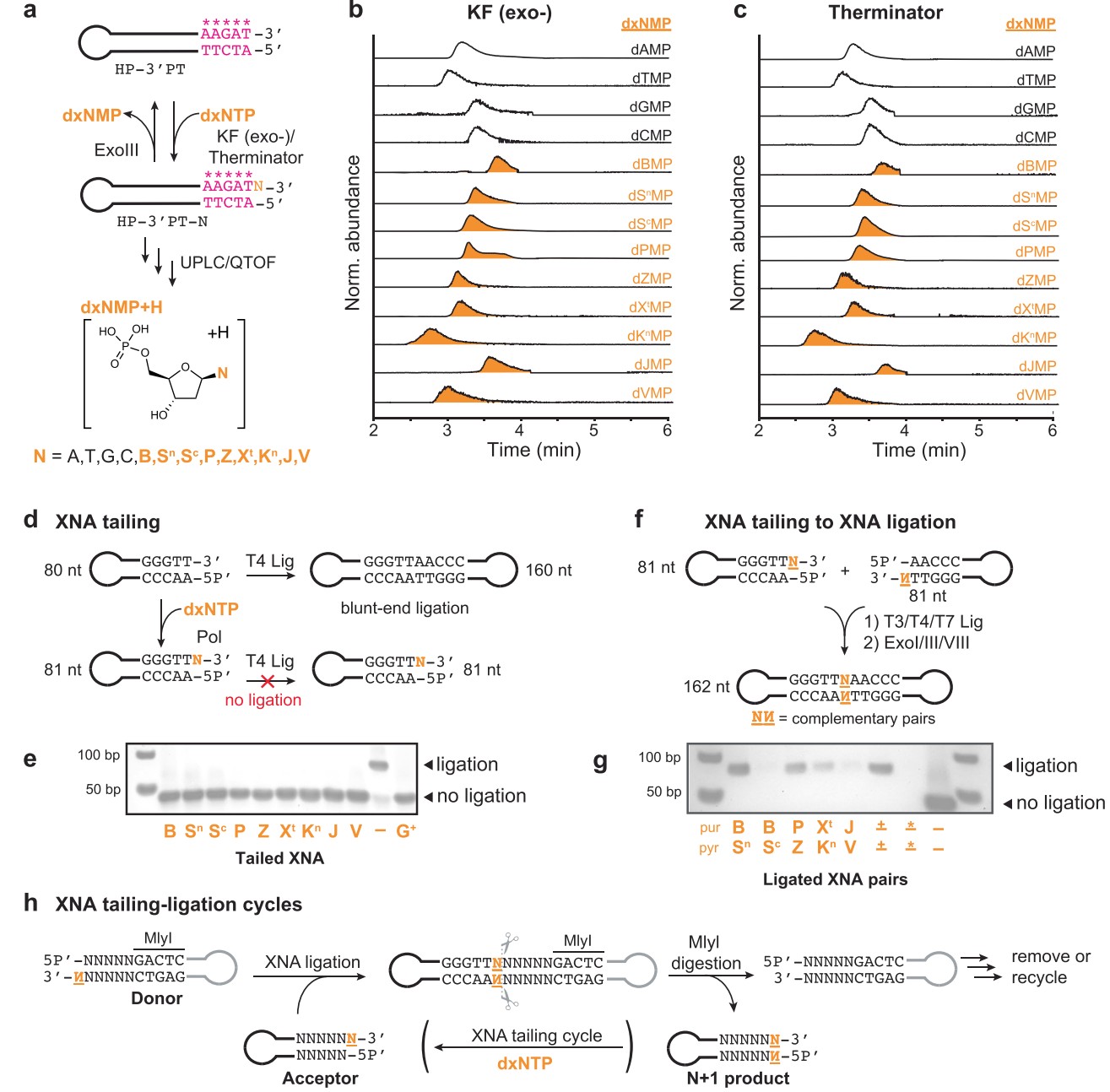

**Fig. 2 | XNA tailing and XNA ligation enable a facile means for enzymatic XNA incorporation. a** Polymerase XNA tailing activity screened by detection of released 2′-deoxy-xenonucleoside monophosphates (dxNMPs). Hairpin HP-3′PT was used as tailing substrate (Supplementary Table 2); '*' indicate positions of phosphorothioate bonds. Extracted ion chromatograms for each dNMP and dxNMP in assays indicate dNTP and dxNTP tailing by **b** Klenow Fragment (exo-) and **c** Therminator polymerase. Source data are provided as a Source Data file. **d** Assay measuring extent of XNA tailing by T4 ligation. Tailed hairpins are not substrates for T4 ligation. **e** XNA tailing of hairpin using optimized conditions showing XNA tailed hairpin is the major product. (−) is blunt-ended hairpin negative control. G⁺ is a hairpin synthesized to contain a single nucleotide 3′-G overhang as the positive control (gel representative of 3 experimental replicates; yield estimates are listed

in Supplementary Table 9). Source data are provided as a Source Data file. **f** Assay to ligate two DNA hairpins with complementary single nucleotide XNA overhangs. Ligated hairpins are protected from exonucleases as they lack free 5′ and 3′-ends. **g** XNA ligation of hairpins tailed with complementary purine (pur) and pyrimidine (pyr) XNA bases using optimized reaction conditions. (±) is a positive control that used blunt DNA substrate. (*) is a negative control that used blunt DNA substrate without DNA ligase. (−) is a negative control without ligase or exonuclease, shown quantitatively for comparison with XNA ligation products (gel representative of 3 experimental replicates; yield estimates are listed in Supplementary Table 11). Source data are provided as a Source Data file. **h** XNA tailing and XNA ligation steps can be cycled for consecutive additions using Type IIS restriction enzyme MlyI.

reactions was the same (Supplementary Table 2), we observed variable ligation yields suggesting the chosen DNA ligases had varying xenonucleotide base pair tolerance. After optimizing reaction conditions for XNA tailing, we show it is indeed possible to incorporate all xenonucleotide base pairs (B:Sⁿ, B:Sᶜ, P:Z, Xᵗ:Kⁿ, and J:V) into DNA with varying yields (estimated yield: B ≡ Sⁿ ≈ 73%; B ≡ Sᶜ ≈ 7%; P ≡ Z ≈ 53%;

Xᵗ ≡ Kⁿ ≈ 31%, and J ≡ V ≈ 15% Fig. 2f, g, Supplementary Figs. 14–17, Supplementary Tables 10, 11). In totality, the described enzyme-assisted synthesis reaction is comprised of these two reactions that use only dxNTPs and commercially available enzymes: (1) xenonucleotide tailing and (2) xenonucleotide ligation. More so, we can propose strategies for extending the scope of these steps beyond singular XNA

base pair insertions. For example, following one round of XNA tailing and XNA ligation, we can add a type IIS restriction enzyme that regenerates blunt-ended starting material, allowing one to perform consecutive dxNTP additions (Fig. 2h, Supplementary Fig. 18).

### Generation of XNA libraries for nanopore model building

From these advances in XNA writing, we next turned our attention to advancing our capacity for XNA reading with a commercial nanopore platform. Nanopore sequencing (from Oxford Nanopore Technology) has features that make it adaptable for sequencing supernumerary DNA: it can sequence single DNA molecules without amplification, without the requirement for fluorescently labeled building blocks, and with high throughput (100k–10M reads per run)[40,41]. In nanopore sequencing, an ion current signal is generated as single-stranded DNA is threaded through a protein nanopore. Conversion of signal-to-sequence, or basecalling, is performed computationally by either statistical or machine learning models[26,29]. However, since commercial nanopore basecalling algorithms were empirically trained on standard 4-letter DNA (A, T, G, C), they are unable to decode our xenonucleobases (B, $S^n$, $S^c$, P, Z, $X^t$, $K^n$, J, V; Supplementary Fig. 19).

With this in mind, we set out to build and measure diverse DNA-XNA libraries that could be used to construct de novo ground-up models for sequencing single xenonucleotides within a natural DNA context. Here, we took inspiration from the early predictive 'kmer models' for nanopore sequencing. In these models, the current signal produced by any given DNA sequence is only a function of the sequence kmer, which consists of the incident nucleotide in the pore and its surrounding nucleotide context[43–45]. Sequencing models built with longer kmer sequences will benchmark with a higher overall accuracy. There is, however, a diminishing return in accuracy improvements as kmer size increases which is balanced against the exponential increase in library complexity and data collection requirements with longer kmers. Balancing performance and complexity, we decided to measure the signal produced by every 4-nt kmer that contains a single xenonucleobase from our set. The synthetic capabilities of XNA tailing and XNA ligation made it possible to generate libraries containing all 4-nt kmers with a single xenonucleotide pair ($4^4 = 256$ kmers per xenonucleotide). To cover the entirety of the 4-nt-long kmer sequence space, we designed a dual-barcoded DNA hairpin library that could be synthesized on a chip (NNN-Pool; Fig. 3a, Supplementary Fig. 20, Supplementary Tables 3–5, 12, 13). To establish

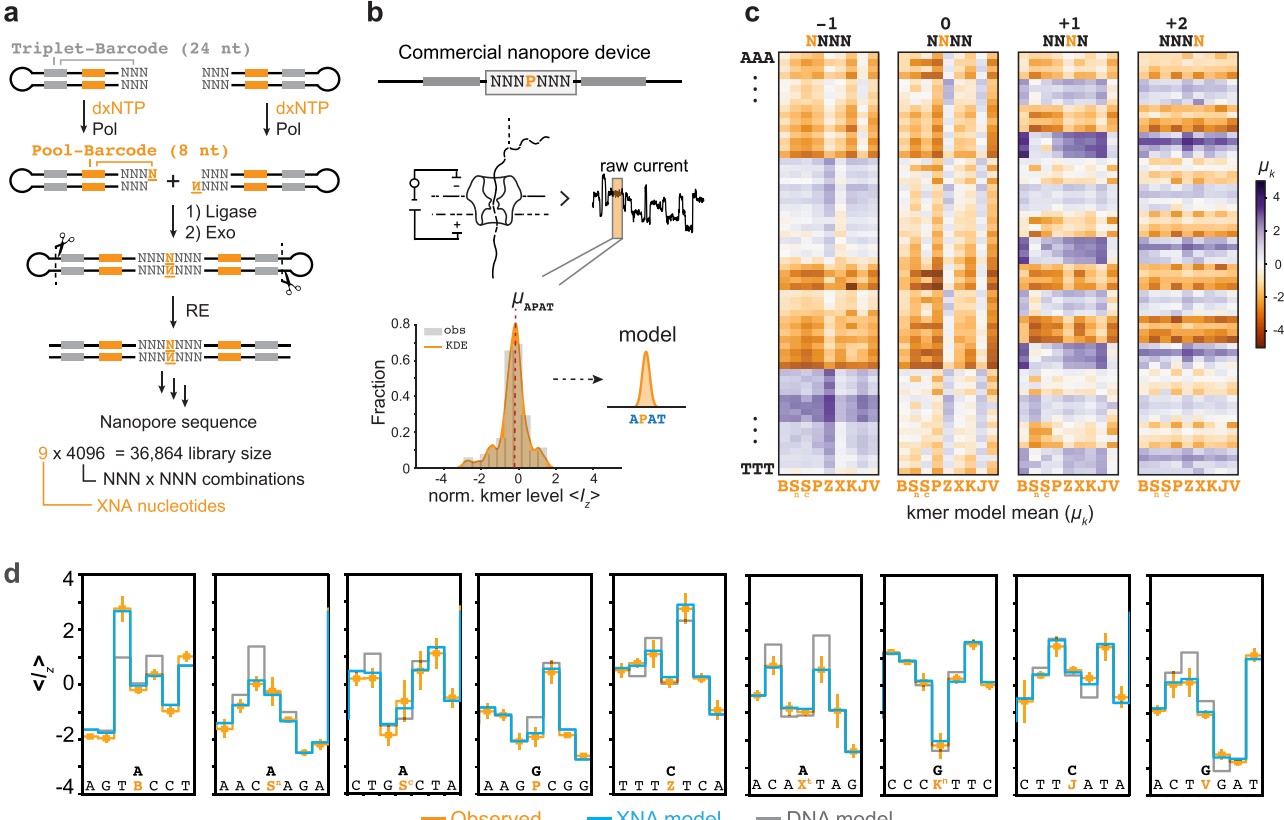

**Fig. 3 | Generation of 12-letter (ATGCBSPZXKJV) nanopore sequencing kmer models. a** Overview of construction of NNNNNNN libraries, starting from two synthetic oligo pools (NNN-Pool) that contain blunt, NNN-3' ends. The 24-nt triplet-barcodes in these hairpins are linked to the 3'-NNN sequence, allowing for proper identification of bases adjacent to XNA inserts. Complementary XNA base pairs are added to the library hairpins using XNA tailing and XNA ligation. The 8-nt pool-barcode is used to identify which XNA was tailed to the 3'-end. Restriction enzymes (RE) remove the hairpin ends. Final libraries contain an XNA base insert in every possible NNN × NNN context (N = A, T, G, C; 64 × 64 = 4096 unique sequences per XNA base). **b** 4-nt kmer models were generated by decomposing every sequenced heptamer (NNNNNNN; N = modified nucleotide) into its corresponding 4-nt kmers. For a kmer's observed current signals, mean values from the observed signal (obs) or from a kernel density estimate (KDE) can be calculated. **c** All measured normalized current signal means ($\mu_k$, 2304 total values from kernel density estimate) for each 4-nt kmer, with positive values in deeper purple and negative values in deeper orange. Heatmaps are binned by kmer position containing the xenonucleobase (−1, 0, +1, or +2). 'N' is denoted in the x-axis and the remaining NNN is denoted by row, sorted alphabetically (AAA to TTT). **d** Example traces overlaying observed mean signal (orange) with expected signals produced by either the XNA model (blue) or a model for standard DNA (gray) kmer model. For the standard DNA model, the most similar standard base chosen for each XNA was determined from empirical observation (Supplementary Figs. 19 and 21). n = number of reads used: B (n = 18); $S^n$ (n = 24); $S^c$ (n = 40); P (n = 32); Z (n = 28); $X^t$ (n = 18); $K^n$ (n = 12); J (n = 14); and V (n = 18). Error bars indicate the standard deviation of the observed normalized signal level.

the ground truth for each read, full factorial *NNN* coverage at the blunt end was linked to a unique 24-nt barcode (Triplet-barcode), while the identity of the tailed xenonucleotide was linked to a unique 8-nt barcode (Pool-barcode). These barcode sequences are distal to the site of XNA ligation and can therefore be decoded through standard ATGC basecalling. Though ligation biases could make it difficult to acquire reads of certain combinations (*NNNNNNN*; *N* = modified nucleotide, *N* = A, T, G, C), only a subset of total sequence space would be required to obtain full coverage of all required 4-nt kmers.

## Building a 4-nt XNA kmer model

Each *NNNNNNN* library was sequenced independently for model building, generating between 150k and 800k raw reads per library (Supplementary Tables 14, 15). We then segmented and aligned signals to each barcoded reference sequence while filtering reads that aligned to possible ligation side products (Fig. 3b, Supplementary Figs. 14 and 21). From these signal-to-sequence alignments, we see XNA-heptamer nanopore signals deviate from the signal expected for a canonical DNA sequence (Supplementary Fig. 22). After binning signal-to-sequence alignments into their constitutive kmers (Supplementary Table 16), these differences can be quantified to give us a measure of how the presence of a xenonucleotide in a sequence produces subtle, yet measurable deviations in the observed normalized current signal levels $<I_z>$ (Supplementary Fig. 23).

These empirical kmer signal distribution measurements formed the basis for our xenonucleotide kmer model. As has been shown previously, we can model the probability that a given 4-nt kmer will produce an ionic signal current as a normal distribution (Fig. 3b)[26]. Example kmer signal distributions for a sample sequence are shown in Supplementary Fig. 24. Mean signal currents spanning all 2304 xenonucleotide-containing kmers, $\mu_k$, are shown in Fig. 3c with comparisons to the most similar standard base shown in Supplementary Figs. 25 and 26.

## Basecalling single xenonucleotide substitutions

Next, we showcase how this model can be applied to predict signals emitted by sequences that contain a single xenonucleotide (B, $S^n$, $S^c$, P, Z, $X^t$, $K^n$, J, or V). For any such sequence, the expected signal is found by the decomposition of a heptamer sequence into its constitutive kmers, then using measured kmer means to model current transitions (e.g., AGT**B**CCT → $[\mu_{AGTB}, \mu_{GTBC}, \mu_{TBCC}, \mu_{BCCT}]$). Figure 3d shows examples of signal-level predictions generated by our model (XNA model) overlayed over observations of that library sequence and the most similar standard-bases-only model (DNA model).

We integrated the 4-nt kmer model into an end-to-end basecaller for single xenonucleotide substitutions (Fig. 4a, Supplementary Fig. 27, Supplementary Tables 17, 18, Supplementary Note 1). For any given set of observed signals, the modeled probability density function can be used to calculate the likelihood that an observed set of signal levels was emitted from a particular sequence. The correct basecall should be the one that has the maximum likelihood of observation. The modularity of the 4-nt kmer model allows us to make a diverse set of comparisons between a xenonucleotide and (1) a standard base (e.g., P vs. G), (2) any of the standard bases (e.g., P vs. A, T, G, C), or (3) any of the full supernumerary letters (e.g., P vs. A, T, G, C, B, $S^c$, Z, $X^t$, $K^n$, J, V).

To test the recall of our XNAs, we used XNA tailing and XNA ligation to enzymatically synthesize a new validation library composed of contextually diverse sequences. In this library, the nucleotide sequences adjacent to the XNA-containing heptamer were diversified making them further removed in sequence space from those used to build the 4-nt kmer models. This validation library was built combinatorially using synthetic hairpin pools as starting material. Each set of hairpins contained 10 unique sequences. To avoid biasing which sequence contexts are chosen for validation, the 20 bp at the

3′-end of each hairpin was designed by randomly selecting standard bases from a uniform probability distribution. Individual hairpin sets were tailed with XNA bases using XNA tailing. Two sets of hairpins with complementary tails could then be ligated, producing a library of 100 possible sequences (10 × 10), with each sequence containing a single XNA base pair. These ligated hairpin libraries were pooled together and sequenced for benchmarking (Fig. 4b, c, Supplementary Fig. 28).

We carried out XNA basecalling model benchmarking by calculating two major performance metrics: recall (true positive rate) and specificity (1−false discovery rate). Recall and specificity were calculated per-read, as a per-read consensus, or as a signal-averaged per-sequence consensus. We found that per-read, the 4-nt kmer model was able to recall between 60% and 87% of XNA nucleotides correctly when comparing against the respective most similar standard base (Supplementary Tables 19–21). By consensus basecalling of at least 10 reads (per-read consensus), correct sequence recall for all XNA sequences ranged from 63% to 99%. In an all-by-all comparison of the validation sequences, the 4-nt kmer model had sufficiently high recall to properly basecall each non-standard base as the per-read consensus (Fig. 4b, S = $S^c$). To determine specificity, we tested basecalling using the 4-nt kmer model against a standard DNA library (i.e., A, T, G, C only). Of note, per-read specificity was found to be high, ranging from 80% to 93% (per-read) and 89–99% (per-read consensus). ROC curves generated for XNA vs. most similar standard base comparisons indicate overall high performance despite simplicity of the 4-nt kmer model, with values for area under the curve between 0.8 and 0.96 (Fig. 4c). Additional recall and specificity benchmarking for the kmer models, including per-read consensus and per-sequence recall/specificity, are summarized in Supplementary Tables 19–21.

As an example of how these sequencing models can be applied to accelerate XNA research, we then revisited a landmark experiment on PCR development for the P ≡ Z base pairs carried out over a decade ago[22]. Originally, analysis of successful P ≡ Z amplification was carried out using low throughput agarose gel electrophoresis assays. Showcasing the leap to the NGS era, in a single multiplex nanopore run we show how the PZ kmer models enable simultaneous measurement of PCR amplification efficiency for a P ≡ Z base pair amplified under various dxNTP and dNTP concentrations (Supplementary Fig. 29, Supplementary Tables 22, 23). In agreement with this earlier work, our sequencing results show near complete retention of P ≡ Z base pair using optimized dxNTP (0.6 mM dPTP; 0.05 mM dZTP) and dNTP (0.1 mM dATP, dGTP, dTTP; 0.6 mM dCTP) concentration, with increasing loss of P ≡ Z bases as dxNTPs become limiting. Given the throughput of nanopore flow cells (1–10M reads, MinION flow cell), it should now be possible to use nanopore sequencing to screen PCR replication efficiency across hundreds to thousands of conditions (e.g., polymerase mutants, buffer composition, dxNTP/NTP concentrations) simultaneously.

## Synthesis and sequencing of 12-letter DNA

Thus far, we have shown that (1) enzyme-assisted synthesis can be used to add a single xenonucleotide base pair and (2) 4-nt kmer models can properly basecall individual xenonucleotides with high recall and specificity. We next show a proof of principle that takes the methods developed in this work to their alphabetical limits—synthesizing and sequencing DNA that contains a full 12-letter code: A, T, G, C, B, $S^n$ or $S^c$, P, Z, $X^t$, $K^n$, J, and V (Fig. 5, Supplementary Table 24). Using synthetic 4-letter DNA as a starting point, the elementary tailing and ligation synthesis steps were coupled with an additional Golden Gate ligation to generate two proof-of-concept 12-letter supernumerary dsDNA

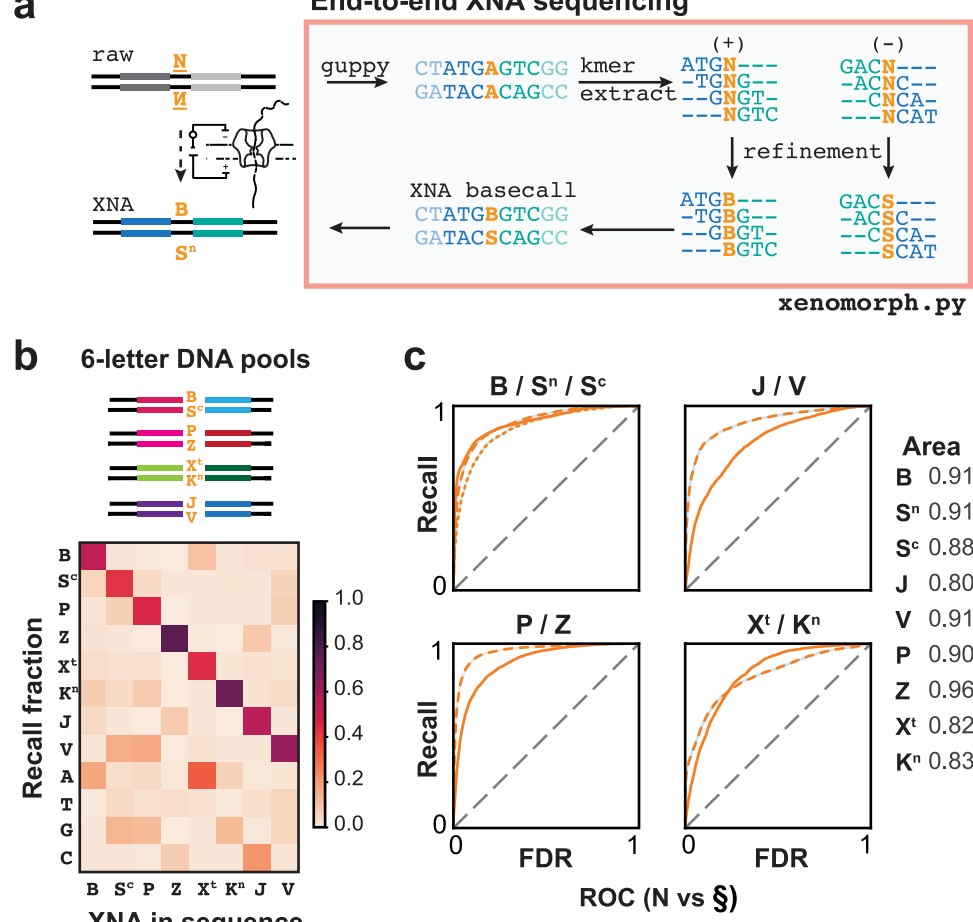

**Fig. 4 | Construction and end-to-end nanopore sequencing of 6-letter DNA alphabets. a** Proof of concept deployment of an XNA-refinement pipeline using 4-nt kmer models measured in this work. Pipeline is used to transform raw commercial nanopore reads into likely XNA basecalls for the sense (+) and antisense (−) strands. **b** Confusion matrix showing per-read recall of the validation libraries (Supplementary Fig. 28) using the full 12-letter supernumerary DNA model ($n = 5000$ reads of each 6-letter set). Example shown with S = $S^c$ kmer model used to analyze BS$^c$ reads. **c** Receiver operating characteristic (ROC) curve plots recall versus false discovery rate (FDR) for 4-nt kmer model basecalling of each XNA in the validation libraries, performing comparison between XNA (**N**) and most likely guppy basecall from the natural bases (**§**); (B, J, P, $X^t$ = solid line; $S^n$, V, Z, $K^n$ = dash line; $S^c$ = dotted line). Legend shows the area under the curve for each base. Source data are provided as a Source Data file. Additional benchmarking of 4-nt XNA kmer models is tabulated in Supplementary Tables 19–21.

hairpins: $S^c$uper-12 and $S^n$uper-12 (Supplementary Fig. 30, Supplementary Tables 7, 12, and 13). In the construction procedure, exonucleases are added to remove intermediary DNA products generating the desired 244 bp 12-letter dsDNA product. In this proof-of-concept example, basecalling was performed two different ways: (1) by comparing the XNA base at a position against a model that contains all 12 possible nucleobases, and (2) by comparing the XNA base at a position against a model that contains only the XNA and the most similar standard nucleobase. Even when all 12 letters are present in the model, the basecalling model was able to properly decode XNAs in $S^c$uper-12 with 39–89% per-read recall (Fig. 5, Supplementary Tables 25, 26). For the $S^n$uper-12 sequence, all but one XNA was properly decoded in the 12-letter model, with the exception being $K^n$ (per-read recall of 14%). When only performing most similar standard base comparisons, all XNAs in $S^n$uper-12 were properly recalled (67–93% per-read recall). Indeed, given the complexity of possible current signals when 12-letter models are invoked for basecalling, one should expect the chosen 12-letter-containing sequence to have a large influence on recall (Supplementary Figs. 31 and 32, Supplementary Tables 27–30, Supplementary Note 2). Despite only being a proof of concept, this foray into 12-letter DNA space represents an important milestone, showing that DNA containing six orthogonal base pairs can be synthesized and sequenced.

## Discussion

We have demonstrated a general strategy for incorporating up to four additional orthogonal base pairs into standard DNA, and used these methods to build openly accessible models for sequencing XNAs (B, $S^n$, $S^c$, P, Z, $X^t$, $K^n$, J, V) in a standard DNA context (A, T, G, C) on commercial nanopore devices. The enzymatic synthesis strategy we developed utilizes unmodified 2′-deoxy-xenonucleoside triphosphates as the elementary building blocks, critically avoiding the use of phosphoramidites or caged-triphosphates. To further eliminate barriers to entry, we have benchmarked 4-nt kmer sequencing models and showed that simultaneous basecalling of 6-letter and 12-letter DNA is possible. This latter development brings XNA base pair sequencing from the "zeroth-generation" of sequencing to the third-generation sequencing era.

Nanopore sequencing of XNAs as implemented here can be performed by any lab anywhere using a commercially available device, significantly expanding the accessibility of sequencing XNAs. As history in sequencing progress has shown, additional widespread adoption and collection of XNA nanopore sequencing data will help further catalyze the improvement of sequencing models with newer basecalling algorithms, including data-intensive deep learning models. As these methods improve and adoption widens, strategies for synthesis and sequencing of higher complexity nucleic acids will also become

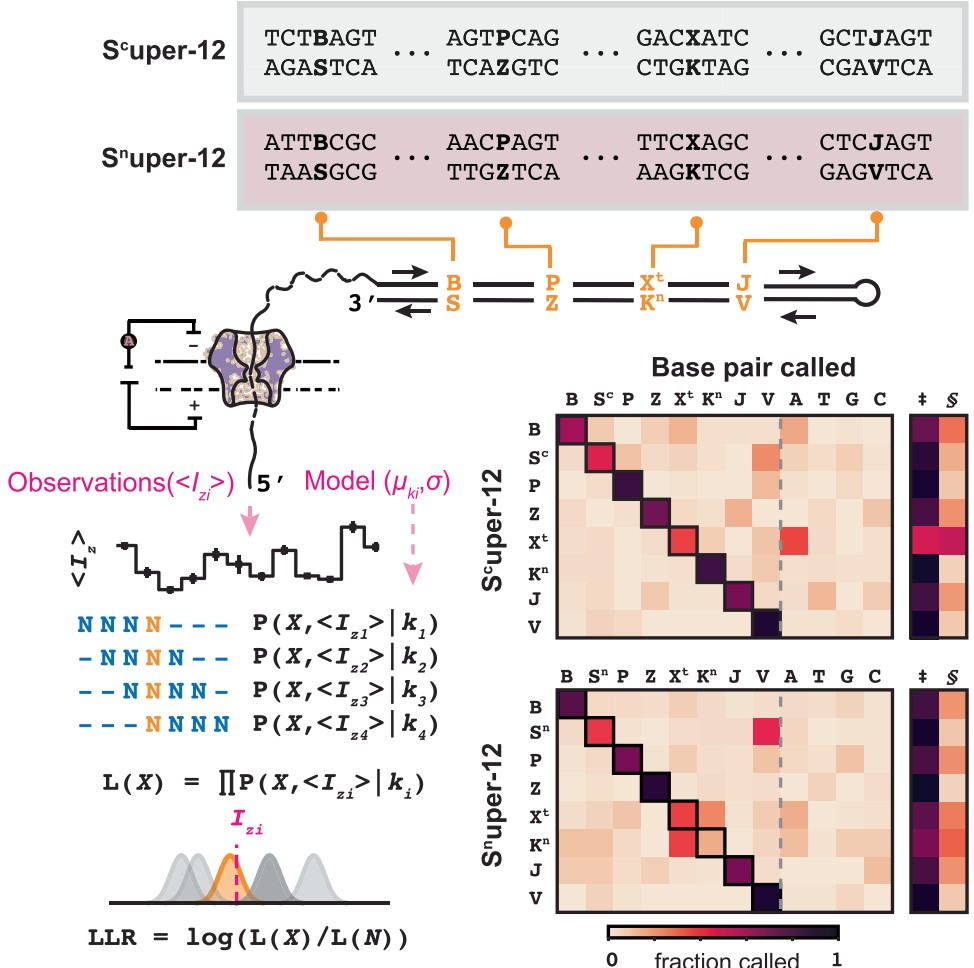

**Fig. 5 | Enzyme-assisted synthesis and third-generation sequencing of super-numerary 12-letter DNA.** Enzyme-assisted synthesis was used to construct two supernumerary 12-letter dsDNA hairpins with either S = $S^c$ ($S^c$uper−12) or S = $S^n$ ($S^n$uper−12) bases. Sequenced reads are processed to produce signal-to-sequence alignments and subsequently segmented into their corresponding kmer sequences and kmer signals. The kmer probability density function (observed signal mean $<I_z>$, model mean $\mu_{ki}$, model standard deviation $\sigma$) is used to calculate log-likelihoods while a maximum likelihood with outlier-robust log-likelihood ratios is used to determine base call. Confusion matrices show (left) fraction of base called at each xenonucleotide position in $S^c$uper−12 ($n = 824$ reads) and $S^n$uper-12 ($n = 1438$ reads); (right) base called using model with simplified priors (‡ = xeno-nucleotide at the position called, $\mathbb{S}$ = most similar standard base called). Box denotes base pair called from paired analysis. Values of confusion matrices are tabulated in Supplementary Tables 25, 26.

possible. For example, variations of XNA tailing and XNA ligation that would allow one to incorporate multiple consecutive XNA bases, such as the example of MlyI cycling (Supplementary Fig. 18), would create an opportunity for 4-nt kmer models with multiple xenonucleobases present in close proximity.

In the immediate future, the generalizability of the synthesis approach eliminates many barriers to accessing site-specifically modified DNA for applications in therapeutics, biomaterials, and genetic engineering, all while bringing supernumerary genetics to the third generation of sequencing. In the area of genetic code expansion, a single insertion of these additional base pairs allows for various arrangements of up to 448 possible codon-anticodon pairs (made up of 64 canonical codons and 96 additional codon-anticodon pairs for each XNA base pair, constrained to one XNA per codon or anticodon). In the design and discovery of DNAzymes/aptamers, an additional base pair enables site-specific incorporation of chemically modified groups, including the addition of nucleobases such as Z that can act as a Brønsted base. In the realm of DNA digital information storage, these additional bases markedly increase information density as we can encode from $\log_2(4) = 2$ bits per base to $\log_2(12) = 3.58$ bits per base. Beyond the 12-letter DNA alphabet presented in this work, the described enzyme-assisted synthesis strategies and nanopore

sequencing pipeline will likely persist as low barrier points-of-entry for writing and reading with other modified nucleotides—including epigenomic and epitranscriptomic modifications. The dual synthesis and sequencing capabilities presented here open a frontier for openly accessible reading and writing DNA with up to 12 letters and aid us in studying what might be possible when we operate at the limit of Nature's rules for hydrogen bonding base pairing.

## Methods

### Commercial materials

Agarose (0710-500G; electrophoresis grade) was purchased from Thermo Fisher Scientific (Waltham, MA). Adenosine triphosphate sodium salt (ATP; A6419-5G), acetonitrile (A955-4; LC/MS-grade), formic acid (A118P-500), ammonium acetate (A637-500), ammonium carbonate (207861-25G), Tris base (10708976001), 5M betaine solution (B0300-1VL), 6 N hydrochloric acid (1430071000), GelGreen (SCT124), and sodium chloride (S3014-5KG) were purchased from Sigma-Aldrich (St. Louis, MO). AMPure XP beads (A63880) were purchased from Beckman Coulter (Brea, CA). Restriction enzymes, T4 DNA ligase, high concentration T4 DNA ligase (M0202M, M0202L), T7 DNA ligase (M0318L), T3 DNA ligase (M0317S), yeast inorganic pyrophosphatase (YiPP; M2403L), thermolabile proteinase K (P8111S), Exo

III (M0206L), thermolabile Exo I (M0568L), Exo I (M0293L), Exo VII (M0379L), Exo VIII (truncated; M0545S), Klenow Fragment (exo-; M0212L), Taq polymerase (M0267L), Bsu polymerase (M0330S), Deep Vent (exo-) polymerase (M0259S), Bst polymerase (M0275S), *Sulfolobus* DNA polymerase IV (M0327S), Therminator polymerase (M0261L), NEBNext® Ultra™ II End Repair/dA-Tailing Module (E7546S), 50 bp DNA ladder (N3236S), NEBNext® Quick Ligation Module (E6056S), and 2′-Deoxynucleoside triphosphates (dNTPs; $N$ = A, T, G, C; N0446S) were purchased from New England Biolabs (Ipswich, MA). Gene ruler 1 kb plus DNA ladder (SM1331) was purchased from Invitrogen (Carlsbad, CA). Oligonucleotides and oligo pools were purchased from Integrated DNA Technologies (Coralville, IA), resuspended at a stock concentration of 100 μM in elution buffer (10 mM Tris−HCl, pH = 8.2) and stored at either 4 °C for immediate usage or −20 °C for long-term storage. 2'-Deoxyxenonucleoside triphosphates dS$^c$TP, dPTP, dZTP, dBTP (dSTP-401S, dPTP-201, dZTP-101, dBTP-301P) were purchased from Firebird Biomolecular Sciences LLC (Alachua, FL). Xenonucleoside triphosphate dS$^n$TP (M-1015) was purchased from TriLink Bio-Technologies (San Diego, CA). DNA purification kits (ZD4034, ZD7011) Zymo Research (Irvine, CA). Flongle Flow Cell R9.4.1 (76521-802) and MinION Flow Cell R9.4.1 (76487-106) were purchased from VWR (Radnor, PA). MinION sequencing device (MIN-101B), Flongle Adapter, Ligation Sequencing Kit (SQK-LSK110), and Flongle Sequencing Expansion kit (EXP-CTL001) were purchased from Oxford Nanopore Technologies (ONT; Oxford, United Kingdom). Unless otherwise specified, other commodity chemicals used in this work were purchased from major domestic distributors (Sigma-Aldrich, St. Louis, MO; Thermo Fisher Scientific, Waltham, MA).

## Polymerase-exonuclease coupled assays to measure tailing activity via nucleoside monophosphate release

A hairpin oligo with five consecutive phosphorothioate bonds on the 3′ end was purchased from IDT (HP-3′PT, Supplementary Table 2). Prior to tailing, 10 μM of HP-3′PT was incubated with rCutSmart™ and 200 units of Exo III at 37 °C for 2 h to digest hairpins, then cleaned using the Zymo ssDNA/RNA Clean & Concentrator Kit and eluted in 15 μL of elution buffer. The eluted oligo was then folded in 100 mM of NaCl and 10 mM Tris−HCl (pH 8.2) buffer by incubating at 90 °C for 3 minutes, then cooling at 0.1 °C/s until reaching 20 °C. 15 μL of this refolded oligo was incubated with 0.17 mM dNTP or dxNTP, 300 units of Exo III and either KF (exo-) with rCutSmart™ buffer or Therminator with ThermoPol® buffer for 16 h. For reactions using KF, the reaction was incubated with 15 units of KF at 37 °C. For reactions using Therminator, the reaction was incubated with 6 units of Therminator at 48 °C. Samples were then prepared for UPLC/MS-QTOF using the methods described in "General procedure for high-resolution HPLC/MS analysis of polar 2′-deoxynucleotides".

## Assays to measure tailing extent through HPLC/MS analysis of oligonucleotides

A hairpin oligo was purchased from IDT (5′Phos-ScaI-HP, Supplementary Table 2). In these reactions, oligos are first refolded by incubating 40 μM of oligo in a 100 mM NaCl, 10 mM Tris−HCl buffer (pH 8.2) at 90 °C for 3 min then cooling at 0.1 °C/s until reaching 20 °C. The refolded oligos are then tailed by incubating 23.8 μM of oligo in the presence of dNTP or dxNTP (1.19 or 2.38 mM), YiPP (0.005 U/μL; except for the dATP tailing reaction which did not contain YiPP), polymerase (0.71 U/μL Klenow Fragment (KF exo-), 0.29 U/μL Therminator polymerase, or 0.71 U/μL Taq polymerase), and polymerase buffer (either rCutsmart™ or ThermoPol® buffer). Full conditions are tabulated in Supplementary Table 8. Reactions were either incubated for 8 h at 37 °C (KF exo-); 1, 4, 8, or 16 h at 60 °C (Therminator); or 1 h at 60 °C (Taq). Following incubation, KF exo- reactions were terminated by heat inactivation at 72 °C for 20 min. Therminator and Taq reactions were terminated by the addition of 1X rCutSmart™ buffer and 0.005 U/

μL of thermolabile proteinase K at 37 °C for 15 min, followed by subsequent heat inactivation at 72 °C for 20 min. Following either set of heat inactivation steps, hairpins were refolded. Afterward, 19.8 μM of oligo was incubated with 1.8 U/ μL of ScaI-HF at 37 °C for 2 h, followed by subsequent heat inactivation at 80 °C for 20 min. Samples were then prepared for UPLC/MS-QTOF using the methods described in "General procedure for high-resolution HPLC/MS analysis of oligonucleotides."

## XNA tailing conditions and reaction components

5′-phosphorylated hairpin oligos with either a 3′-blunt end (5′Phos-11HP) or 3′-single nucleotide overhangs (G: 5′Phos-HP-3′G; or C: 5′Phos-HP-3′C) were purchased from IDT (Supplementary Table 2). For tailing dNTP and dxNTP nucleotides to 3′-blunt ends, 5′Phos-11HP oligo was used as the substrate. In these reactions, oligos are first refolded by incubating 20 μM of oligo in a 100 mM NaCl, 10 mM Tris-HCl buffer (pH 8.2) at 90 °C for 3 minutes then cooling at 0.1 °C/s until reaching 20 °C. The refolded oligos are then tailed by incubating 11.9 μM of oligo in the presence of dNTP or dxNTP (1.19 or 2.38 mM), YiPP (0.005 U/μL; except for the dATP tailing reaction which did not contain YiPP), polymerase (0.71 U/μL Klenow Fragment (KF exo-), 0.29 U/μL Therminator polymerase, or 0.71 U/μL Taq polymerase), and polymerase buffer (either rCutsmart™ or ThermoPol® buffer). Reactions were either incubated for 8 h at 37 °C (KF exo-); 1, 4, 8, or 16 h at 60 °C (Therminator); or 1 h at 60 °C (Taq). Following incubation, KF exo- reactions were terminated by heat inactivation at 72 °C for 20 min. Therminator and Taq reactions were terminated by the addition of 0.005 U/μL of thermolabile proteinase K at 37 °C for 15 min, followed by subsequent heat inactivation at 72 °C for 20 min. Following either set of heat inactivation steps, hairpins were refolded. Resulting hairpins contained a mixture of the product (tailed hairpins) and unreacted starting material (3′-blunt end hairpins). T4 DNA ligase was then used to screen reactions for remaining unreacted 3′-blunt ends by adding 80 U/μL of T4 DNA ligase alongside 1X T4 DNA ligase reaction buffer. These T4 ligation reactions were incubated at 25 °C for 2 h, after which T4 ligase was heat inactivated at 65 °C for 10 min. As a positive control for the tailing reaction, a synthetic oligo hairpin with a 3′-G overhang (5′Phos-HP-3′G, Supplementary Table 2) was used in the T4 ligation reaction. As a negative control for tailing, the starting material (5′Phos-11HP) was used in the T4 ligation reaction. Reaction products were run on a 2% (w/v) agarose gel, stained with GelGreen, and visualized using a blue light transilluminator. Optimized conditions for tailing each substrate are tabulated in Supplementary Table 8.

## XNA ligation conditions and reaction components

5′-phosphorylated blunt-ended hairpin oligo (5′Phos-11HP; Supplementary Table 2) was tailed with either dNTPs or dxNTPs using conditions described in "XNA tailing conditions and reaction components." Following the tailing reaction, two sets of tailed hairpin oligos with complementary nucleotide overhangs (XNA complementary bases shown in Supplementary Table 1) were ligated by incubating oligos (2.4 μM of each, except for the B:S$^c$ base pair which was 1.3 μM of each) in a reaction containing a DNA ligase (either 272 U/μL of T3 DNA ligase; 36 U/μL of T4 DNA ligase; 272 U/μL or 750 U/μL of T7 DNA ligase) and 1X NEB StickTogether™ buffer, which contains 7.5% (w/v) PEG 6000, for 16 h at 16 °C. Following incubation, all ligation reactions were heat-inactivated at 65 °C for 10 min. The desired product possesses no free 3′-OH end, making it resistant to 3′-exonuclease treatment. Unreacted hairpins or incomplete ligation products were removed by exonuclease treatment performed at 37 °C for 1 h and using a combination of: 7.7 U/μL of Exo III; 1.5 U/μL of thermolabile Exo I or Exo I; 0.4 U/μL or 0.77 U/μL of Exo VIII (truncated). Exonuclease reactions were heat-inactivated by incubation at either 80 °C for 20 min (for reactions containing Exo I) or at 70 °C

for 20 min (for reactions containing thermolabile Exo I). Reaction products were run on a 2% (w/v) agarose gel, stained with Gel-Green, and visualized using a blue light transilluminator.

### Consecutive insertion of XNA base pairs using MlyI type IIS restriction enzyme

5′-phosphorylated hairpin oligos were purchased from IDT (5′Phos-11HP, 5′Phos-15HP, and 5′Phos-ScaI-HP; Supplementary Table 2). 5′-Phos-15HP contains an MlyI restriction site adjacent to site of XNA ligation. MlyI is a type IIS restriction enzyme (5′-GAGTCNNNNN ↓ -3′) that leaves a blunt end after cutting. 5′Phos-15HP (donor hairpin with MlyI site; abbreviated $HP_D$) and 5′Phos-11HP (acceptor hairpin; abbreviated $HP_A$) were tailed with P and Z, respectively, generating $HP_D$-P and $HP_A$-Z. These two hairpins were then ligated and subsequently treated with exonuclease following the optimized conditions described in "XNA ligation conditions and reaction components." This material was purified using Zymo's DNA Clean and Concentrator and eluted in 30 μL of elution buffer. The purified construct contains a single P ≡ Z base pair insertion and was digested using 1.24 U/μL of MlyI and 1X rCutSmart™ buffer at 37 °C for 2 h then heat-inactivated at 65 °C for 20 min. MlyI digestion results in a hairpin with a terminal P ≡ Z, which also possesses the required termini for another tailing reaction (5′-PO₄ and 3′-OH blunt end). In the second round of cycling, this hairpin (which already contained a P ≡ Z base pair) was subjected to Z-tailing generating $HP_A$-ZZ. A third hairpin (5′Phos-ScaI-HP, lacks MlyI site, abbreviated $HP_P$) was P-tailed to generate $HP_P$-P. Hairpins were then ligated, and incomplete ligation products were removed by adding 1 U/μL of MlyI, 7.7 U/μL of Exo III, 1.5 U/μL of thermolabile ExoI, and 0.77 U/μL of ExoVIII (truncated) and incubating at 37 °C for 1 h, followed by a heat inactivation step at 72 °C for 20 min. Products were analyzed on a 2% (w/v) agarose gel, stained with GelGreen, and visualized using a blue light transilluminator.

### General procedure for high-resolution HPLC/MS analysis of polar 2′-deoxynucleotides

Prior to HPLC/MS analysis, samples were mixed with an equal volume of 4% formic acid in methanol (v/v) and centrifuged at 20,000×g for 10 min at room temperature. Soluble fraction of the resulting sample containing nucleoside or nucleoside mono, di, or triphosphates were analyzed using an Agilent 1290 Infinity II Bio UPLC on a HILIC-Z column (Poroshell 120 HILIC-Z 2.7 μm, 2.1 mm × 50 mm; Agilent) using the following mobile phases: Buffer A (20 mM ammonium carbonate in water) and Buffer B (100% acetonitrile) at room temperature. A linear gradient from 85% to 20% Buffer B over 7.5 min followed by a linear gradient from 20% to 10% Buffer B over 1 min was applied at a flow rate of 0.3 mL/min. Mass spectra were acquired in positive ionization mode using an Agilent 6530C QTOF (2 GHz) mode with the following source and acquisition parameters: gas temperature 300 °C; drying gas 8 L/min; nebulizer 35 psi; capillary voltage 3500 V; fragmentor 175 V; skimmer 65 V; oct 1 RF vpp 750 V; acquisition rate 1 spectrum/s; acquisition time 1 s/spectrum.

### General procedure for high-resolution HPLC/MS analysis of oligonucleotides

Prior to HPLC/MS analysis, samples were mixed with 0.85 volumes of 4% formic acid in methanol (v/v) and centrifuged at 20,000×g for 10 min at room temperature. Soluble fractions of the resulting sample containing oligonucleotides were analyzed using an Agilent 1290 Infinity II Bio UPLC on a HILIC-Z column (Poroshell 120 HILIC-Z 2.7 μm, 2.1 mm × 100 mm; Agilent) using the following mobile phases: Buffer A (15 mM ammonium acetate in 70% water and 30% acetonitrile) and Buffer B (15 mM ammonium acetate in 30% water and 70% acetonitrile) with the column at 30 °C. A linear gradient from 85% to 60% Buffer B over 10 min followed by a linear gradient from 60% to 40% Buffer B over 2 min was applied at a flow rate of 0.4 mL/min. Mass spectra were

acquired in positive ionization mode using an Agilent 6530C QTOF (4 GHz) mode with the following source and acquisition parameters: gas temperature 350 °C; drying gas 13 L/min; nebulizer 35 psi; capillary voltage 4,500 V; fragmentor 180 V; skimmer 65 V; oct 1 RF vpp 750 V; acquisition rate 1 spectrum/s; acquisition time 1 s/spectrum.

### NNNNNN library design

5′-phosphorylated oligo pools (purchased as oPools™ from Integrated DNA Technologies) were designed to form blunt-end hairpins with two barcodes: a 24-nt Triplet-barcode [NNN-BC] and an 8 nt pool-barcode [Pool-BC] (Fig. 3a, Supplementary Tables 3–5). The Triplet-barcode is linked to the NNN sequence at the 3′-blunt end of the hairpin, while the pool-barcode is used to decode which dxNTP/dNTP was tailed (Supplementary Table 12). Each Triplet-barcode maps 1:1 with a corresponding NNN sequence adjacent to an XNA base. Each NNN-oligo pool contained 64 ($NNN = 4^3 = 64$) unique sequences and was synthesized at a scale of 50 pmol/oligo. Ligation reactions for libraries generate combinations of two different pool barcodes. Restriction enzyme cut sites were included upstream of Triplet-barcodes to remove hairpins following ligation reactions and prepare DNA for nanopore sequencing. Full hairpin sequences in each library can be found in Supplementary Data 2.

### Val-20 validation library design

5′-phosphorylated oligo pools (purchased as oPools™ from Integrated DNA Technologies) were designed to form blunt-ended hairpins with a variable 20-nt region at the end (Supplementary Tables 3, 6). The variable 20-nt region was designed computationally by randomization with a uniform prior probability for each base. Candidate sequences were passed through the IDT oligo analyzer tool to remove sequences that might form secondary structures that could disrupt hairpin formation. Each validation oligo pool contained 10 unique sequences (six total pools: Val_A-F; Supplementary Table 6) and was synthesized at a scale of 50 pmol/oligo. Two different validation oligo pools can be tailed with a dxNTP. Ligating two pools together (with complementary N + 1 tails) results in a library with 100 possible sequences (10 × 10 combinations). Restriction enzyme cut sites were included upstream of these variable regions for nanopore library preparation following ligation. Validation libraries containing different XNA base pairs were prepared in independent reactions and pooled together for sequencing; a full list of these sequences can be found in Supplementary Data 2. Schematic of the library construction is shown in Supplementary Fig. 28.

### 12-letter DNA design

5′-phosphorylated oligos were designed to form blunt-ended hairpins with a barcode sequence (Supplementary Table 7). The barcode is linked to a variable 3-nt sequence at the 3′-end, as well as the dxNTP tailed to the blunt 3′-end. Oligos can be tailed with a dxNTP and ligated to a complementary pair forming a sequence with a single xenonucleotide base pair insertion. By including BbsI Golden Gate sites, four single insertion constructs could then undergo Golden Gate ligation to form a single dsDNA sequence containing all 12 letters (4 standard nucleotides and 8 xenonucleotides). To remove any intermediary 6-letter, 8-letter, or 10-letter DNA products, unsuccessfully assembled hairpins can be digested by restriction exonucleases. The assembled product contains two different restriction sites for hairpin removal, 5′-GATATC-3′ (EcoRV) and 5′-AGTACT-3′ (ScaI). Asymmetric presence of restriction sites on the hairpins allows us to remove a singular hairpin and therefore generate a blunt end on the assembled product. The resulting dsDNA contains only a single 3′- and 5′-end. Subsequent library preparation and sequencing of dsDNA results in reads where both sense and antisense strands, containing all 12-nucleobases, can be read in a single sequencing event (Sᶜuper-12 and Sⁿuper-12; Fig. 5, Supplementary Fig. 30).

### *NNNNNNN* library, validation library, and 12-letter DNA preparation by XNA tailing and XNA ligation

For tailing dxNTP to the 3′-end of oligo pools (*NNN*-oligo pools, Val-oligo pools, Supplementary Table 3) or 12-letter DNA oligos (HP12, Supplementary Table 7), oligos were first refolded by incubating 20 μM of oligo pool in a 100 mM NaCl, 10 mM Tris−HCl (pH 8.2) buffer at 90 °C for 3 min then allowing for cooling at 0.1 °C/s until reaching 20 °C. After refolding, oligos or oligo pools were tailed with a corresponding dxNTP using tailing conditions listed in Supplementary Table 8. Reactions tailed with KF exo- were heat-inactivated, while those tailed with Therminator were inactivated by thermolabile proteinase K treatment. Following the inactivation of polymerase, oligos were refolded. Tailed oligo or oligo pools with complementary 3′-ends were then ligated with either T4 DNA ligase, T3 DNA ligase, or T7 DNA ligase using ligation conditions listed in Supplementary Table 10. As a negative control for tailing, the starting material 3′-blunt end oligo or oligo pool (e.g., HP_v1-*NNN*-P1; Val_A; HP12-A1) was used. All ligation reactions were heat-inactivated at 65 °C for 10 min. Following ligation, unreacted hairpins or incomplete ligation products were removed by adding 7.7 U/μL of Exo III (3′→5′ dsDNA exonuclease), 1.5 U/μL of thermolabile Exo I (3′→5′ ssDNA exonuclease), and 0.77 U/μL of Exo VIII (truncated, 5′→3′ dsDNA exonuclease) and incubating at 37 °C for 1 h, followed by a heat inactivation step at 72 °C for 20 min. This combination of exonucleases was used for rapid undesired product removal, but other exonuclease combinations could also accomplish the same goal.

For *NNNNNNN* library preparation, ligated *NNN*-oligo pool reactions were then purified using Zymo DNA Clean and Concentrator and eluted in 30 μL of elution buffer (10 mM Tris−HCl, pH 8.2). Purified *NNN*-oligo pools were then digested for 1 h at 37 °C using 1 U/μL of BbsI-HF and rCutSmart™ buffer, then purified again using AMPure XP with a 2:1 bead-to-sample ratio and eluted in 30 μL of nuclease-free water. Purified *NNNNNNN* library samples were then prepared for nanopore sequencing following the details in "Nanopore sample preparation and data acquisition".

For validation library preparation, ligated validation oligo pool reactions were purified using AMPure XP with a 3:1 bead-to-sample ratio and eluted in 30 μL of elution buffer (10 mM Tris−HCl, pH 8.2), then combined to a final concentration of 0.2 μM/pool before enzymatic digestion for 1 h at 37 °C using 1 U/μL of BbsI-HF and 1X rCutSmart™ buffer. Validation library samples were then prepared for nanopore sequencing following the details in "Nanopore sample preparation and data acquisition."

For the 12-letter DNA preparation (S<sup>c</sup>uper-12 and S<sup>n</sup>uper-12), ligated oligo reactions were first purified using Zymo DNA Clean and Concentrator and eluted in 30 μL of elution buffer (10 mM Tris-HCl, pH 8.2). Each ligated oligo set was then combined at a final equimolar concentration of 0.05 or 0.075 μM/oligo before proceeding to a Golden Gate ligation with the addition of 1 U/μL of BbsI-HF, 20 U/μL of T4 DNA ligase, 1X rCutSmart™ buffer, and 1X T4 DNA Ligase Reaction Buffer (Supplementary Fig. 30a). The Golden Gate ligation included 60 cycles of (1) 37 °C for 5 min, (2) 16 °C for 5 min, finalized by a step at 37 °C for 10 min, and a heat inactivation step at 65 °C for 20 min. Following the Golden Gate ligation, the reaction was further digested to remove incomplete ligation products by the addition of 0.45 U/μL of BbsI-HF, 0.45 U/μL of thermolabile Exo I, 2.27 U/μL of Exo III, and 0.23 U/μL of Exo VIII (truncated), incubating at 37 °C for 1 h, followed by a heat inactivation step at 70 °C for 20 min. This reaction was then purified using AMPure XP with a 1.8:1 bead-to-sample ratio and eluted in 30 μL of nuclease-free water. The hairpin on either end of the complete, desired product was removed by splitting the reaction in half and adding 1X rCutsmart™ and 2.78 U/μL of either ScaI-HF or EcoRV-HF. These reactions were incubated at 37 °C for 1 h, followed by a heat inactivation step at 80 °C for 20 min. The split samples were then recombined and prepared for nanopore sequencing following the details in "Nanopore sample preparation and data acquisition."

### Nanopore sample preparation and data acquisition

Nanopore sample preparation followed standard Flongle or MinION Genomic DNA by Ligation protocol (available on the ONT community) using the SQK-LSK110 preparation kit with the following modifications. During the DNA repair and end prep step, the NEBNext FFPE Repair Mix was omitted to avoid potential XNA removal by repair enzymes. The volume of the repair mix was replaced by nuclease-free water. To preserve short fragments, AMPure XP bead-to-sample ratio was increased to 2:1 for the *NNNNNNN* library, and 3:1 for the validation. For the validation library, the first AMPure purification step was omitted to avoid sample loss. Both the Flongle and MinION flow cells used in this work were from the R9.4.1 series. Flow cells were used once per sample, without washing, and collected between 0.15 and 1M reads (Flongle) or 1–10M reads (MinION). A summary of nanopore sequencing runs is shown in Supplementary Tables 14 and 15. Depending on available pores, data collection was allowed to proceed between 24 and 48 h. The collected raw nanopore reads are then passed to the data preprocessing pipeline for basecalling and signal-to-sequence mapping.

### Statistics and reproducibility

Statistics utilized to analyze this data are described in the following methods sections. Analysis can be reproduced with datasets that are deposited to the SRA (see Supplementary Table 31 and "Data availability" statement) and the code developed and utilized in this work (see "Code availability" statement). Read filtering is specified in the text; as a general guideline, all read with a *q*-score < 9 and signal match score >3 were filtered out in this work. Sample sizes for analyses that used subsets of data are presented with figure legends. Model-building dataset sizes are described in Supplementary Table 15. The first 1 million reads from the deposited in PZ_XPCR data (see Supplementary Table 31) were used to generate Supplementary Fig. 29. No statistical method was used to predetermine sample size (all data that passed the filter threshold were used in individual analyses). The experiments were not randomized. The Investigators were not blinded to allocation during experiments and outcome assessment.

### Raw nanopore data preprocessing and signal-to-sequence mapping

Signal-to-sequence mapping uses the Tombo ([https://github.com/nanoporetech/tombo](https://github.com/nanoporetech/tombo), ONT) pipeline. First, raw multi-FAST5 files are split into single FAST5 using the ont-fast5-api ([https://github.com/nanoporetech/ont_fast5_api](https://github.com/nanoporetech/ont_fast5_api), ONT) command `multi_to_single_fast5`. Single FAST5 files are then basecalled using guppy (version 6.1.5 + 446c355, ONT) with the high accuracy configuration settings (dna_r9.4.1_450bps_hac.cfg). FASTQ basecalls passing default guppy quality score settings are assigned to their corresponding single FAST5 files using Tombo command `tombo preprocess annotate_raw_with_fastqs`. For signal-to-sequence mapping, Tombo requires a reference FASTA file that contains ground-truth sequences. The reference FASTA file was generated programmatically by considering every possible combination of ligation products including mismatch homo-ligation (e.g., `P1-A + P1-A`, see Supplementary Table 12), blunt-end ligations leading to a gap (e.g., `P1-P2`, `P1-P1`, `P2-P2`), or pyrophosphorolysis ligation products. Full reference alignment files are deposited in the SRA (Supplementary Table 31). For sequences containing an XNA, the ground truth XNA (B, S<sup>n</sup>, S<sup>c</sup>, P, Z, J, V, X<sup>t</sup>, K<sup>n</sup>) base needs to be substituted for a canonical base (A, T, G, C) for processing in a FASTA format. When processing data for model building, XNAs in reference sequences were substituted for the canonical bases that minimized observed variance in kmer levels;

determined empirically (B → A; Sⁿ → A; Sᶜ → A; P → G; Z → C; X → A; K → G; J → C; V → G). Substituted bases are in general agreement with observations from basecalling XNA-containing reads with guppy (Supplementary Figs. 19 and 21). Signal-to-sequence mapping then proceeds using `tombo resquiggle`. The `tombo resquiggle` command uses mappy (minimap2 version 2.22-r1101 with ONT configuration) to first assign each single FAST5 read to a reference FASTA sequence based on the given FASTQ basecall. Following sequence assignment, Tombo uses dynamic programming for signal segmentation and proceeds to perform per-read signal normalization. As a general comment on the limitations of segmentation-based basecalling, Tombo is sensitive to the reference canonical base chosen for the signal assignment. The per-read, median normalized level signal for each base is then extracted using the `tombo resquiggle` results through the Tombo Python API. Details regarding how Tombo performs mapping, matching, and normalization, along with the Tombo Python API usage, can be found in the Tombo documentation (https://nanoporetech.github.io/tombo/). The resulting preprocessed and normalized signal-extracted data is exported to a CSV file for downstream processing (Supplementary Tables 17, 18). The entire data preprocessing steps, including command groups and parameter settings, are wrapped into a single command (`xenomorph preprocess`) and available on the Xenomorph repository.

### XNA kmer model parameterization

*NNNNNNN* libraries for a given XNA base pair are prepared as previously described in "*NNNNNNN* library, validation library, and 12-letter DNA preparation by XNA tailing and XNA ligation" and sequenced on a Flongle (r9.4.1) flow cell. Signal-to-sequence mapping is then performed using the previously described pipeline in "Raw nanopore data preprocessing and signal-to-sequence mapping" with the following specifications. Reads that do not fully map with full coverage of triplet-barcodes and pool-barcodes of the XNA position are filtered out. Likewise, reads with a q-score <9 and signal match score >3 are not used in our model building. Signal-to-sequence mapping is also carried out with blunt-end ligation products (i.e., *NNNNNN*, or no XNA insertion), such that sequences that map better to blunt-end ligation products are not used. Though ligation reactions were designed to minimize blunt-end ligation product formation, this additional filtering helps further reduce blunt-end ligation products. Similarly, pyrophosphorolysis products are also included in the null alignment, and reads that map better to these products are removed from the analysis.

Kmers of length 4 nt ($k = 4$) were chosen as the basis for the XNA kmer model. The 4-nt kmer was chosen in this work as a proof of concept since reasonable kmer coverage could be obtained for the full NNNNNNN library (512 kmers per XNA base pair insertion) in a single Flongle flow cell run. Compared to using a larger kmer model (e.g., 5-nt or 6-nt) or machine learning, 4-nt kmer models have orders of magnitude lower data requirements making this model size both attainable and desirable. Larger kmer models are possible and generally result in higher accuracy. Each kmer consists of four nucleotide bases centered around the 0th position nucleotide, as exemplified in Supplementary Table 16. Therefore, each heptamer sequence (*NNNNNNN*) is composed of four, 4-nt kmers (i.e., +2 pos *NNN**N***, +1 pos *NN**N**N*, 0 pos *N**N**NN*, −1 pos ***N**NNN*). Observed kmer levels are modeled as normal distributions parameterized with a mean ($\mu_k$) and standard deviation ($\sigma_k$). These parameters are used to describe observed kmer signal-level probability density functions:

$$P(x) = \frac{1}{\sigma_k \sqrt{2\pi}} e^{-\frac{(l_k - \mu_k)}{2\sigma_k}}$$

$P(x)$ = probability that observed kmer signal $x$ came from kmer '$k$'

$\mu_k$ — normalized kmer level mean for kmer '$k$'

$\sigma_k$ — standard levels for kmer '$k$'

$l_k$ — observed median normalized kmer level

In choosing appropriate $\mu_k$ estimates, similar performance in alternative hypothesis testing was found using the mean of observed levels, median of observed levels, or observed mean levels from kernel density estimate (KDE). All parameter measurements are provided in Supplementary Data 1 and are available on the Xenomorph repository.

For kernel density estimates, level model means were approximated using the following kmer-specific bandwidth selection:

$$A = 0.9 * \text{argmin}\left(\frac{\text{IQR}}{1.34}, \sigma_k\right)$$

$$\text{BW} = A * n_k^{-\frac{1}{5}}$$

$A$ — Silverman's rule of thumb $A$ factor

IQR — Interquartile range of median normalized kmer levels for kmer '$k$'

$\sigma_k$ — standard deviation of median normalized kmer levels for kmer '$k$'

$n_k$ — number of observations (measurements) of kmer '$k$'

BW — bandwidth used for kernel density estimate

For practical purposes detailed in the Tombo documentation (https://github.com/nanoporetech/tombo), we set a global standard deviation taken as the average observed standard deviation across all kmers in the model (i.e., $\sigma_k = \sigma \approx 0.4$ for all $k$). Generally, we find that a global model $\sigma$ outperforms kmer-specific choices for $\sigma$ in the kmer probability density function. In the deployed code for single xenonucleotide detection, option to use a global $\sigma$, kmer-specific $\sigma$, or manually set $\sigma$ is available to users.

Tabulated kmer model values alongside coverage, mean, min, max, median, and standard deviation of observed levels determined from this work can be found in Supplementary Data 1. These values can be used to test alternative models that could differ in performance based on application or desired metric (e.g., recall vs. specificity). Custom models can also be measured and linked. Documentation for model building and code used to generate kmer models can be found in the Xenomorph repository (https://github.com/xenobiolab/xenomorph). For quality control, the entire experimental and computational procedure, from building libraries to generating 4-nt kmer models, was performed in duplicate. Models were built from data collected in a single run. The specific nanopore runs used to build models are found in Supplementary Table 15. Raw FAST5 reads for reproducing model building, testing model building replicates, or experimenting with alternative models can be found in the SRA under Bioproject PRJNA932328.

## Alternative hypothesis testing of canonical vs. XNA kmer models

Alternative hypothesis testing is used as the basis for xenonucleotide detection and can either be performed at the per-read or per-sequence level. Though the results shown in the main body of the text are from per-read alternative hypothesis testing, both options are available for experimentation with the deployed code. Per-read testing uses the signal observed from a single read, while per-sequence testing averages the signal across all observations that map to the same sequence. For each heptamer sequence (NNN**N**NNN) a set of mapping kmer sequences (NNN**N**, NN**N**N, N**N**NN, **N**NNN) and observed signal levels ($I_{NNN\underline{N}}$, $I_{NN\underline{N}N}$, $I_{N\underline{N}NN}$, $I_{\underline{N}NNN}$) ($I_{NNNX}$, $I_{NNXN}$, $I_{NXNN}$, $I_{XNNN}$) are extracted. See Supplementary Table 16 for additional information on the numbering nomenclature of kmer sequences within a heptamer region. The kmer probability density function, described previously in "XNA kmer model parameterization," is used to estimate the probability that each observed level (e.g., $I_{ATPG}$) came from the corresponding kmer (e.g., ATPG) or an alternative kmer with a substituted base (e.g., AT$\underline{G}$G). Individual log probabilities are added to calculate the log-likelihood that the observed signal level observations came from a given sequence (e.g., AAT$\underline{P}$GCC). Log likelihoods of XNA-containing or canonical-only sequences can then be used for hypothesis testing based on a log-likelihood ratio (LLR) or outlier-robust log-likelihood ratio (ORLLR). By default, the basecalling for alternative hypothesis testing uses agnostic maximum likelihood criteria for rejecting the null hypothesis: if LLR or ORLLR > 0, then the XNA base is more likely than the proposed alternative. All alternative hypothesis testing of XNA models in this work uses ORLLR rather than LLR as the main test statistic. Code for alternative hypothesis testing is available in the Xenomorph repository using the `xenomorph morph` command and choice of LLR or ORLLR for test statistic can be specified by users. Additionally, the likelihood ratio threshold is an adjustable parameter that can be used to improve case-specific performance.

## Log-likelihood ratio (LLR) calculations

LLR statistics can be used to test if observed signal levels better match kmers containing a specified XNA or kmers containing an alternative base. For every kmer in a heptamer sequence, the LLR ratio is calculated. The sum of the LLRs over each kmer is then taken as the LLR of the entire heptamer. LLR ratio > 0 is used as the default criteria for deciding if the XNA model is more likely than an alternative model for a given observed sequence of signals.

$$\text{LLR} = \log_{10} P(I_k | \mu_{ki}) - \log_{10} P(I_k | \mu_{kj})$$

$I_k$ = observed signal level for kmer '$k$' in heptamer sequence

$P(I_k | \mu_{ki})$ = probability observed signal level $I_k$ belongs to kmer '$i$'

$P(I_k | \mu_{kj})$ = probability observed signal level $I_k$ belongs to kmer '$j$'

LLR = log likelihood ratio

## Outlier robust log-likelihood ratio (ORLLR)

ORLLR is a modified LLR test statistic that is nominally more robust towards outliers. The ORLLR scaling parameters were fixed for all analysis and set as the default used by Tombo (Sf = 4; $Sf_2$ = 3; Sp = 0.3). Additional information on the usage of ORLLR for alternative hypothesis testing can be found in the Tombo documentation (https://github.com/nanoporetech/tombo). ORLLR ratio > 0 indicates the specified XNA model is more likely than the alternative DNA model for

a given observed sequence of signals. The ORLLR test statistic is defined as follows:

$$\text{Sc}_{\text{diff}} = I_k - \frac{\mu_{ki} + \mu_{kj}}{2}$$

$$\mu_{ki-kj} = \left| \mu_{ki} - \mu_{kj} \right|$$

$$\text{ORLLR} = \frac{e^{-\left(\frac{\text{Sc}_{\text{diff}}^2}{\text{Sf}\sigma^2}\right)} \cdot \text{LLR}}{\sigma^2 \cdot \mu_{ki-kj}^{\text{Sp}} \cdot \text{Sf}_2}$$

$I_k$ = observed normalized signal level for kmer '$k$' in heptamer sequence

$\mu_{ki}$ = median normalized kmer level for kmer '$k = i$'

$\mu_{kj}$ = median normalized kmer level for kmer '$k = j$'

$\text{Sc}_{\text{diff}}$ = scale difference

$\sigma$ = global standard deviation of median normalized kmer levels

ORLLR = outlier robust log−likelihood ratio

Sf, $\text{Sf}_2$, Sp = ORLLR scaling parameters

## Recall and specificity calculations

Alternative hypothesis testing is used to refine reads and generate a per-read assignment for deciding if a given heptamer sequence contains an XNA (i.e., NNN**N**NNN) or an alternative base (i.e., NNN**Y**NNN; $Y \neq N$). As metrics to describe how well our 4-nt XNA kmer models perform at identifying XNA bases correctly, two statistics were calculated: recall and specificity. Recall and specificity are calculated either at the per-read level ($n = 1$) or at the consensus level with a specified minimum number of reads mapping to a heptamer required (e.g., $n \geq 10$). Consensus recall and specificity perform sequence-level assignments in calculations (rather than per-read level). Specificity of kmer models was calculated by alternative hypothesis testing on sequences that did not contain any XNAs. The definition of each statistic is provided below:

$$\text{recall} = \frac{\text{TP}}{\text{TP} + \text{FN}}$$

TP = True positive

FN = False negative

$$\text{specificity} = 1 - \text{FDR} = 1 - \frac{\text{FP}}{\text{FP} + \text{TN}}$$

FP = False positive

TN = True negative

FDR = False discovery rate

## Receiver operating characteristic

Receiver operating characteristic (ROC) curves were generated using the `roc_curve` function from the scikit-learn python library. The log-likelihood ratios obtained from basecall outputs were used in the function as target scores and used to compute the recall (or true positive rate) and false discovery rate (FDR) at different classification thresholds. The area under the curve (AUC) was calculated by the `auc` function from the scikit-learn python library.

## Proof-of-concept model validation in a new sequence context

Kmer models in this work were built from an *NNNNNNN* heptamer sequence embedded within a largely fixed sequence context. As a proof of concept that this model can be applied to sequences outside of those found in the *NNNNNNN* library, we enzymatically synthesized a smaller validation library for each XNA base pair, each of which contained 100 unique sequences (Fig. 4, Supplementary Tables 3, 6, Supplementary Data 2). Validation sequences contained XNA bases flanked by 20 randomly chosen canonical bases. Recall on the validation set was calculated at the per-read and consensus level as described previously in "Recall and specificity calculations."

## PCR amplification and basecalling of P ≡ Z template DNA

Two complementary oligos containing P and Z (PCR_Template_P, PCR_Template_Z, Supplementary Table 22) were synthesized by Firebird Biomolecular Sciences (Alachua, FL) and hybridized in a 1:1 molar ratio. 25 ng of this hybridized PZ DNA construct was used as the template for a PCR reaction. PCR reactions contained 0.2 µM of each forward and reverse primer (PCR_Amp_F, PCR_Amp_R1-4, Supplementary Table 22), 5 U/µL of Taq polymerase in 1X ThermoPol® buffer (pH 8.0). Triphosphate concentrations for dxNTPs and dNTPs varied by condition (no dxNTP, limiting, equimolar, optimal) and are tabulated in Supplementary Fig. 29. The PCR reaction then proceeded with thermocycler conditions tabulated in Supplementary Table 23. PCR reactions were purified using Zymo DNA Clean and Concentrator and eluted in 30 µL of nuclease-free water. All PCR products, as well as the template synthetic sequence, were pooled equivalently by mass and prepared for nanopore sequencing following the details in "Nanopore sample preparation and data acquisition." Reactions were sequenced on a MinION 9.4.1 flow cell as part of a larger multiplex run (1M total reads in the run, with a subset belonging to this work). All mapped reads were used for analysis (between 2000–3000 reads mapped to each barcode). PZ basecalling was performed per-read, using methods outlined in this work with Outlier-Robust LLR test with P to G and Z to C kmer comparisons. In the absence of dxNTPs, the P ≡ Z base pair is observed to mutate to a G ≡ C base pair.

## Simulation of genetic codes

Simulated reads were used to test theoretical recall of nanopore-based sequencing for various genetic codes using the 4-nt kmer models described in this work. To simulate reads, every possible heptamer sequence of *NNNNNNN* (*N* = A, T, G, or C; *N* = A, T, G, C, B, $S^n$, $S^c$, P, Z, $X^t$, $K^n$, J, V) was generated. Heptamer sequences were then split into their corresponding kmer sequences (i.e., *NNNN*, *NNNN*, *NNNN*, *NNNN*). For each kmer, 1000 signal levels were simulated by random sampling from the corresponding kmer probability density function using 4-nt kmer model means ($\mu_k$) and a fixed model standard deviation ($\sigma_g = 0.4$). Simulated kmer signals were then recompiled to their sequences to form simulated sequence-signal pairs. A total of 4,096,000 reads (4096 × 1000) were simulated for each substitution base (*N*) in a given genetic code. Alternative hypothesis testing was performed on simulated reads using the Xenomorph pipeline and recall was calculated for every *NNNNNNN* sequence. Code used in this publication to simulate reads is available in the Xenomorph repository under `xenosim.py`.

## The Xenomorph XNA sequencing pipeline

One of the goals of this work was to build a publicly available end-to-end pipeline for routine validation of XNA incorporation in target sequences. As a proof of concept, we created a tool in python called "Xenomorph" comprised of a pipeline consisting of two steps: (1) preprocessing—`xenomorph preprocess` and (2) alternative hypothesis testing—`xenomorph morph`. For preprocessing using `xenomorph preprocess`, Xenomorph runs raw FASTA5 data through the preprocessing pipeline with an additional FASTA handling modification that allows users to input reference sequences with XNA base pairs. Outputs for preprocessing steps are provided in a .csv file (see Supplementary Table 17 for header description), which is used as an input for `xenomorph morph`. For alternative hypothesis testing with the `xenomorph morph` command, Xenomorph uses the XNA base pairs found in the reference sequence to perform LLR or ORLLR testing against user-defined alternatives. For example, for a sequence containing A, T, G, C, B, $S^n$ base pairs, users can calculate the most likely base at the XNA position against the most similar canonical base (e.g., B vs. A), purines/pyrimidines (e.g., B vs. A, G), canonical bases (e.g., B vs. A, T, G, C), or all bases (e.g., B vs. A, T, G, C, $S^n$). Alternative hypothesis testing can be performed on a per-read basis or a global basis. XNA kmers models generated in this work are built-in and can be viewed using `xenomorph models`. Model compilation is performed ad hoc, allowing users to experiment with kmer models. Outputs for alternative hypothesis testing are provided as a .csv file (see Supplementary Table 18 for header description). Users can experimentally generate their own kmer models for arbitrary base pairs and integrate them into the Xenomorph tool by linking model.csv files to available model selections in `models/config.csv`. Since this work only considers single-insertions of an XNA base, kmer models are inherently independent (i.e., signal observations of *NNNBNNN* are independent of *NNNSNNN* observations) and therefore modular. Xenomorph was built to be flexible, allowing users to add more kmer models or modify them, and simple, requiring only two commands to go from raw nanopore data to XNA-refined sequences. A graphical overview of the preprocessing pipeline can be found in Supplementary Fig. 27. Xenomorph can be found in the Xenomorph repository (https://github.com/xenobiolab/xenomorph) alongside all code, documentation, and parameters used in this work. Experimental data for model building and basecalling can be downloaded from the SRA Bioproject PRJNA932328. Additional overview of how the Xenomorph pipeline performs XNA basecalling is found in Supplementary Note 1.

## Reporting summary

Further information on research design is available in the Nature Portfolio Reporting Summary linked to this article.

# Data availability

Models measured in this work used for basecalling are provided in Supplementary Data 1, and can also be found on the Xenomorph github repository (https://github.com/xenobiolab/xenomorph/tree/main/models). The raw nanopore sequences (FAST5) and guppy basecalls (FASTQ) used in this work to build models, validate models, and test 12-letter DNA sequencing have been deposited in the sequence reads archive (SRA) under Bioproject PRJNA932328 and can be accessed without restriction (Supplementary Table 31). Full sequences for hairpin libraries purchased for this work can be found in Supplementary Data 2. Source data are provided with this paper. All other data described in this work are available in the main text, supplied in the supplementary materials, or can be reproduced using deposited datasets and github code. Source data are provided with this paper.

## Code availability

Code for end-to-end processing of nanopore reads and basecalling xenonucleotides described in this work is available without restriction on the Xenomorph github repository (https://github.com/xenobiolab/xenomorph) and is also available on Zenodo under the https://doi.org/10.5281/zenodo.8356450[46].

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

## Acknowledgements

This work was supported by National Institutes of Health Grant R01GM128186 (S.H., M.-J.K., M.-S.K., S.A.B.); National Science Foundation Grant MCB-1939086 (S.H., M.-J.K., M.-S.K., S.A.B.); National Human Genome Research Institute Technology Development Coordinating Center Opportunity Fund U24HG011735 (C.A.T., S.H., J.H.G., S.A.B., A.H.L.); and National Institutes of Health Grant R01HG005115 (J.M.C.). N.K. acknowledges the support of a National Science Foundation Graduate Research Fellowship.

## Author contributions

Project conceptualization was performed by J.A.M., H.K. Methodology for this work was developed by J.A.M., H.K., C.A.T., J.M.C., A.H.L., J.H.G. Investigation was performed by J.A.M., H.K., L.M., M.-J.K., M.-S.K., S.H. Visualization of data and results was performed by J.A.M., N.K., H.K., C.A.T., J.M.C., A.H.L. Funding for this work was acquired by J.A.M., A.H.L., J.H.G., S.A.B. This project was administered and supervised by J.A.M. Writing of the original draft was carried out by J.A.M., H.K. Reviewing and editing of the manuscript was performed by J.A.M., H.K., C.A.T., J.M.C., J.H.G., A.H.L., S.H., S.A.B.

## Competing interests

S.A.B. owns Firebird Biomolecular Sciences, which makes various hachimoji reagents available for sale. M.-J.K., M.-S.K. and S.H. are affiliated with Firebird Biomolecular Sciences. The remaining authors declare no competing interests.
