## [Peer Review File · Nature Communications]

REVIEWER COMMENTS

Reviewer #1 (Remarks to the Author):

This manuscript by Hinako Kawabe described a new synthetic strategy, enzyme based, to synthesize expanded alphabetic letters into DNA sequences. This enzymatic strategy used a side reaction of particular DNA polymerases to incorporate a single nucleotide overhang on a blunt end DNA. This overhang DNA can be an XNA nucleotide. With this overhang product, the authors further ligated the overhang DNA with another overhang DNA and in principle this strategy could be further repetitively used to extend the sequence to be synthesized so that any combination of DNA sequences containing the expanded alphabetic letters DNA can be synthesized. The authors used these synthetic strands as libraries to build the sequencing base caller algorithm, with which, any DNA containing the expanded DNA letters can be sequenced by nanopore.

The maintext of the manuscript is easily understood however the maintext also didn't provide quantitative and useful information critical to support the core value of the manuscript. Then the SI becomes extremely difficult to understand. I have also found a lot of issues for this manuscript, prohibiting it from publishing at Nature Communications, at least for now. For the ease of communication, I am listing my major concerns prior to any editorial decision for this manuscript.

1. This reference (J. Am. Chem. Soc. 2023, 145, 15, 8560–8568) also from the same group is extremely relevant to the results described in this manuscript, which might damage the novelty of this study. In principle, the mentioned JACS paper reported nanopore sequencing of the hachimoji nucleotides (ZPSB). Thus, though the authors reported that they are sequencing a 12 letter DNA system in this manuscript, only 4 new letters (XKJV) were newly sequenced. However, the difference is that all the hachimoji nucleotides (ZPSB) in the JACS paper were all organic synthesized but the XNAs in this paper were all enzymatic synthesized. But this difference is just minor, if viewed from the innovation of nanopore sequencing. So my concern is, with the JACS paper already published, whether the materials demonstrated in this manuscript has the sufficient innovation to be suitable for Nature Communications anymore, is becoming quite embarrassed. I would leave the decision to the editorial team. There is a very big chance that the editors didn't know that JACS reference because the authors didn't mention that till very late part of the discussion (ref 34). However, technically, due to such strong relevance to this whole manuscript, the authors should mention it in the introduction clearly.

2. According to this manuscript and the principle of nanopore sequencing, a K-mer, presumably a 4-mer sequence determines a single step nanopore readout. Even for DNA, which contains 4 letters, the raw sequencing accuracy is not very high. For DNA, the needed combinations are 4 to the 4=256. However, for this system, the needed resolution would be, 4 to the 12=16777216. I am actually quite shocked by how big this number is, however it seems that these states can all be resolved and discriminated by this paper? How does the expanded DNA letters interfere with reading of other DNA alphabets? So my

conclusion is that the core problem, the resolution of the nanopore sequencer has not really be solved, not to mention to include new DNA alphabets. I agree that the included XNA can be recognized by the nanopore sequencer. I am however not convinced that it would not significantly damage the overall accuracy considering that a 16777216 now needs to be dealt with?

3. If I understand it correctly, the mentioned enzymatic synthesis technique is extremely lowly efficient and has to be carried out manually? Then how big the K-mer library is? The authors mentioned that they used Chip-DNA synthesis as part of the synthesis? But the only the XNA incorporation part is manually carried out? This part is very important, describe with more technical details. I am just a bit shocked about how big this library should be and how efficiently it was prepared? If the authors only built an extremely small library which fails to cover all sequence combinations of all 12 possible DNA alphabets, then the authors are overstating their results, which will make the manuscript not suitable for publication, at least at Nature Communications.

4. The readers of Nature Communications are of extremely broad expertise background, so the authors are encouraged to use easily understood way of writing. It is hard to understand terms such as NNN-R, NNN-BCR in Table S1C without any explanation of what they are. The generation of the K-mer library is the key of this paper. This has to be well described.

5. Fig 5, confusion matrix? No quantitative values? The basecall accuracy are only color coded but no quantitative values given? Issues like this should be solved as the readers may be interested in the exact numbers.

6. The authors only demonstrated synthesis of a single letter of XNA but no demonstration of any consecutive combinations of multiple XNAs? If that is the case, then what happens if my target DNA sequence do contain consecutive XNAs?

Eventually, it is a manuscript containing a lot of workload, which I sincerely respect. I am generally interested to see that nanopore can deal with an expanded alphabet letter. However, it seems that the manuscript only attempted the possibility of nanopore sequencing of these XNAs but has not solved the problem at all, at least from the view of nanopore. Also considering that there is already a nanopore paper that deals with hachimoji nucleotides (ZPSB) and is from a similar group of authors, I am also concerned about the absolute innovation it demonstrates to be qualified for a Nature Communications paper.

Reviewer #2 (Remarks to the Author):

This nicely written manuscript describes development of a 12-letter DNA code. In comparison to previous studies on the same subject, the present study stands out by its ingenious approach to enzymatic synthesis of DNA containing unnatural nucleotides. In addition to writing DNA strands

containing unnatural nucleotides, the authors show that they can read the nucleotide sequence of such strands by means of commercial nanopore sequencing.

Overall, this is a solid work that greatly advances the broader field of synthetic biology.

Addressing the following will improve the manuscript.

The claims regarding novelty, in the abstract and on page 11 (line 252) are a bit overblow. The work published last year (Nano Letter, 2022 which is Ref 37 in the manuscript) has already introduced the concept of embedding unnatural nucleotides within a DNA strand and discerning their presence by means of nanopore sequencing. The key aspect differentiating the present study is the method of producing such DNA, which is enzymatic here and chemical in the 2022 study. The authors are asked to properly introduce that study in the introduction.

Although the manuscript's writing is of high quality, the writing is dense, in particular in some parts of the Results. As Nature Communications is rather forgiving regarding the length of the manuscript, please consider providing more details in the main text. This is particularly true for the last section of Results.

Also for the last section, please specify the actual length of the DNA constructs containing all 12 letters.

It would be nice if the authors could discuss how they envision further development of the base calling method to identify k-mers containing more than one synthetic nucleotide.

On the same note, please add a comment stressing the fact that the present approach to base calling requires to have single synthetic nucleotides flanked by natural bases.

Reviewer #3 (Remarks to the Author):

Dear Professor Marchand,

I found your paper very interesting and it seems to make a substantial contribution to the field. I have written a much more extensive report but it is hard to transfer the formatting here, so please see the attached pdf file. It contains a long list of comments that you may consider if given the opportunity to revise your manuscript.

Please find below a point-by-point response to comments raised. Documents with tracked changes to main manuscript and supplementary information are also provided. The following text formatting was used in this response to help improve readability:

- *Original reviewer comment = blue italics*
- **Authors response = black**
- **Edited manuscript text excerpts = green**

Reviewer #1 (Remarks to the Author):

This manuscript by Hinako Kawabe described a new synthetic strategy, enzyme based, to synthesize expanded alphabetic letters into DNA sequences. This enzymatic strategy used a side reaction of particular DNA polymerases to incorporate a single nucleotide overhang on a blunt end DNA. This overhang DNA can be an XNA nucleotide. With this overhang product, the authors further ligated the overhang DNA with another overhang DNA and in principle this strategy could be further repetitively used to extend the sequence to be synthesized so that any combination of DNA sequences containing the expanded alphabetic letters DNA can be synthesized. The authors used these synthetic strands as libraries to build the sequencing base caller algorithm, with which, any DNA containing the expanded DNA letters can be sequenced by nanopore.

The maintext of the manuscript is easily understood however the maintext also didn't provide quantitative and useful information critical to support the core value of the manuscript. Then the SI becomes extremely difficult to understand.

- We understand Reviewer 1's request for additional quantitative information in the main text and in response to this request have made the following changes:
 1. Various portions of the main text were rewritten for improvement of clarity and addition of detail (see manuscript with highlighted changes for more detailed overview).
 2. XNA tailing yield estimates are now included in the main text written portion (previously only SI)
 3. XNA ligation yield estimates are now included in the main text written portion (previously only SI)
 4. Area under the curve ranges for ROC are now included in the main text written portion (previously, area under the curve was listed as a legend for figure).
 5. Raw values confusion matrices in Figure 5 are now included in SI. Note - leaving these out was an error on our end, as we had already included other raw values for confusion matrices in the SI.

I have also found a lot of issues for this manuscript, prohibiting it from publishing at Nature Communications, at least for now. For the ease of communication, I am listing my major concerns prior to any editorial decision for this manuscript.

This reference (J. Am. Chem. Soc. 2023, 145, 15, 8560–8568) also from the same group is extremely relevant to the results described in this manuscript, which might damage the novelty of this study. In principle, the mentioned JACS paper reported nanopore sequencing of the hachimoji nucleotides (ZPSB). Thus, though the authors reported that they are sequencing a 12 letter DNA system in this manuscript, only 4 new letters (XKJV) were newly sequenced. However, the difference is that all the hachimoji nucleotides (ZPSB) in the JACS paper were all organic synthesized but the XNAs in this paper were all enzymatic synthesized. But this difference is just minor, if viewed from the innovation of nanopore sequencing. So my concern is, with the JACS paper already published, whether the materials demonstrated in this manuscript has the sufficient innovation to be suitable for Nature Communications anymore, is becoming quite embarrassed. I would leave the decision to the editorial team. There is a very big chance that the editors didn't know that JACS reference because the authors didn't mention that till very late part of the discussion (ref 34). However, technically, due to such strong relevance to this whole manuscript, the authors should mention it in the introduction clearly.

We completely agree with the reviewer's suggestion and have done a better job explaining the "state-of-the-field". Reference 34 (J. Am. Chem. Soc. 2023, 145, 15, 8560–8568) is now included earlier in the work. To accommodate this change, we have rewritten a portion of the introduction to properly introduce the state-of-the-field to the reader, including what has been previously done and what challenges remain. The new introduction text is included at the end of the response to this question.

We respectfully disagree with the conclusion that the only difference between the JACS 2023 paper and our work is the addition of four bases and that our method used enzymatic synthesis instead of chemical synthesis. JACS 2023 paper described sequencing of an 8-letter alphabet (ours has 12 letters, 5 nucleotides not examined in JACS 2023) made by chemical synthesis (ours is made enzymatically, which required development of new methods) with a custom nanopore system available only in the developing lab (we describe results with a commercial system available to everyone). The JACS 2023 paper did not build sequencing models (ours built 4-nt kmer models) and therefore does not constitute a sequencing solution. We agree with the referee that this is an editorial decision, but this summary makes, in our view, a compelling case for a favorable editorial decision, especially considering the interests of the readers.

The similarities the reviewer notices between this work and JACS 2023 only exist at face value: much like Illumina sequencing, PacBio sequencing, and Sanger sequencing all use fluorescence measurements as a readout, this work and JACS 2023 paper use an ion current readout. However, at both a fundamental and applied level – this work is different, novel, and we would argue more impactful. We also want to highlight that other than a subset of co-authors being shared, there is no overlap in research efforts, materials, data, or methods.

In more detail, we highlighted what makes this work distinct, unique, and innovative compared to other work published in JACS 2023.

Sequencing Devices:

- Authors of JACS 2023 (co-authors in this work) use experimental nanopores on research-built devices (think of these as self-manufactured, manually built devices). These devices use a single pore protein (MspA pore protein) with expensive electronic equipment (\$20k amplifiers) to take single-molecule

measurements. Only a handful of research groups in the world can perform these types of experiments. These researcher-built devices collect few (10s-100s) reads per run. Reads data is collected in series, not in parallel, and logged manually. Each device has a single pore, meaning only one strand of DNA is being measured at a given time. In summary, JACS 2023 nanopores perform single-molecule and real-time sequencing but they are not multiplexable, high throughput, or commercially available.

- This work uses commercial nanopore devices (Oxford Nanopore Sequencing) that use different pore and motor proteins (an engineered CsgG variant, and an unknown Hel308 variant), the devices are inexpensive (\$1,000), and openly accessible by any researcher with few barriers. The devices also allow multiplexed data collection (100,000-10,000,000 reads per run). The major goal of this research was to make translatable technology that can be easily adopted (notably, next-generation sequencing applications). In summary, this work uses devices that perform single-molecule, real-time, multiplexable, and high throughput sequencing.

Data acquisition:

- JACS 2023 uses variable voltage data collection. This is a technique pioneered by JACS 2023 authors and not applicable to commercial nanopore sequencing devices.
- This work collects data under constant voltage, as is routinely used for commercial nanopore sequencing. Therefore, the results from this work can be translated and applied by anyone who uses commercial nanopores.

Type of research:

- JACS 2023 work could best be classified as fundamental science and basic research (notably, in biophysics, analytical chemistry).
- This work can best be described as applied or translational research. There is a heavy emphasis on usability, access, and portability for the efforts produced.

Basecalling models:

- Work in JACS 2023 does not build basecalling models. The classifications they built are not general, and only empirically describe sequences that they had measured.
- This work leverages high throughput capacity of commercial devices to build the first “ground-up” predictive kmer models for sequencing XNAs. Our models are published which allows others to sequence these XNAs. Additionally, with the raw data we deposited others in the community could try to build and experiment with their own models .

Number of reads measured:

- Work in JACS 2023 took measurements on the order of low 10^3 reads (N=868 reads for sequencing, N = 2,343 total in work).
- This work measured 10^6 - 10^7 reads in a highly parallelized, multiplex format. This work contained 1000x more data collected than JACS 2023 work.

Sequence diversity measured:

- Work in JACS 2023 measured 22 unique sequences, with only 8 being used for sequencing portion
- This work measured XNAs in every NNNXNNN context (4,086 unique sequences per XNA, with 36,774 unique XNA-containing sequences for model building). These are highly diverse sequences. This work then measured hundreds of validation sequences, and sequences that contain 12 letters. Again, here this work operates at orders of magnitude higher amounts of data collection and sequence space than JACS 2023.

Bases measured:

- Work in JACS 2023 focused on studying an 8-letter alphabet (ATGCBS:PZ)

- This work used enzymatic techniques to overcome synthesis challenges, making 12-letter sequencing possible for the first time. The focus was on an individual letter basis (9 additional letters). Models produced allow for mixing and matching for basecalling all 12.

Data accession:

- Raw data in this work has been deposited in the SRA and will be freely accessible as data that other researchers can use for improving/training basecallers. This is the first such dataset available to the public.

Basecalling tools and XNA models on github:

- Since this data was collected on commercial devices, the models we built are useful for anyone performing ONT nanopore sequencing. We integrated these models into the first basecalling tools built to sequence these models.

NEW TEXT ADDED TO INTRODUCTION

Currently, research and development in the field of XNA face fundamental barriers to entry in the form of sequencing, which generally requires highly specialized equipment and analytical expertise. One possible solution is to adapt existing first-, second-, or third-generation DNA sequencing technology to work with more DNA letters. However, modern sequencing infrastructure is inherently inflexible and highly specialized for ATGC sequencing. Adapting fluorescence-based DNA sequencing techniques for XNA sequencing, such as Illumina sequencing, would require a plethora of innovations including new reagents (e.g., XNA nucleotides with unique fluorophores), engineered polymerases capable of replicating XNAs, modification of instrumentation to handle more cycles, and creation of new data collection/analysis pipelines. Any fluorescence-based XNA next-generation sequencing strategy is not realistically attainable at the present. As an alternative approach, other next-generation sequencing methods built for DNA might be more amenable to serving as XNA sequencing solutions.

Nanopore sequencing has the ability to sequence non-canonical bases such as epigenetic and epitranscriptomic modifications.²⁵⁻²⁷ In recently published work, co-authors assessed the practicality of nanopore sequencing for 8-letter hachimoji DNA (A, T, G, C, B, S^c, P, Z) using the Hel308 motor protein with an MspA pore.^{8,28} This work critically established that third-generation (high throughput, multiplexable, single molecule, real-time) sequencing of supernumerary DNA is theoretically possible despite the “k-mer explosion” in possible current signals induced by an expanded DNA alphabet. As a limitation, this previous work did not attempt to build models for decoding the nanopore current signals to nucleic acid sequences. In addition, it was performed on a non-commercial research platform consisting of a single nanopore run by a technician (low throughput, non-multiplexable). While useful for the development of sequencing technology, this previous nanopore setup cannot be easily adopted by those in other fields.

Beyond these efforts on non-commercial devices, others have approached classification of non-standard bases using commercial nanopores (GridION, ONT).²⁹ Importantly this previous work showed that commercial nanopore sequencing platforms are indeed capable of sequencing chemically modified nucleobases including 2,4-diamino-purine, 5-nitro-indole, and 5-octadiynyldeoxyuracil. However, orthogonal base pairing nucleotides were never tested and only a small sequence space was explored, both of which exclude their applicability to expanded and evolvable genetic alphabets. For commercial nanopore sequencing to be applicable to 4⁺-letter genetic alphabet systems that contain orthogonal XNA base pairs, bespoke nanopore sequencing models will be required.

According to this manuscript and the principle of nanopore sequencing, a K-mer, presumably a 4-mer sequence determines a single step nanopore readout. Even for DNA, which contains 4 letters, the raw sequencing accuracy is not very high. For DNA, the needed combinations are 4 to the 4=256. However, for this system, the needed resolution would be, 4 to the 12=16777216. I am actually quite shocked by how big this number is, however it seems that these states can all be resolved and discriminated by this paper? How does the expanded DNA letters interfere with reading of other DNA alphabets? So my conclusion is that the core problem, the resolution of the nanopore sequencer has not really be solved, not to mention to include new DNA alphabets. I agree that the included XNA can be recognized by the nanopore sequencer. I am however not convinced that it would not significantly damage the overall accuracy considering that a 16777216 now needs to be dealt with?

We thank the reviewer for their consideration regarding the kmer space that must be explored disagree with their conclusion. The full combinatorial *de novo* sequencing space is large but not as large (12 bases, 4 positions, is $12^4 = 20,736$ possible kmers). Accessing the full combinatorial 12^4 kmer sequencing space is not possible at this time but this is due to synthesis limitations. For sequencing, we easily measured model parameters of 2,304 kmers which is only $1/10^{\text{th}}$ of what would be required for 12^4 space. This work addresses an exhaustive subset of this possible sequence space where kmers can have at most a single XNA. We note that this does not mean that your nucleic acids can have more than 1 XNA, rather, it sets an upper bound to how close together two XNAs can be in a given sequence.

To further address this comment, we can highlight here what our achievement is for a bottom-up model. Our model can indeed handle all 12 letters, and multiple instances of the same letter, given they are at least a heptamer apart (1 in 7 bases). The math works out to 2,304 measured kmers for all 9 XNAs, which covers a possible ~36,864 heptamer sequences. We cover 10% of the total combinatorial space. We note here that the previous state-of-the-art was not next-generation sequencing. Alternative methods were all low throughput, not single molecule, and could generally not handle more than one XNA per DNA strand.

Moving forward, how we achieve 12^4 is an open challenge and interesting problem. Synthesis roadblocks remain but we believe that our work is the entry point to this next era of improvements, and we welcome the community tackle 12^4 sequencing as a challenge for this decade. Scaling down this problem, we can imagine various 6-letter codes (6^4) being a more trackable problem to solve, though challenges here are also present.

We also want to highlight that a wide range of applications do not require definition of this entire space. For example, receptors and catalysts emerging from laboratory in vitro evolution, including those that deliver drugs to cancer cells, have added nucleotides dispersed only sparsely (1-3), functional additions embedded in the context of standard nucleotides. This is exactly the type of sequence that the data reported here parameterize. Though 12 letters is the upper limit of what this work showed, most applications will have various combinations of 6-letter alphabets. Other areas where full 12-letter space is not required, but impactful, include: (1) characterization of polymerases for PCR amplification of XNA bases, (2) generation of codons and anticodons for genetic code expansion, (3) characterization of XNA base retention in an engineered host, such as in a plasmid.

We hope that this clarification is sufficient to convince the reviewer that the kmer space necessary to build our model is much more reasonable and has incredible utility. Since this clarification is important, we have included additional edits throughout the paper, and major changes in the introduction for further emphasis.

NEW TEXT ADDED IN INTRODUCTION

Here we report progress on both synthesis and sequencing that makes supernumerary DNA sequences containing 6-letter, 8-letter, 10-letter, or 12-letter alphabets easily accessible for the first time. In the area of XNA synthesis, we introduce an enzyme-assisted strategy that can be used to incorporate single orthogonal XNA base pairs (B≡Sⁿ or B≡S^c; P≡Z; X^l≡Kⁿ; and J≡V) into synthetic 4-letter DNA. For XNA sequencing, we put theory to practice and develop commercial nanopore basecalling models capable of sequencing single XNA bases (B, Sⁿ, S^c, P, Z, X^l, Kⁿ, J, and V) embedded in a standard DNA (i.e., A, T, G, C only) context.

If I understand it correctly, the mentioned enzymatic synthesis technique is extremely lowly efficient and has to be carried out manually? Then how big the K-mer library is? The authors mentioned that they used Chip-DNA synthesis as part of the synthesis? But the only the XNA incorporation part is manually carried out? This part is very important, describe with more technical details. I am just a bit shocked about how big this library should be and how efficiently it was prepared? If the authors only built an extremely small library which fails to cover all sequence combinations of all 12 possible DNA alphabets, then the authors are overstating their results, which will make the manuscript not suitable for publication, at least at Nature Communications.

We thank the reviewer for taking the time to consider these points. We have addressed these points below:

- A total of 36,864 sequences can be constructed from kmers of $k = 4$, where only a single XNA is allowed (2,304 possible kmers).
- Single XNA incorporation steps can be combined with other traditional DNA manipulation methods to increase XNA sequence complexity. In this work, we used Golden Gate ligation to make two such examples of 12-letter DNA (shown in Fig. 5).
- In this work, the XNA incorporation is carried out manually. We presented these results through the story of how we developed these enzymatic reactions that can integrate with existing phosphoramidite synthesis infrastructure. For the sequencing work, the incorporation is carried out manually on libraries of DNA and this is how we access large sequence space.
- That said, the reactions themselves *do not need* to be carried out manually. Many avenues exist for integrating these reactions at commercial scales. Reaction steps are technically automatable and can be adapted to similar pipelines companies might use for semi-synthesis. One example would be performing these steps on an automatable platform with DNA on a bead.
- For clarification, we do not claim we synthesized a full 12^4 space. It remains a possibility some variation of this method can be used to make sequences that approach that complexity, but this would need to be performed at scale. This work remains a very clear advancement over the previous state-of-the-art in both sequencing (no NGS alternative) and synthesis (low barrier entry).
- Libraries synthesized with Chip-DNA synthesis (IDT Oligo pools) are made in a combinatorial fashion, discussed in main text and tabulated in the SI. Addressed later, we are including Supplementary Data 2 which contains sequences for all oligo pools used in this work.
- We do use Chip-DNA (IDT oligo pools) synthesis as part of the library generation; the Chip-DNA synthesis provides us with diverse DNA sequence contexts, and we insert the XNA bases enzymatically. In order to clarify this point, we have moved Supplementary Figure 15a to main text Figure 3a, and removed Figure 3b as this data were already shown in Supplementary Figure 20.

The adjusted main figure is also shown here, along with an edited caption to clarify that we utilized a combination of Chip-DNA synthesis and enzymatic synthesis to build our libraries.

- Finally, while we did not sequence combinations of all 12 possible DNA alphabets (12^4), we respectfully disagree with the reviewer that we are overstating our results regarding this specific claim. We specify throughout our work that this is a model for sequencing single xenonucleotide substitutions in DNA. Since we agree with reviewer that this distinction is critically important, we have included various edits throughout the paper to re-emphasize our efforts.
- We hope that these explanations provide clarity and address the reviewer's questions in a satisfactory manner.

EDIT TO FIGURE 3 IN MAIN TEXT:

Fig. 3. Generation of 12-letter (ATGCBSPZXXKJV) nanopore sequencing kmer models. (a) Overview of construction of NNNNNNN libraries, starting from two synthetic oligo pools (NNN-Pool) that contain blunt, NNN-3' ends. The 24-nt triplet-barcodes in these hairpins are linked to the 3'-NNN sequence, allowing for proper identification of bases adjacent to XNA inserts. Complementary XNA base pairs are added to the library hairpins using x XNA tailing and XNA ligation. The 8-nt pool-barcode is used to identify which XNA was tailed to the 3'-end. Final libraries contain an XNA base insert in every possible NNN x NNN context (N = A, T, G, C; $64 \times 64 = 4,096$ unique sequences per XNA base). (b) 4-nt kmer models

were generated by decomposing every sequenced heptamer (NNNNNNN; N = modified nucleotide) into its corresponding 4-nt kmers. **(c)** All measured normalized current signal means (μ_k , 2,304 total values from kernel density estimate) for each 4-nt kmer, with positive values in deeper purple and negative values in deeper orange. Heatmaps are binned by kmer position containing the xenonucleobase (-1, 0, +1, or +2). 'N' is denoted in the x-axis and the remaining NNN is denoted by row, sorted alphabetically (AAA to TTT). **(d)** Example traces overlaying observed mean signal (orange) with expected signals produced by either the XNA model (blue) or a model for standard DNA (gray) kmer model. For the standard DNA model, the most similar standard base chosen for each XNA was determined from empirical observation (**Supplementary Fig. 15 and 17**). Number of reads used: B ($n = 18$); Sⁿ ($n = 24$); S^c ($n = 40$); P ($n = 32$); Z ($n = 28$); X^t ($n = 18$); Kⁿ ($n = 12$); J ($n = 14$); and V ($n = 18$). Error bars indicate standard deviation of observed normalized signal level.

The readers of Nature Communications are of extremely broad expertise background, so the authors are encouraged to use easily understood way of writing. It is hard to understand terms such as NNN-R, NNN-BCR in Table S1C without any explanation of what they are. The generation of the K-mer library is the key of this paper. This has to be well described.

We agree with reviewer here and have added additional material to improve clarity. Among major additions, we included new supplementary data that contains all sequences for libraries used in this work (**Supplementary Data 2**). These sequences are formatted in to be ready-to-order from IDT or other synthesis companies. This supplementary data file also contains explicit columns that mark the barcode sequences in each hairpin. We hope this will resolve reader confusion and provide an easy avenue for others in the field to replicate or appropriate these sequences for their own.

Additionally, we have modified the current SI captions to add clarity regarding the designs. Importantly, we have changed how we refer to the NNN-barcode (now calling it the triplet-barcode). We more explicitly mention how this 24-nt triplet-barcode lets us decode the NNN region adjacent to each XNA.

Fig 5, confusion matrix? No quantitative values? The basecall accuracy are only color coded but no quantitative values given? Issues like this should be solved as the readers may be interested in the exact numbers.

We thank the reviewer spotting this as leaving this table out was an oversight on our end. The quantitative values for this figure have now been included in **Supplementary Table 10** and referenced in the figure caption.

The authors only demonstrated synthesis of a single letter of XNA but no demonstration of any consecutive combinations of multiple XNAs? If that is the case, then what happens if my target DNA sequence do contain consecutive XNAs?

We thank the reviewer for this comment since they bring up a great point, and addressing this comment with experiments adds value to the work. The major focus of this work was single XNA incorporation for building kmer models for sequencing. In Fig. 2h we suggest this reaction can be cycled. We had previously

tested reaction cycling but decided not to include it in the work since it takes the story in a different direction. Instead, we had offered the readers a general strategy for performing this reaction.

We now include our experiments showing how this reaction would work. Currently there are upper limits for how many times you can successfully cycle without product amplification (e.g., with PCR) or starting with more material (scale up).

We will be adding reaction development for cycling to add two consecutive XNAs (two PZ base pairs). This has been included **Supplementary Figure 14**. This figure is now referenced in the main text.

Briefly: We conducted an experiment in which we cycled XNA tailing and XNA ligation along with MlyI cutting as shown in **Figure 2h**. MlyI is an uncommon class of Type IIS restriction enzyme that unlike other Type IIS enzymes, leaves a blunt end after digestion. This means that we can have one hairpin of DNA function as an acceptor, and one hairpin function as a donor. The donor hairpin has a Type IIS restriction enzyme cut site 5 nt away from the XNA tail. To regenerate starting material, we can digest with MlyI after XNA ligation.

NEW METHOD ADDED TO SI:

Consecutive insertion of XNA base pairs using MlyI Type IIS restriction enzyme. 5'-phosphorylated hairpin oligos were purchased from IDT (5'Phos-11HP, 5'Phos-15HP, and 5'Phos-ScaI-HP; **Supplementary Table 1b**). 5'-Phos-15HP contains an MlyI restriction site adjacent to site of XNA ligation. MlyI is a Type IIS restriction enzyme (5'-GAGTCNNNNN↓-3') that leaves a blunt end after cutting. 5'Phos-15HP (donor hairpin with MlyI site; abbreviated HP_D) and 5'Phos-11HP (acceptor hairpin; abbreviated HP_A) were tailed with P and Z, respectively, generating HP_D-P and HP_A-Z. These two hairpins were then ligated and subsequently treated with exonuclease following the optimized conditions described in this work. This material was purified using Zymo's DNA Clean and Concentrator and eluted in 30 µL of elution buffer. The purified construct contains a single P Z base pair insertion and was digested using 1.24 U/µL of MlyI and 1X rCutSmart™ buffer at 37 °C for 2 h then heat inactivated at 65 °C for 20 min. MlyI digestion results in a hairpin with a terminal P Z, which also possesses the required termini for another tailing reaction (5'-PO₄ and 3'-OH blunt end). In the second round of cycling, this hairpin (which already contained a P Z base pair) was subjected to Z-tailing generating HP_A-ZZ. A third hairpin (5'Phos-ScaI-HP, lacks MlyI site, abbreviated HP_P) was P-tailed to generate HP_P-P. Hairpins were then ligated, and incomplete ligation products were removed by adding 1 U/µL of MlyI, 7.7 U/µL of Exo III, 1.5 U/µL of thermolabile ExoI, and 0.77 U/µL units of ExoVIII (truncated) and incubating at 37 °C for 1 h, followed by a heat inactivation step at 72 °C for 20 min. Products were analyzed on a 2% (w/v) agarose gel, stained with GelGreen, and visualized using a blue light transilluminator.

NEW FIGURE ADDED TO SI:

Fig. S14. Proof of concept for XNA tailing and XNA ligation cycling to insert two consecutive P Z base pairs. (a) Agarose gel showing key steps in consecutive XNA insertion. Each lane is described in the schematics that follow. (b) A hairpin containing an MlyI restriction site adjacent to

the site of XNA ligation is used (donor hairpin, HP_D). MlyI is a Type IIS restriction enzyme (5'-**GAGTCNNNNN**↓-3') that leaves a blunt end after cutting. A donor hairpin with an MlyI site and an acceptor hairpin were tailed with P and Z respectively (generating HP_D-P, HP_A-Z), ligated and treated with exonucleases following the optimized conditions described in this work, and then purified (**lane 1**). The purified construct contains a single P Z base pair insertion. Product from lane 1 was digested using MlyI, resulting in products observed in **lane 2**: (major product) blunt end hairpin products HP_D (regenerated donor hairpin) and HP_A-Z (acceptor hairpin with a 3'- P Z base pair); (minor product) undigested product from lane 1. **(c)** Separately, a donor hairpin without an MlyI site was prepared by XNA tailing (HP_P-P). XNA ligation followed by MlyI and exonuclease treatment does not result in formation of a ligation product (**lane 3**). **(d)** In a second round, reaction product mixture from lane 2 were tailed with Z to produce Z-tailed donor hairpin (HP_D-Z) and Z-tailed PZ-acceptor hairpin (HP_A-ZZ). XNA ligation followed by MlyI and exonuclease treatment does not result in formation of a ligation product (**lane 4**). Minor ligation product previously observed in lane 2 is also no longer observed, suggesting this additional MlyI + exonuclease digestion round effectively removes MlyI-containing products. **(e)** Tailed donor and acceptor hairpins (HP_D-Z, HP_A-ZZ, HP_P-P) were ligated. XNA ligation followed by MlyI and exonuclease treatment results in formation of a product with two consecutive P Z base pair insertions (**lane 5, ρ**). Incorrect ligation product from two donor hairpins (HP_D-Z + HP_P-P) would not be present since HP_D-Z contains an MlyI site. **(f)** MlyI cycling control reactions were carried out to assay how yield is generally affected by multiple ligation cycles. Blunt-end HP_D was ligated using T4 DNA ligase (blunt-end ligation), treated with exonuclease, and purified (**lane 6**). Reaction product was then digested with MlyI (**lane 7**), then subjected to an additional round of blunt-end using T4 DNA ligase (**lane 8**). Consecutive rounds of MlyI digestion and ligation result in a visible decrease in blunt end ligation product yield. For additional details regarding hairpin sequences used, please see methods section "Consecutive insertion of XNA base pairs using MlyI Type IIS restriction enzyme". Nucleic acid end abbreviation: 3' indicates 3'-OH, 5P'- indicates 5'-PO₄.

Eventually, it is a manuscript containing a lot of workload, which I sincerely respect. I am generally interested to see that nanopore can deal with an expanded alphabet letter. However, it seems that the manuscript only attempted the possibility of nanopore sequencing of these XNAs but has not solved the problem at all, at least from the view of nanopore. Also considering that there is already a nanopore paper that deals with hachimoji nucleotides (ZPSB) and is from a similar group of authors, I am also concerned about the absolute innovation it demonstrates to be qualified for a Nature Communications paper.

We thank the reviewer for their evaluation of our work. Portions of this comment were addressed elsewhere in our reply, but we will highlight our responses here to the major points of concern:

- This work does perform nanopore sequencing on millions of reads that contain XNA bases. The model benchmark is summarized in the ROC curves in the main text.
- This work does cover *bona fide* development of commercial nanopore sequencing models for single insertion of BSPZXXKJV in an ATGC context, and therefore allows for multiple XNAs inserted at once (as exemplified with the 12-letter sequences made) given sufficient spacing.
- The models we built are bottom-up and made from fully empirical measurements. These models fully cover any NNNXNNN sequence (for X = XNA, N = ATGC).
- Reviewer is correct that this does not mean we can sequence XXXXXXXX (X = ATGC BSPZXXKJV). We tried to make this distinction clear and have added additional language to clarify this. Sequencing XXXXXXXX is currently impossible, but please do note that our work represents a major leap forward for XNA next-generation sequencing.
- We published the end-to-end pipelines for going for basecalling with our models on github.
- Putting this development in context of Oxford Nanopore sequencing, the kmer measurements (and the kmer-based approach) are most similar to early Nanopore basecallers for ATGC. In nanopore sequencing development, kmer models preceded contemporary machine learning methods, and in fact, kmer models are inputs to long-short term memory model training.
- We provide example use cases how these methods enable new areas of investigation in responses to other reviewers. For example, this method enables using nanopore sequencing for studying how DNA polymerases replicate these bases in PCR reactions in a high-throughput, multiplexed fashion. This is a major improvement over the state-of-the-art, which involved using agarose or SDS-PAGE gel assays.
- The JACS 2023 work (which shares a subset of co-authors, but led by different groups) is performed on non-commercial devices, different detectors, a different electrochemical setup, different pore proteins and motor proteins, and does not build sequencing models. An analogy here is comparing Illumina sequencing (high-throughput, multiplexable) to Sanger sequencing (low throughput, non-multiplexable). Both methods use fluorescence as a readout to sequence nucleic acids, but have many other associated differences that make their use case almost orthogonal.
- To this point, I will add that we built methods and models that can be readily adopted. In both synthesis and sequencing, we focus on working with commercially available materials/devices. In addition to the work covered in the written manuscript, we also publish the models, raw datasets, end-to-end sequencing pipelines so that anyone can enter this field. Any group that buys a \$1000 commercial nanopore sequencing device can replicate our work without additional expertise or equipment. This critical distinction is important for us to mention.
- Taking the sequencing story aside, we also want to highlight that there is more than one innovation being presented in this work. The story on enzymatic synthesis, while it might seem simple at face value, is also completely novel. The enzymatic synthesis decreases the barrier to entry for anyone to add XNA base pairs to their DNA. Previously, the only way to access these sequences

was >\$100/nt if purchased from IDT or Firebird Biotechnologies. This enzymatic synthesis strategy makes J:V and X:K base pairs easily accessible for the first time. Previously, there was no strategy to obtain these even from commercial services that offer phosphoramidite synthesis.

- Lastly - please do not conflate a sharing of authors with a lack of originality or innovation. Coauthors of this work came together to share knowledge, expertise, and enthusiasm for this area of research. From the author contributions statements, you will notice this work was performed completely independently, and led by different groups. There is no overlap in methods, materials, or equipment between these two sets of work. This project was started independently and led by the corresponding author and his research group (none of which were involved in JACS 2023 work). Given the scale of challenges required to modernize this field, it is natural that more than one group will be contributing to these efforts.

Reviewer #2 (Remarks to the Author):

This nicely written manuscript describes development of a 12-letter DNA code. In comparison to previous studies on the same subject, the present study stands out by its ingenious approach to enzymatic synthesis of DNA containing unnatural nucleotides. In addition to writing DNA strands containing unnatural nucleotides, the authors show that they can read the nucleotide sequence of such strands by means of commercial nanopore sequencing.

Overall, this is a solid work that greatly advances the broader field of synthetic biology. Addressing the following will improve the manuscript.

We thank Reviewer #2 for their kind comments, and hope that our response addresses their comment to a high degree of satisfaction.

The claims regarding novelty, in the abstract and on page 11 (line 252) are a bit overblow. The work published last year (Nano Letter, 2022 which is Ref 37 in the manuscript) has already introduced the concept of embedding unnatural nucleotides within a DNA strand and discerning their presence by means of nanopore sequencing. The key aspect differentiating the present study is the method of producing such DNA, which is enzymatic here and chemical in the 2022 study. The authors are asked to properly introduce that study in the introduction.

We thank the reviewer for highlighting the novelty of our enzymatic work. We have reworked our introduction to more readily summarize the state-of-the-field, and how this work represents a major leap towards XNA sequencing of unnatural base pairs with commercial devices.

In case it is warranted, we also wanted respond to the reviewer with comments regarding differences in case there is doubt regarding differences/novelty with our work. Both Nano Letter 2022 and this work use nanopore sequencing on modified nucleotides, but there are important distinctions that make this work of a completely different in scope and present a different type of achievement.

One important distinction is that unnatural bases can form orthogonal Watson-Crick base pairs following different hydrogen bonding patterns. Therefore, in our system, the information on a DNA molecule can be retained and amplified – at least theoretically (DNA polymerases that can amplify all 12-letters have not been found currently). Conversely, unnatural bases that lack orthogonality will be lost upon amplification.

We have modified the introduction to state the novelty of our work in the context of Tabatabaei et al., as well as Thomas et al. per suggestion of Reviewer #1's comments. The specific comments that address the relevant state-of-the-art are shown below for convenience.

NEW TEXT ADDED IN INTRODUCTION

Nanopore sequencing has the ability to sequence non-canonical bases such as epigenetic and epitranscriptomic modifications.²⁵⁻²⁷ In recently published work, co-authors assessed the practicality of nanopore sequencing for 8-letter hachimoji DNA (A, T, G, C, B, S^c, P, Z) using the Hel308 motor protein with an MspA pore.^{8,28} This work critically established that third-generation (high throughput, multiplexable, single molecule, real-time) sequencing of supernumerary DNA is theoretically possible despite the “k-mer explosion” in possible current signals induced by an expanded DNA alphabet. As a limitation, this previous work did not attempt to build models for decoding the nanopore current signals to

nucleic acid sequences. In addition, it was performed on a non-commercial research platform consisting of a single nanopore run by a technician (low throughput, non-multiplexable). While useful for the development of sequencing technology, this previous nanopore setup cannot be easily adopted by those in other fields.

Beyond these efforts on non-commercial devices, others have approached classification of non-standard bases using commercial nanopores (GridION, ONT).²⁹ Importantly this previous work showed that commercial nanopore sequencing platforms are indeed capable of sequencing chemically modified nucleobases including 2,4-diamino-purine, 5-nitro-indole, and 5-octadiynyldeoxyuracil. However, orthogonal base pairing nucleotides were never tested and only a small sequence space was explored, both of which exclude their applicability to expanded and evolvable genetic alphabets. For commercial nanopore sequencing to be applicable to 4⁺-letter genetic alphabet systems that contain orthogonal XNA base pairs, bespoke nanopore sequencing models will be required.

Although the manuscript's writing is of high quality, the writing is dense, in particular in some parts of the Results. As Nature Communications is rather forgiving regarding the length of the manuscript, please consider providing more details in the main text. This is particularly true for the last section of Results.

This is a great point, and one that we will take to heart. We have made various changes to manuscript to improve clarity and readability, including results section. Major excerpts of places where text was added are covered below (Note: this is a non-exhaustive list of edits made for readability).

EDITED TEXT – Improving explanation and motivation behind LCMS assays for tailing activity

[...] To reach the full extent of the 12-letter alphabet, we chemically synthesized the 2'-deoxy-xenonucleoside triphosphates of the remaining bases: dXⁿTP, dKⁿTP, dJTP, dVTP (see **Schemes 1-4 in Supplementary Information**). Next, we developed a sensitive liquid chromatography/mass spectrometry (UPLC/QTOF) assay for detecting tailing activity. In this assay, a DNA polymerase and 3'→5' exonuclease are simultaneously used to perform N+1 tailing and N+1 removal of a dxNTP substrate on an exo-resistant, blunt-end DNA hairpin (**Fig. 2a**). The net reaction results in formation of dxNMP + PP_i, and requires the presence of a 3'-OH blunt-end DNA, exonuclease, DNA polymerase, and dxNTP. From this UPLC/QTOF assay, it was evident that both KF (exo-) and Terminator polymerase were fully capable of non-templated N+1 addition of all four standard dNTPs and all nine dxNTPs tested, including both the N-nucleoside (Sⁿ) and C-nucleoside (S^c) of S. (**Fig. 2b, c, Supplementary Fig. 3 and 4**).

EDITED TEXT – Improving explanation of design choice for model building

With this in mind, we set out to build and measure diverse DNA-XNA libraries that could be used to construct de novo ground-up models for sequencing single xenonucleotides within a natural DNA context. Here, we took inspiration from the early predictive 'kmer models' for nanopore sequencing. In these models, the current signal produced by any given DNA sequence is only a function of the sequence kmer, which consists of the incident nucleotide in the pore and its surrounding nucleotide context.⁴⁴⁻⁴⁶ Sequencing models built with longer kmer sequences will benchmark with a higher overall accuracy. There is, however, a diminishing return in accuracy improvements as kmer size increases which is balanced against the exponential increase in library complexity and data collection requirements with

longer kmers. Balancing performance and complexity, we decided to measure the signal produced by every 4-nt kmer that contains a single xenonucleobase from our set. The synthetic capabilities of XNA tailing and XNA ligation made it possible to generate libraries containing all 4-nt kmers with a single xenonucleotide pair ($4^4 = 256$ kmers per xenonucleotide). To cover the entirety of the 4-nt-long kmer sequence space, we designed a dual-barcoded DNA hairpin library that could be synthesized on a chip (NNN-Pool; **Fig. 3a, Supplementary Fig. 16, Supplementary Table 1c-e**). To establish the ground truth for each read, full factorial NNN coverage at the blunt end was linked to a unique 24-nt barcode (Triplet-barcode), while the identity of the tailed xenonucleotide was linked to a unique 8-nt barcode (Pool-barcode). These barcode sequences are distal to the site of XNA ligation and can therefore be decoded through standard ATGC basecalling. Though ligation biases could make it difficult to acquire reads of certain combinations (NNNNNNN; N = modified nucleotide, N = A, T, G, C), only a subset of total sequence space would be required to obtain full coverage of all required 4-nt kmers.

EDITED TEXT – Improving description of 12-letter sequencing results and highlighting why this work is considered important

[...] In the construction procedure, exonucleases are added to remove intermediary DNA products generating the desired 244 bp 12-letter dsDNA product. In this proof-of-concept example, basecalling was performed two different ways: 1) by comparing the XNA base at a position against a model that contains all 12 possible nucleobases, and 2) by comparing the XNA base at a position against a model that contains only the XNA and the most similar standard nucleobase. Even when all 12 letters are present in the model, the basecalling model was able to properly decode XNAs in S^cuper-12 with 39-89% per-read recall (**Fig. 5, Supplementary Table 10**). For the Sⁿuper-12 sequence, all but one XNA were properly decoded in the 12-letter model, with the exception being Kⁿ (per-read recall of 14%). When only performing most similar standard base comparisons, all XNAs in Sⁿuper-12 were properly recalled (6793% per-read recall). Indeed, given the complexity of possible current signals when 12-letter models are invoked for basecalling, one should expect the chosen 12-letter-containing sequence to have a large influence on recall (**Supplementary Fig. 27 and 28, Supplementary Table 11**). Despite only being a proof of concept, this foray into 12-letter DNA space represents an important milestone, marking the first time that DNA containing 6 orthogonal base pairs has been synthesized and sequenced.

Also for the last section, please specify the actual length of the DNA constructs containing all 12 letters.

We have added the length (244 bp) in the results section.

It would be nice if the authors could discuss how they envision further development of the base calling method to identify k-mers containing more than one synthetic nucleotide.

This is a two part response: one directly addressing the reviewer and the second with changes we made to the manuscript.

First: Since this is a topic we routinely think about, we thank the reviewer for encouraging us to expand on this point. Our work focused on single XNA bp insertion, but we did allude to how this method could be used to insert multiple base pairs (**Fig. 2h**). We have carried out additional experiments showing how one can cycle XNA ligation/tailing reactions (data added to **SI Supplementary Figure 14**). Practical limitations here, of course, are related to final yields.

Regarding the theory of how larger models could be built: For any 6-letter set (ATGC + XY, where XY = BS, PZ, XK, or JV) sequence space required for building models can be reduced to a mask that describes spacing between standard bases and XNAs. Using approaches described in our work, bounding regions are synthesized for free (NNN, phosphoramidites), while any interior region (between two XNA insertions) needs to be synthesized from enzymatic coupling steps. Therefore (and roughly speaking), through combinatorial synthesis (pooled sets with either X or Y being added) and with XNA tailing/ligation cycling, we can achieve full models for any 6-letter code with the following strategy.

Model mask	Training	# XNA bp pos	# Cycles	~ Pools needed
X	NNNXNNN	1	1	1 (this work)
XX	NNNXXNNN	2	2	2
XNX	NNNXNXNNN	2	3	2
XNNX	NNNXNNXNNN	2	3-4	2
XXX	NNNXXXNNN	3	3	4
XNXX	NNNXNXXNN	3	3-4	4
XXXX	NNNXXXXNNN	4	3-4	8

Number of pools required is given by the number of XNA bases included (2^{n-1}).

Number of ligation/tailing cycles is determined by the number of bases between XNAs in a sequence (inclusive)

Since we are building dsDNA, we always also capture the reverse complement sequence. (XNNN gives us NNNY). Depending on how the cycles are performed (parallelized or in series) you could do more (or less) reactions. What you will notice is that if we wanted to include doublets, we would need to build 3 additional libraries to cover XX, XNX, and XNNX combinations in 4-nt kmer models. This would also require 3-4 rounds of cycling.

We think this is possible, and one direction the XNA sequencing field should consider taking. Beyond using our method, it is entirely plausible that some day companies like IDT or Twist would offer pooled synthesis of nucleic acids that contain these bases. We are hoping the demand and interest will be there from the scientific community in this decade, which will hopefully attract industry to follow.

Second: To address this point with the manuscript, we have included the following text in the discussion section:

EDITED TEXT: Modified discussion

[...]As these methods improve and adoption widens, strategies for synthesis and sequencing of higher complexity nucleic acids will also become possible. For example, variations of XNA tailing and XNA ligation that would allow one to incorporate multiple consecutive XNA bases, such as the example of MlyI cycling (**Supplementary Fig. 14**), would create an opportunity for 4-nt kmer models XNA with multiple xenonucleobases present in close proximity.

On the same note, please add a comment stressing the fact that the present approach to base calling requires to have single synthetic nucleotides flanked by natural bases.

Thank you for bringing this to our attention. We have made a various changes that specific highlight this as the achievement of this work.

At the end of the introduction, we have added the sentence for clarity.

EDITED TEXT: Introduction

[...] For XNA sequencing, we put theory to practice and develop commercial nanopore basecalling models capable of sequencing single XNA bases (B, Sⁿ, S^c, P, Z, X^l, Kⁿ, J, and V) embedded in a standard DNA (i.e., A, T, G, C only) context.

In the discussion, we also added a sentence to emphasize the fact that currently, we are sequencing an XNA flanked by natural bases. The discussion now opens with:

EDITED TEXT: Discussion

[...]We have demonstrated a general strategy for incorporating up to four additional orthogonal base pairs into standard DNA, and used these methods to build the first openly accessible XNA models for sequencing XNAs (B, Sⁿ, Sc, P, Z, X^l, Kⁿ, J, V) in a standard DNA context (A, T, G, C) on commercial nanopore devices.

Reviewer #3 (Remarks to the Author):

Dear Professor Marchand,

I found your paper very interesting and it seems to make a substantial contribution to the field. I have written a much more extensive report but it is hard to transfer the formatting here, so please see the attached pdf file. It contains a long list of comments that you may consider if given the opportunity to revise your manuscript.

We thank Reviewer #3 for this comment. We also greatly appreciate the amount of time Reviewer #3 spent on reading everything so thoroughly; this was incredibly helpful, and we hope that the edits and responses that came from these comments are satisfactory.

Summary of Synthesis and Sequencing of 12-letter Supernumerary DNA

Understanding and manipulation of DNA (made using the four standard nucleobases A, T, G, and C) has resulted in significant advances in biotechnology, healthcare, and even information storage. The four standard bases form two Watson-Crick base pairs (A:T and G:C). The two rules governing Watson-Crick base pairing, size complementarity (larger purines pairing with smaller pyrimidines) and hydrogen bonding complementarity (hydrogen bond donors on one base pairing with hydrogen bond acceptors on the complementary base), allow up to 4 additional orthogonal base pairs to be formed from at least 8 additional nucleobases (the hydrogen bonding combinations could be satisfied by more than one heterocyclic system, e.g., Sn vs Sc). These additional xenonucleotide bases pairs allow 'supernumerary' DNA codes (with more than four letters) to be constructed. DNA comprising additional nucleobases is structurally and chemically more diverse, with potential benefits in the development of diagnostics, therapeutics, enzyme-like catalytic nucleic acids (XNAzymes), biotechnology (e.g., allowing more codons for encoding noncanonical amino acids), and information storage (3.58 bits per base for a twelve-base code, compared to 2 bits per base for a four-base code). Unfortunately, the tools and infrastructure for synthesizing and sequencing DNA consisting of more than the four standard bases are still under development and are accessible to only a limited number of laboratories with highly specialised equipment and analytical and synthetic expertise. While several nonstandard building blocks are commercially available, generic strategies for incorporation of these residues into DNA is hampered by, for example, the incompatibility of some of the nonstandard bases (e.g., J, Sn, and Xt) with phosphoramidite chemistry. This has limited chemically accessible DNA to eight bases (like the Hachimoji set of the standard four bases plus B, Sc, P, and Z, previously published by some of the authors). Notably, the authors point out that xenonucleic acid sequencing is less advanced (lower throughput, lower sensitivity, and inherently less generalizable) than the four-base “Sanger sequencing” technologies developed almost half a century ago. To facilitate research on xenonucleobases expanded DNA alphabets, the authors set out to reduce the major barriers to entry that impede progress in this field. These two major barriers are the lack of simple and generalisable approaches for the synthesis and sequencing of xenonucleic acids. The authors were inspired by recent progress in enzymatic synthesis of DNA using, for example, terminal deoxynucleotidyl transferase (TdT). Controlled enzymatic DNA synthesis requires the ability to incorporate one nucleotide at a time, for which either polymerase-nucleotide conjugates (Palluk 2018; <https://doi.org/10.1038/nbt.4173>) or 3'-protected dNTPs (e.g., Flamme 2022; <https://doi.org/10.1038/s42004-022-00685-5>) would be required,

none of which are commercially available. The authors therefore employed another strategy for enzymatic DNA synthesis, based on the “blunt-end addition reactions” catalysed by some (mostly 3'-exonuclease deficient) DNA polymerases. This activity, first reported by Clark et al. in 1987, allows polymerases to add an additional ‘untemplated’ nucleotide at the 3' end of a DNA molecule, creating a singlebase overhang. All four of the standard nucleotides (A, T, G, and C) can be transferred, but addition of A is strongly favoured in mixtures of dNTPs. This tailing reaction is the basis for T:A cloning and adapter ligation for NGS library construction. This is therefore a very well-known reaction, but it is still poorly understood. Surprisingly, the authors demonstrated that this technique can be used to tail blunt-ended hairpin DNA substrates with all four of the natural and all nine of their noncanonical dNTPs (surprising because the natural preference for A makes the reaction seem inefficient with other nucleotides). Of course, tailing DNA with a single base is insufficient for studying the properties of novel base pairs, so the authors proceeded to show that two bluntended hairpins, tailed with complementary xenonucleotides, can be ligated by standard DNA ligases to form double-stranded products that contain the xenonucleobase pairs. There is a type IIS restriction endonuclease (MlyI) that leaves a blunt end, so sequential incorporations of the noncanonical bases could be envisioned (although this is not experimentally demonstrated). Furthermore, Golden Gate assembly was used to assemble several of these short duplexes into the first 12-base supernumerary DNA. Blunt-ended DNA hairpins ending in all possible natural three-nucleotide (NNN) sequence combinations ($4 \times 4 \times 4 = 64$ hairpins) were tailed with each of the noncanonical bases, and each tailed hairpin was ligated to its complementary base-extended hairpin. The result is a set of libraries where each xenonucleobase (one per library) is flanked on both sides by all possible NNN combinations (4096 variations), therefore covering all possible sequence contexts (actually, only a small fraction of this library needs to be sequenced to cover all the 4-nt kmers of interest). These libraries were sequenced on a commercially available Oxford Nanopore MinION flow cell, generating data on current flow as each of the noncanonical bases transition through the nanopore, in all possible NNNXNNN contexts (X being the xenonucleobase). The presence of the xenonucleobase affects the signal resulting from the surrounding bases, which would make it hard to train the base caller. To ensure that the NNNs surrounding the nonstandard base are interpreted correctly, they were physically linked to 24-nucleotide barcodes that helped correctly interpret the NNN regions, despite the effect of the X on signal produced by these bases. Another barcode revealed the identity of the tailed dXTP. Both the X and the surrounding NNNs could be accurately read using this approach, enabling accurate 4-nt XNA kmer model training. Despite the higher accuracy of larger kmer models, the authors argue that 4-nt kmer models require orders of magnitude less data, making a 4-nt kmer model both desirable and more attainable. The per-read recall of the the 4-nt kmer model was between 60-87% of XNA nucleotides, which increased to 63-99% by basecalling the consensus of at least ten reads. The authors argue that this brings the sequencing of xenonucleic acids from the zeroth to the third generation. Other members of the scientific community may now use the XNA synthesis and sequencing techniques described here to generate more data that may be used to build more advanced basecalling algorithms. Taken together, the authors describe a collection of approaches that allow XNA to be synthesised and sequenced using standard tools that are available in many laboratories.

Significance of the work

A difficulty in working with xenonucleobases is that they are usually incompatible with standard tools like PCR and sequencing technologies. Therefore, the development of tools to amplify and analyse XNAs is as important as the tools for their synthesis. A major obstacle for XNA sequencing is the need to read through unnatural bases without truncation, and an inherent drawback of sequencing-by-synthesis and sequencing-by-ligation is that new reagents are required for each additional xenonucleobase sequenced (unless the promiscuous behaviour of polymerases is used to read the presence of modifications via misincorporation

“fingerprints” at the modification sites). Nanopore sequencing is therefore very attractive since its mechanism of action in principle allows new bases to be ‘detected’ without depending on modified versions of the nonstandard dNTPs. However, the high error rates of nanopore sequencing means that several reads need to be analysed to generate a consensus. Since extra base pairs are very unlikely to all be copied efficiently, DNA prepared by other means (e.g., PCR) would not have the very well-defined sequence of DNA constructed using the method described in this paper, which in turn would complicate model building and in turn base calling. The authors have essentially found a path around the chicken-egg problem by devising a clever way of synthesising DNA containing xenonucleobases in all possible sequence contexts and then sequencing these libraries using a commercially available nanopore sequencing platform.

I would appreciate some discussion of how the authors’ XNA sequencing workflow could be used to study the properties of e.g., XNA replication (i.e., is the sequencing fidelity high enough to study the fidelity of polymerases with dxTPs, how would such a problem be approached).

This is a great point of inquiry, and we thank the reviewer for mentioning this. In particular, we want to point out that studying XNA replication was one of the use cases we envisioned for this work (and was part of our motivation behind pursuing single molecule sequencing). Is it good enough to resolve differences:

Short answer: Yes

To address the reviewer remark, we conducted additional experiments. As a model for studying amplification differences we revisited work performed in 2011 (Yang et al, *J. Am. Chem. Soc.* **2011**, <https://doi.org/10.1021/ja204910n>) on PCR amplification of PZ containing sequences. In this work, various dNTP/dxNTP concentrations were tested to find those that lead to highest yield. We were able to reproduce this work for one tested polymerase (Taq polymerase) on the P:Z base pair. For P and Z will mutate to G and C (respectively) in the absence of any dPTP/dZTP in the reaction. In basecalling, we used this as a comparison and included a positive control synthetic DNA standard. We included these results in the manuscript.

New experimental data, methods, and tables were added to manuscript and included at the end of this response.

Long answer: Yes, but depends how you design experiments and what magnitude differences you care to resolve.

From experiments (shown below) we can clearly resolve differences between optimized and limiting dxNTP conditions despite the nominal accuracy of kmer models for PZ being 80-90%. The reason this does work is that exponential amplification of PCR will produce exponentially diverging populations of nucleic acids. Differences between two conditions will grow with number of cycles, while the resolution of detection is fixed. With 20-25 cycles we should be able to detect differences as small as 1% between two polymerases or conditions (assuming a base between 90-99% replication fidelity) with our methods. If you choose a sequence that is already easy to resolve by the kmer models (i.e., the signal difference between P and G is far apart, such that there is rarely overlap) then you can improve your resolution. Since we can multiplex samples (as was shown in the new SI figure), we should theoretically be able to test between 1000-10,000 conditions simultaneously on a single MinION flow cell (~500-700\$). This level of throughput is a significant advancement over the previous state of the art. In the Yang 2011 work in

particular, PCR condition optimization was performed using agarose gels as the primary analytical technique.

Assumptions for why we think this is possible for detecting polymerase differences, and limits to consider:

1. Define an XNA replication error rate ($\epsilon > 0$) that is a property of polymerase or condition. This could be polymerase error rate, or error rate due to buffer, divalent cation choice, dxNTP, dNTP concentration, etc.

2. Replication error at XNA position after n cycles: $E_n = (1 - \epsilon)^n$

3. Absolute between two conditions with two error rates: $D(\epsilon_1, \epsilon_2) = |(1 - \epsilon_1)^n - (1 - \epsilon_2)^n|$

4. If $D(\epsilon_1, \epsilon_2) \gg$ measurement error, then we can measure this difference

4. Note that $D(\epsilon_1, \epsilon_2)$ is not monotonically increasing for $\epsilon_1, \epsilon_2 > 0$, and does have a maxima which depends on both on ϵ_1, ϵ_2 .

0. $\lim_{l \rightarrow \infty} D(\epsilon_1, \epsilon_2) = 0$ (i.e., convergence at full information loss)

5. Example plotted: Difference between 95% and 90% XNA replication efficiency conditions as a function of cycle number. This difference would be trivial for us to detect (25% difference in incorporation between 10-15 cycles of amplification).

6. Trivial case: any given polymerase can also be compared against a “perfect polymerase” such that absolute error can also be quantified. The reference here is $\epsilon_2 = 0$, such that $D(\epsilon_1, 0)$ is monotonically increasing.

EDITED TEXT: Main text section added to discuss sequencing for PCR

As an example of how these sequencing models can be applied to accelerate XNA research, we then revisited a landmark experiment on PCR development for P Z base pairs carried out over a decade ago.²² Originally, analysis of successful P Z amplification was carried out using a low throughput agarose gel electrophoresis. Showcasing the leap to the NGS era, in a single multiplex nanopore run we show how the

PZ kmer models enable simultaneous measurement of PCR amplification efficiency for a P Z base pair amplified under various dxNTP and dNTP concentrations (**Supplementary Fig. 25, Supplementary Table 8**). In agreement with this earlier work, our sequencing results show near complete retention of P Z base pair using optimized dxNTP (0.6 mM dPTP; 0.05 mM dZTP) and dNTP (0.1 mM dATP, dGTP, dTPT; 0.6 mM dCTP) concentration, with increasing loss of P Z bases as dxNTPs become limiting. Given the throughput of nanopore flow cells (1-10M reads, MinION flow cell), it should now be possible to use nanopore sequencing to screen PCR replication efficiency across hundreds to thousands of conditions (e.g., polymerase mutants, buffer composition, dxNTP/NTP concentrations) simultaneously.

FIGURE ADDED TO SI

Fig. S25. PCR amplification and sequencing of a DNA template with a P Z base pair. (a) Synthetic template DNA containing a P Z (**Supplementary Table 8a**) base pair was amplified with Taq polymerase in a pH 8.0 buffer with varying concentrations of dxNTP and dNTP. PCR products were sequenced on a MinION nanopore flow cell then basecalled for PZ detection. Read fractions that basecalled to (b) P and (c) Z for each condition are shown. PCR conditions differ only by concentration of dxNTP and dNTPs used. The remaining fraction for each base corresponds to G and C basecalls (the most likely standard mutation for P and Z), respectively. Unamplified, synthetic P Z DNA was sequenced as a positive control (Std) for basecalling.

SUPPLEMENTARY TABLE ADDED TO SI

Table S8. Template sequences, primer sequences, thermocycler settings used for PCR of P Z base pair.

(A) Synthetic oligo template sequence with a P Z base pair (red, bold), purchased from Firebird Biosciences (Alachua, FL). Oligo template sequences are hybridized prior to use as a PCR template. Primer sequences used to amplify template. Each condition used a different barcoded reverse primer (PCR_Amp_R1: Equimolar; PCR_Amp_R2: Optimal; PCR_Amp_R3: No dxNTP; PCR_Amp_R4 Limiting). All conditions used the same forward primer (PCR_Amp_F). Sequences shown in 5' to 3' direction. (B) Thermocycler conditions used to amplify P Z template. 25 total cycles were performed.

A.

Name	Template Sequence
PCR_Template_P	GGTCTGGTGCCACTGGTAACTGGGACAGCTGAAGTFCAGTCAGCCAGGGAAACACGATAGGCAACCACACC
PCR_Template_Z	GGTGTGGTTGCCTATCGTGTTCCTCCCTGGCTGACTGZACTTCAGCTGTCCAGTTACCAGTGGCACCAGACC

Name	Primer Sequence
PCR_Amp_F	CGATTCCACAAAGACACCGACAACCTTCTTGGTCTGGTGCCACTGGT
PCR_Amp_R1	CGATTCAAGGATTCATTCCACGGTAACACGGTGTGGTTGCCTATCGTG
PCR_Amp_R2	CGATTACGTAACCTGGTTTGTTCCTGAAGGTGTGGTTGCCTATCGTG
PCR_Amp_R3	CGATTCAACCAAGACTCGCTGTGCCTAGTTGGTGTGGTTGCCTATCGTG
PCR_Amp_R4	CGATTGAGAGGACAAAGGTTTCAACGCTTGGTGTGGTTGCCTATCGTG

B.

Step	Temperature (°C)	Time
1	95	2 min
2	95	15 s
3	58.5	15 s
4	72	10 s
5	Cycle steps 2-4	24x
6	72	1 min
7	10	Hold

What follows is a more detailed list of comments that may help the authors revise their manuscript if it is accepted for publication.

Detailed comments

Line 36; ‘advancements’ > ‘advances’

- This wording has been corrected.

Line 47; no comma before ‘DNA’

- This comma has been removed.

Line 67; ‘transliteration sequencing’ is not a standard term, please refer more specifically to the technology used in reference 22 (Sanger sequencing).

- We apologize for this oversight; thank you for pointing this out.
- The sentence: “Notably, methods for routine sequencing of xenonucleic acids (XNAs) are decades behind that of DNA and RNA, and rely on low-throughput, non-multiplexed measurements, such as gel-shift assays^{19,20}, mass spectrometry²¹, and transliteration sequencing.²²”
- Has been corrected to: *“Notably, methods for routine sequencing of xenonucleic acids (XNAs) are decades behind that of DNA and RNA, and rely on low-throughput, non-multiplexed measurements, such as gel-shift assays^{19,20}, mass spectrometry²¹, and selective conversion of XNAs to standard bases followed by Sanger sequencing.²²”*

Line 72; comma before ‘deeming’

- This comma has been added.

Line 74; ‘or’ > ‘and’

- This wording has been corrected.

Lines 98-99; ‘deoxy-nucleotide transferase’ > ‘deoxynucleotidyl transferase’

- This wording has been corrected.

Line 99; ‘Tdt’ > ‘TdT’, also note that this is the same as ‘terminal deoxynucleotidyl transferase’, not just an example

- Thank you for pointing this out.
- The phrase: “Enzymes like terminal deoxynucleotidyl transferase (e.g., Tdt)”
- Has been corrected to: “Enzymes like terminal deoxynucleotidyl transferase (TdT)”

Line 100; Please insert a reference to Palluk et al., 2018, De novo DNA synthesis using polymerase-nucleotide conjugates. Nat. Biotechnol., 36 (7), 645-650.

- We apologize for overlooking this reference; thank you for bringing it to our attention. This reference by Palluk et al. has been added to the main text.

Lines 101-102; ‘Tdt’ > ‘TdT’

- This typo has been corrected.

Line 103; ‘specially protected building blocks’ > ‘specially protected building blocks or polymerase-nucleotide conjugates’

- Thank you for pointing this out; the phrase “polymerase-nucleotide conjugates” has been added.

25. Line 110, ref 29; Clark et al reported that tailing with a dNTP may be enhanced in the presence of the complimentary nucleoside monophosphate (e.g., dTTP and dAMP) and I was curious whether the authors considered this as a potential strategy for enhancing tailing by the more difficult dNTPs (e.g., Sc in Table S2A). While the authors managed to tail using all nine nonstandard nucleotides used in this paper, there may be some that are even worse tailing substrates, and some comment on the generalisability of the approach would be welcome.

We thank the reviewer for this thoughtful suggestion to consider Clark et al.’s addition of the monophosphate to increase the tailing activity of the complementary base. This phenomenon is something we have indeed previously thought about attempting, however, we were dissuaded after reading about the dNMP requirements. Specifically, authors of this work used 50 mM of the dNMP in their reactions. Synthetic access to dxNMPs for the bases we work with are extremely limited and not produced currently at a commercial or even pilot scales.

Nonetheless, we decided to give this a try. To understand whether lower concentrations of the nucleoside monophosphate could still lead to enhanced tailing, we tailed dS^cTP and dPTP with their complementary monophosphates present (dBMP and dZMP, respectively). The following experimental setup was used:

- **Reaction time:** Since under optimized conditions we had observed tailing reactions go to near completion (>95% starting material reacted), we choose to reduce reaction time in order to be able to detect a different.
- **Detection:** For detection, we used oligo LCMS method described in the work (UPLC-QTOF) which allowed for simultaneous detection of reacted and unreacted product.
- **dZMP source:** We were able to obtain chemically synthesized dZMP from Firebird but at 10 mM scale. We were able to attempt the dPTP tailing reaction at 3.3 mM dZMP added.
- **dBMP source:** We enzymatically synthesized dBMP from dB and ATP using a nucleoside kinase described in another work:
(Chen et al. Plos One, 2017, <https://doi.org/10.1371/journal.pone.0174163>)

While there was slightly less blunt end (N) product observed with the monophosphate present, we did not see significant differences in the tailed product as estimated by peak area (N+1). From these results, we

decided not to pursue this reaction further. Additionally, while these results are negative, they do not prove that this strategy could work. The dxNMP amount we used was an order of magnitude lower to what was tested by Clark et al, so it remains a possibility. That said, given the required concentrations of dxNMP that would be needed make this strategy impractical (at least for now).

We hope that these additional data, shown below, are helpful in addressing the Reviewer's questions (short answer – obtaining high dxNMP concentration is another barrier to testing this. From what we tested, we did not see obvious differences).

RESPONSE FIGURE (NOT INCLUDED WITH MANUSCRIPT)

Fig. R1. XNA tailing in presence of the complementary deoxynucleoside monophosphate. (a) 24 μ M of hairpin oligo 5'Phos-ScaI-HP was tailed using dPTP and dS^cTP, along with the complementary deoxynucleoside monophosphate (dZMP and dBMP, respectively). dPTP was tailed either with or without the addition of 3.3 mM chemically synthesized dZMP for 4 h, half the optimized time, to avoid completion of the tailing reaction; otherwise, XNA tailing conditions described in this work were used. dS^cTP was tailed either with or without the addition of 3.9 mM of dBMP (amount estimated from ATP consumption by LC-MS), enzymatically synthesized using the deoxynucleoside kinase DmdNK Q81E; otherwise, conditions described in this work were used. Afterwards, 19.8 μ M of oligo was incubated with 1.8 U/ μ L of ScaI-HF at 37 °C for 2 h, followed by subsequent heat inactivation at 80 °C for 20 min. Samples were then prepared for UPLC/MS-QTOF using the methods described in “General procedure for high resolution HPLC/MS analysis of oligonucleotides.” (b) Extracted ion chromatograms (EIC) for dPTP tailing with and without the addition of dZMP showing formation of: (left) starting material (N) and (right) tailed product (N+1). (c) dBMP synthesis using the deoxynucleoside kinase DmdNK Q81E. 13.3 mM of dB and ATP, either with or without the addition of 23.3 μ M DmdNK Q81E, were incubated in 70 mM Tris and 5 mM MgCl₂ buffer (pH = 7.5) at 37 °C for 16 h. Samples were prepared for UPLC/MS-QTOF using the methods described in “General procedure for high resolution HPLC/MS analysis of polar deoxynucleotides.” (d) Extracted ion chromatograms (EIC) for dS^cTP tailing with and without the addition of dBMP showing formation of: (left) starting material (N) and (right) tailed product (N+1).

Line 115; '6-,8-,10-, > '6-, 8-, 10-,'

- These spacings have been added.

Line 118; Reference to Table S1 does not make sense, please refer to Table S2 instead. - This reference has been corrected.

Line 125; 'Supplemental Materials' > 'Supplementary Information'

- This wording has been corrected.

Line 126; No space in 'exo —' and the dash should be a minus ('—' > '-')

- The space has been removed and the dash has been changed to a minus.

Line 128; 'N-nucleoside (Sⁿ) and C-nucleoside' > 'N-nucleoside (Sⁿ) and C-nucleoside' (italics)

- Both letters have been italicized.

Line 129; Please also refer to Table S3 (Yield estimates for XNA tailing and XNA ligation)

- Thank you for pointing this out; Supplementary Table 3 has been added here.

Line 141; There is a space between 'dPTP' and the comma.

- This typo has been corrected.

The reference to Figure 2d-e does not seem appropriate since the information of interest is in parts b, c, f, and g.

- We have changed this reference to 2f instead of 2d-f, as this figure shows the ideal tailing conditions for our model hairpin. Thank you for this comment. Hopefully this clarifies our intent with the reference.

Lines 141-142; General comment that also applies to the Supplementary Information: The use of small or capital letters for writing Figure/figure and Table/table is very inconsistent. Please fix throughout the manuscript.

- Thank you for this comment. We have fixed every instance of “fig.” to “Fig.” and capitalized “table” to “Table.” Additionally, following the format of Nature Communications, all supplementary figures and tables have been listed as “Supplementary Figure/Table X.”

Line 148; Shouldn't the reference to Figure 2f be to Figure 2e?

- There was a mistake in figure letter used. This has been corrected.

Line 149-150; 'restriction exonucleases' > 'exonucleases'

- The word “restriction” has been removed in instances where it comes before the word “exonucleases.”

Line 164; 'Fig.' and 'fig.' in the same line.

- The “fig.” has been capitalized to “Fig.”

Reference to Figure 2g: also reference 2f.

- Figure lettering was swapped. This mistake has been corrected.

Space missing before 'table'.

- This typo has been corrected.

Line 167; 'can be combined' > 'could be combined' since the authors did not actually demonstrate this approach, it is only suggested as a means of incorporating more than one consecutive xenonucleobases.

- We thank the reviewer for this correction; the wording has been changed to “could be combined.”

Line 168; 'blunt end starting' > 'blunt-ended starting'

- This wording has been corrected.

Line 180; no space in '1 00k' à this was not in my copy?

- We looked for this typo and could not find it. It is likely only in your copy. In any case, we can confirm that 100k does not have a space on our version.

Line 191; 'developed previously' reads like the authors are referring to a previous publication, not work in the current manuscript.

- Thank you for pointing this out. We have included the following change in the manuscript:
- The phrase: “The synthetic capabilities developed previously made it possible”
- Has been changed to: “The synthetic capabilities of XNA tailing and XNA ligation made it possible”

Line 196; The term 'NNN-barcode' is rather confusing, because it sounds like there is a random 24-base sequence library linked to the NNN end and you have to look at the supporting tables to clarify this.

- This is a great point and we apologize for the confusion. The NNN bases are linked by a respective barcode which we called the NNN-barcode. As an alternative we will suggest “Triplet-barcode”.
- This phrase NNN-barcode has been removed, and additional efforts were made throughout to explain that the link between the 24-nt triplet barcodes and how they are used to decode the identity of NNN sequences adjacent to XNAs.

Line 201; 'raw reads' > 'raw reads per library'

- This phrasing has been updated.

Line 203; Reference to Figure S15 may be more appropriate than to S17.

- Thank you for highlighting this reference. While we'd like to keep S17 to show more details regarding the segmentation process, it is true that S17 also discusses the filtering of blunt end products; we have added S15 in addition to S10 and S17 referenced here.

Line 229; '(1 – false discovery rate)' > '(1 - false discovery rate)'

- This typo has been corrected.

Line 233; As mentioned in a previous comment, 'randomized 20-nt DNA' sounds misleading, as if the twenty bases were 'randomized' to give a $>10^{12}$ -membered library.

- We apologize for the confusion. We've edited this paragraph in more depth in the next comment, and this phrase is no longer used.

Line 234; Without reference to Table S2f, it is hard to understand where the 100 combinations come from, especially in the context of the confusion surrounding the word 'randomized' in line 233.

- We thank the reviewer for bringing this to our attention. In writing the SI section for this work, we wanted to balance conciseness with clarity. We believe that this has made the reconstruction of our libraries harder to follow. As a major point, we will be including an additional Supplementary data file (**Supplementary Data 2**) that contains full sequences of all our libraries in an xlsx format.
- We have also rewritten this paragraph as follows:
- **Original text:** “We carried out model benchmarking by estimating recall (true positive rate) and specificity (1 - false discovery rate). For testing recall of XNAs, we used XNA tailing and XNA ligation to enzymatically synthesize a new validation library composed of contextually-diverse sequences, far removed in sequence space from those used to build the 4-nt kmer models. The validation library consisted of five sets of 6-letter DNA sequences containing one of the XNA base pairs (B:Sⁿ, B:S^c, P: Z, X:Kⁿ, J:V) embedded within a randomized 20-nt DNA context (**Supplementary Table 1c, 1f, 2c, and 2d**). Each 6-letter DNA library contained 100 sequence combinations, which were subsequently pooled together for sequencing (**Fig. 4, Supplementary Fig. 24**).”
- **Has been rewritten:** To test the recall of our XNAs, we used XNA tailing and XNA ligation to enzymatically synthesize a new validation library composed of contextually-diverse sequences. In this library, the nucleotide sequences adjacent to the XNA-containing heptamer were further diversified making them further removed in sequence space from those used to build the 4-nt kmer models. This validation library was built combinatorically using synthetic hairpin pools as starting material. Each set of hairpins contained 10 unique sequences. To avoid biasing which sequence contexts are chosen for validation, the 20 bp at the 3'-end of each hairpin was designed by randomly selecting standard bases from a uniform probability distribution. Individual hairpin sets were tailed with XNA bases using XNA tailing. Two sets of hairpins with complementary tails could then be ligated, producing a library of 100 possible sequences (10 x 10), with each sequence containing a single XNA base pair. These ligated hairpin libraries were pooled together and sequenced for benchmarking (**Fig. 4b, c, Supplementary Fig. 24**).

-

Line 253; 'Golden Gate' is capitalised

- Thank you. "Golden Gate" has been capitalized.

Lines 264-266; Slightly repetitive after lines 260-262.

- We thank the reviewer for this observation; changes in the text have been made to emphasize the accessibility of our methods in a select number of phrases.
- Additionally, per this suggestion and those from other reviewers, the discussion portion has been rewritten

Line 274; comma after 'engineering'

- This typo has been corrected.

Line 276; It is not clear how the number of codon-anticodon pairs (488) was calculated, please check this. With a set of 12 bases I would naively assume there are 1728 possible codons (from AAA to ZZZ).

We apologize for the confusion; this number of codon-anticodon pairs was calculated based off the assumption that we are still working with single XNA base insertions.

For a single XNA base insertion in a codon/anticodon set:

- We have 8 possible XNAs (B, S, P, Z, X¹, Kⁿ, J, V), treating $S^n = S^c = S$
 - We have 3 possible positions an XNA can be in a codon or anticodon (XNN, NXN, NNX)
- Each N has 4 combinations (A, T, G, C)
- Each X has 2 combinations (XNA and its reverse complement)
- The total combinations with a single XNA is therefore: $8 \times 3 \times 4^2 = 384$
 - The total combinations if we include the standard codon table is therefore: $384 + 64 = 448$

To improve clarity, we have rephrased this sentence in the discussion to the following:

EDITED TEXT:

[...] In the area of genetic code expansion, a single insertion of these additional base pairs allows for various arrangements of up to 448 possible codon-anticodon pairs (made up of 64 canonical codons and 96 additional codon-anticodon pairs for each XNA base pair, constrained to one XNA per codon or anticodon).[...]

Line 309; 'work is' > 'work are'

- This wording has been corrected.

References

5; Author Htar's name is not correct.

- T. Htar T has been corrected to Htar, T. T.

6;

Nature Communications is not abbreviated.

Page numbers are incorrect, use the article number (2383).

- This formatting has been corrected.

9; *There is a space before the hyphen in 'pre-and'*

- This typo has been corrected.

12; *Journal name not abbreviated,*

page range incomplete

- This formatting has been corrected and the page range has been updated.

16; *There is a comma after 'P.' and this is not consistent with the rest of the list*

We see this. This might be an editorial question specifically to situations where author list is short and it might be convention to include “&” prior to last author name. We currently do not use oxford commas for author lists with “&”. A solution could be just remove all the “&” and add the comma, or include oxford comma.

27; *Page range is incorrect, please use the article number 2383*

- *The page range has been updated.*

28; *'Escherichia coli' > 'Escherichia coli'*

- This formatting has been corrected.

29; *'Escherichia coli' > 'Escherichia coli'*

- This formatting has been corrected.

30; *'Taq' > 'Taq'*

- This formatting has been corrected.

31; *Journal name not abbreviated*

- This formatting has been corrected.

34; *Update reference with page numbers, check journal name/abbreviation*

- Page numbers have been updated and journal name/abbreviation has been confirmed.

38; *Page range incorrect, please use article number 129*

- The page range has been updated.

43; *Author name formatting*

- This formatting has been corrected.

44; *Author name formatting*

- This formatting has been corrected.

45; *Author name formatting*

- This formatting has been corrected.

46; *Author name formatting*

- This formatting has been corrected.

47; *Journal name not abbreviated*

- This formatting has been corrected.

49; *Author name formatting*

- This formatting has been corrected.

Figures

Figure 1; Last sentence of the legend is unnecessary

- This sentence has been removed.

Figure 2;

- *b) exo- not with a dash*
 - o This format has been corrected.
- *c) Refer to Table S3 for yields*
 - o The phrase “(yields are listed in **Supplementary Table 3**).” Has been added to the caption.
- *e-f) Legends for e and f seem to be swapped*
 - o Thank you for bringing this to our attention; the captions have been swapped.
- *h) ‘could be cycled’ because the approach is proposed, not demonstrated*
 - o Thank you for this correction. We have since provided an example of MlyI cycling as outlined in **Supplementary Figure 14**.

Figure 3; *d) The orange and red are too similar in my copy*

- Thank you for bringing this to our attention; the color for the XNA model has been changed from red to blue.

Supplementary Information

Line 29; ‘Sulfolobus’ > ‘Sulfolobus’ (italics)

- This formatting has been corrected.

Scheme 1 legend; ‘Chloride’ is not capitalised, space before M in ‘0.2M’

- These typos have been corrected.

Line 62; ‘Chloride’ is not capitalised

- This typo has been corrected.

Line 135; ‘Chloride’ is not capitalised à not found

- This typo was not found or was corrected.

Line 139; Write out ‘concentrated’ in ‘c-H2SO4’

- This formatting has been corrected.

Line 160; ‘as added’ > ‘was added’

- This typo has been corrected.

Line 172; Space in '7 N'

- This typo has been corrected.

Line 211; 'Chloro' not capitalised

- This typo has been corrected.

Line 212; Space before M in '0.2M'

- This typo has been corrected.

Line 215; 'stand' > 'left to stand'

- This phrase has been corrected.

Line 225; General comment: Inconsistent use of a full stop at the end of the text blocks describing the NMR

- Thank you for pointing this out; the text now includes a full stop at the end of every text block describing the NMR.

Line 232; 'Chloroformate' is not capitalised, space in '4h'

- These typos have been corrected.

Line 233; Space in '7N' and 'Chloride' not capitalised

- These typos have been corrected.

Line 236; Space before M in '0.2M'

- This typo has been corrected.

Line 255; 'mmole' should be 'mmol' throughout the manuscript

- This spelling has been corrected throughout the manuscript.

Line 265; Not 'mmole'

- This spelling has been corrected to mmol.

Line 267; Space in '2.45g'

- This typo has been corrected.

Line 276; Space in '1h'

- This typo has been corrected.

Line 278; Space in '4h'

- This typo has been corrected.

Line 279; Not 'mmole'

- This spelling has been corrected.

Line 288; Not 'mmole' and 'Iodide' not capitalised

- This spelling and typo have been corrected.

Line 292; Not 'mmole'

- This spelling has been corrected.

Line 390; 'HF.pyridine' > "HF-pyridine"

- This typo has been corrected.

Line 402; 'HF.pyridine' > "HF-pyridine' and space in '1h'

- These typos have been corrected.

Line 406; Space in '1h'

- This typo has been corrected.

Line 409; The '(0.59 g, 59% for 3 steps)' is not part of a sentence

- Thank you for pointing this out; this phrase is now part of the last sentence of this text block.

Line 432; The '(Refer to the comments for detail.)' is not only not part of a sentence, it also does not refer to anything. There is no other instance of the word 'comments' in the document

- Thank you for bringing this to our attention. We have removed this phrase from the text.

Line 435; Remove 'little'

- This word has been removed.

Line 463; No space in '1h'

- This typo has been corrected.

Line 464; Write out 'sat.-NaHCO₃'

- This phrase now has been written out as “saturated NaHCO₃ solution.”

Line 485; Ammonium bicarbonate concentration is not given

- We apologize for this oversight; the ammonium bicarbonate concentration of 1M has been added to the text.

Line 486; '(e = 11800 in H₂O, l = 391 nm, 88.5 umoles, 74%)' is not part of a sentence

- Thank you for pointing this out; this phrase is now part of the last sentence of this text block.

There are many more of these typographical errors, I will not point out all of these.

Please carefully edit the Supplementary Information

- We are extremely appreciated the time and effort that the reviewer put to look for typographical errors. They have gone above and beyond to point out mistakes that were overlooked. We have performed additional rounds of revision to fix additional errors present.

Line 569; '1 spectra' > '1 spectrum' (also elsewhere, e.g., Line 582)

- Both instances of “1 spectra” have been fixed to “1 spectrum” (these were the only two).

Line 619; In this paragraph, please refer to Figure S26a

- We thank the reviewer for being attentive and catching the need for this reference here; Figure S26a has been referenced in this paragraph.

Line 631; The exonuclease strategy (why are I, III, and VIII used?) is not clearly explained

We apologize for the lack of explanation for this strategy. The general goal of the exonuclease treatment steps is to remove any unreacted starting material or side product. These could be remove using exonucleases since the desired product has no free 3' or 5' end (looped hairpins on each end). The combination used were:

1. Exo I – ssDNA 3' - * 5' exonuclease
2. Exo III – dsDNA 3' - * 5' exonuclease
3. Exo VIII (truncated) – dsDNA 5' - * 3' exonuclease

We empirically found that the combination of exonucleases (rather than only using Exo III for example), more quickly removed undesired product or starting material. That said, this combination is not strictly necessary and reactions will work with single exonucleases (e.g., Exo III only), two exonucleases (e.g., Exo I + Exo III), or other combinations of exonucleases (e.g., Exo III + Exo VII + Exo VIII). The idea of using exonuclease combinations for reaction cleanup came from various examples in literature. Some examples include:

- ExoIII and ExoI has been used to eliminate side products in other reactions (Wang et al, *Anal. Chem.* **2020**, doi.org/10.1021/acs.analchem.0c03303).
- ExoV and ExoVIII have been used to isolate circular DNA (Yang et al, *PLoS Genetics* **2022**, <https://doi.org/10.1371/journal.pgen.1010024>).

This point of clarification has been included in the SI with these details added:

“Following ligation, unreacted hairpins or incomplete ligation products were removed by adding 7.7 U/μL of Exo III (3’-5’ dsDNA exonuclease), 1.5 U/μL of thermolabile Exo I (3’-5’ ssDNA exonuclease), and 0.77 U/μL units of Exo VIII (truncated, 5’-3’ dsDNA exonuclease) and incubating at 37 °C for 1 h, followed by a heat inactivation step at 72 °C for 20 min. This combination of exonucleases was used for rapid undesired product removal, but other exonuclease combinations could also accomplish the same goal.”

Line 651; ‘golden gate’ > ‘Golden Gate’

- All instances of “Golden Gate” have been capitalized.

Line 712; ‘a blunt-end ligation products’ > ‘blunt-end ligation products’

- This typo has been fixed.

Line 733; ‘levels levels’ > ‘levels’

- This duplicate has been removed.

Line 763;

- ‘can either performed’ > ‘can either be performed’
 - o The word “be” has been added.
- *The sentence after this one is not clear.*
 - o Thank you for pointing this out.
 - o The sentence: “Though results shown with main body of the text from per-read alternative hypothesis testing, both options are available for experimentation with the deployed code.”
 - o Has been changed to: “Though **the** results shown with **the** main body of the text **are** from per-read alternative hypothesis testing, both options are available for experimentation with the deployed code.”

Line 775; ‘outlier-robust robust’ > ‘outlier-robust’

- This duplicate has been removed.

Line 776; 'an agnostic maximum likelihood criteria' > 'agnostic maximum likelihood criteria' - The "an" has been removed.

Line 797; 'as followed' > 'as follows'

- This grammatical error has been corrected.

Line 871; 'xenomorph models .' > 'xenomorph models.'

- This space has been removed.

Line 931; 'provided' > 'is provided'

- Thank you for this correction.
- The phrase: "Additional reference for each base and base pair provided (Ref) column."
- Has been changed to: "Additional references for each base and base pair are provided in the (Ref) column."

Table S1a; Entries for bases X^t and J end in full stops, unlike the rest

- Thank you for pointing this out. We have taken out the periods for X^t and J such that none of the entries end in a full stop.

Lines 944-945; 'containing 10 randomized 20mers each (library size = 10 x 10).' is confusing because it sounds like a randomised 20mer (420 variants).

- We apologize for the confusion and thank the reviewer for their vigilance. We have modified the description of how these validation sets are constructed by using the phrasing "randomly chosen 20mer". In subsequent instances, we simplify this explanation to saying 10 unique sequences.

Edited text now reads (collected from various portions of manuscript):

- "Each hairpin pool contains 10 unique sequences. Ligating two hairpin pools together generates a final library of 100 possible sequence combinations (10 x 10)."
- "Two pools of validation hairpins (each containing 10 unique sequences) can be ligated together to generate 100 (10 x 10) random combination of sequences"
- "Ligating two pools together (with complementary N+1 tails) results in a library with 100 possible sequences (10 x 10 combinations)."

Line 957; 'The NNN-barcode is a 24 nt sequence' is confusing, see previous comments

- This phrasing has been amended throughout the manuscript. NNN-barcode is now called the triplet-barcode, which is used to decode the 3'-NNN region. Referencing to an [NNN-BC] only exists as

an abbreviation in tables, and corresponding figure captions clearly explain this as the triplet-barcode.

Line 962; 'sequences shown table SIC.' > 'sequences are shown in table SIC.'

- We have amended this sentence to clarify the intended meaning.

Line 965; 'Sequences contained within each validation library pool.' is not a sentence and does not seem to contribute

- We apologize for the lack of clarity here, and have worked on changing this phrasing to be more clear.
- We've edited the phrase: 'Sequences contained within each validation library pool.'
- To the sentence: 'The randomly chosen 20mer sequences contained within each validation library pool are listed.'

Line 966; '20 randomized bp' is confusing, see previous comments on this

- We apologize for the confusion here and have edited how we describe the 20 randomly selected bases throughout the text.

Line 994; 'each dxNTPs and dNTPs tailed' > 'each dxNTP and dNTP tailed'

- The "s" for both nouns have been removed.

Line 1018; 'reactions conditions' > 'reaction conditions'

- The "s" from "reactions" has been removed.

Line 1022; Table units are not explicitly enough stated as fractions. Percentages would be more intuitive.

- Thank you for bringing this to our attention; the table has been updated to a percentage, and the text has also been changed to specify "percent relative intensity."

Line 1030; Percentages are used in this line but the data in the table are presented as fractions, this can be confusing. Also see previous comment.

- We apologize for this oversight and thank the reviewer for their attentiveness. The values in Supplementary Table 3b have been adjusted to percentages.

Supplementary Figures

Figure S2;

- Spaces between numbers and units.
 - o Spaces have been added between numbers and units.
- ‘*Sulfolobus*’ must be italicised.
 - o *Sulfolobus* has been italicised
- This is a general comment that applies to many figures/gels: The ‘no rxn’ annotation is confusing, it seems to mean something closer to ‘no ligation’. Writing ‘no rxn’ is misleading since the polymerase is being assayed, and ‘no rxn’ (no ligation) means that the polymerase reaction worked, and the ligation therefore failed.
 - o This is a great point, and we have changed all “no rxn” to “no ligation” in the figures.
- In Line 1135, ‘full ligation of blunt-end product’ should be ‘full ligation of bluntend hairpin/substrate’ since it is not a product if it was not extended by tailing.
 - o Thank you for this correction. Blunt-end product has been changed to “blunt end hairpin” across the document.

Figure S3; There is a blue square in (a) that does not seem to belong there.

- o This has been fixed

In Line 1143, ‘Chromatograms scales’ should be edited (Chromatogram scales; Same for Figure S4).

- Thank you for catching this. The blue square has been removed.
- The “s” from “chromatograms” has been removed for both figures S3 and S4.

Figure S5; Part (e) is not clear. Was there a ligation step between the Terminator and exonuclease treatments? As I understood the method, there was, but it should be explicitly stated here to make the details easier to follow.

- We apologize for this confusion; figure (e) is figure (d) digested by exonucleases. To clarify that there was a ligation step, we have added the phrase “followed by T4 DNA ligation” in this caption.

Figure S6;

- Why is the unligated DNA (lower band) not degraded by the exonuclease treatment?
 - o This is an astute observation raised by the reviewer, and it is related to the exonuclease choice and relatively high DNA concentration used in this reaction. In this early experiment during method development, we had a slightly different cocktail of exonucleases (Exo I, ExoIII, and ExoVII). The choice of Exo VII at this stage was motivated by literature published by PacBio on SMRT Bell sample preparation.
 - o Soon after this work we tried swapping from ExoVII to ExoVIII as the third exonuclease. This greatly improved our ability to degrade unreacted products in the desired time window and temperature (1h at 37C).
 - o Since the presence of unreacted product was not instrumental in the interpretation of results, we did not follow through with repeating the experiment using ExoVIII.
- Not obvious from the figure whether YiPP helps improve yields or not, please add this interpretation.

- We thank the reviewer for pointing this out and we will make edits to make our conclusion more clear. When comparing between conditions with YiPP as the only differing factor (for example Lanes 5 vs Lane 6) we observe that reactions containing YiPP lead to more ligation product. Likewise, Lanes 1 + 2 can be compared to Lanes 3+4 and indicate that + YiPP conditions had slightly higher yields.
- We have amended the figure caption title, it now reads: “**Addition of yeast inorganic pyrophosphatase (YiPP) leads to slight improvements in XNA tailing reaction yield**”
- *The unusual heat inactivation protocol in Lines 1193-1194 should be explained.*
 - We apologize for the lack of detail here, which is explained by the difference in exonuclease combination used in this particular experiment (Exo VII instead of Exo VIII). Exo VII required a 10-minute 95 °C inactivation, whereas ExoI could be inactivated at 80 °C for 20 min. Without over-optimization, we chose to perform higher temperature heat inactivation followed by a lower temperature heat inactivation. Future work switched to using Exo VIII which can be inactivated at lower temperatures.
 - Related to point previous point, we have made an explicit note in the figure caption that this particular experiment uses Exo VII instead of Exo VIII.

Figure S7;

- *Please explain this experiment. Is over-tailing supposed to result in decreased ligation? How would this be observed, by decreased intensity of the ligation product band?*
 - We apologize for the lack of clarity in our explanation. This was an experiment we included because we had it. Ultimately, these results are corroborated with other experiments but still serve as an independent control for synthetic N+1 vs tailed N+1 DNA.
 - The reviewer’s observations are correct; that is, over-tailing would result in less ligation. Only one hairpin is being tailed enzymatically in this experiment, while the other hairpin in the complementary pair is prepared synthetically (therefore, is not over tailed).
 - We have amended the figure caption to now include the description of the experimental setup and hypothesis, and now explicitly state the conclusion.
 - **Briefly:** Any ligations between N+M (tailed) and an N+1 (synthetic) hairpin would result in a dsDNA product with a gap. This gap makes the DNA susceptible to exonuclease degradation, whereas the N+1 (only) tailed product would result in dsDNA without any free ends. This explanation has now been included in the figure caption.
- *I do not see any controls for this. Since different oligo concentrations were used for T4 and T7 ligations, the authors should comment on whether the amounts of DNA loaded onto the gel were normalised or not, otherwise this figure does not explain anything.*
 - We apologize for the confusion, and thank the reviewer for bringing this to our attention. This experiment was not meant to measure differences between T4 and T7 ligase, but rather the differences in product observed when enzymatically tailed (N+M hairpins; leaving possible M=1,2,3,4 etc) and chemically synthesized (“pure N+1 DNA” hairpins) are used.
 - T4 and T7 DNA ligase can both ligate these substrates, so we tested both.
 - Synthetic DNA (Synth) functions as a positive control since it only contains N+1 tailed DNA.
 - Enzymatically tailed DNA (“N+M” tailed DNA) is the experimental sample.
- *I do not understand how this experiment could be used to investigate ‘measurable differences resulting from over-tailing’.*

- We have made a few changes to the figure caption and text to more clearly explain this experiment, results, and conclusion.
- The caption of the figure has been changed to: “Enzymatic tailing does not lead to measurable differences in ligation when compared to ligation using fully synthetic hairpin with N+1 tails.”
- Please note that ultimately we included this data since we had it and liked the simplicity of the experiment. This type of experiment can be run without high-end analytical equipment. The results regarding over-tailing are independently verified for all XNAs bases covered in this work with oligo LCMS data. If the reviewer suggests this piece of data is not necessary or a poor design, we can remove it from the manuscript without changing conclusions.

Figure S8; In some chromatograms (d, g, m, and n) a minor peak at shorter retention time is visible for the N+1 plots. Do the authors know what this is?

- We thank the reviewer for their observation. While we’re not exactly sure of the peak identity, this peak only appears because of the increased mass tolerance we utilized to display our data. We provide a more detailed description of how we know this is the case below:
- Briefly: Since we are using ESI for ionization, our oligos fly at a wide distribution of mass/charge ratios and isotope abundance. The chromatograms only show the most abundant isotope (as calculated from natural isotope abundance, with m/z used tabulated below) and the most abundant charge state (as empirically determined).
- For proper peak assignment, we fall back to the mass spectra extracted at any given time (which involves matching the presence of multiple peaks (primarily C13 isotope) and relative abundance of these peaks (isotope abundances).
- Minor peaks come from noise due to a large error tolerance we used for detection (+/- 50 ppm). These minor peaks do not have mass spectra that match the isotope abundance expected. As shown below, decreasing the mass tolerance (from 50 ppm to 20 ppm) makes the smaller peak disappear.
- For the choice of ppm window used in this case (even though we have a high resolution instrument): we lose resolution at higher mass ranges and at higher charge states. Higher mass compounds will tend to broaden the distribution of isotopes (more atoms, more possible isotopes), while higher charge states will move spectra closer together. This type of detection pushes the limit of our instrument’s resolution (Agilent 6530c QTOF) but is still possible. If we used lower mass tolerance windows we would observe chromatography artifacts (jagged peaks) since some ions would be binned outside that window.

RESPONSE FIGURE (Not included in manuscript)

Figure: Figure S8 with 20ppm and 50ppm mass tolerance for dVTP tailed product (N+1).

Figure S9; The text ‘T7 ligase preferentially ligates hairpins with a single nucleotide overhang’ is misleading since it selectively ligates cohesive overhangs (not limited to single nucleotide overhangs) rather than blunt ends.

- The text “single nucleotide overhang” has been adjusted to “cohesive nucleotide overhang.”

Figure S11;

Please comment on the large amount of blunt ligation product observed for T7 DNA ligase (Figure S11b, -/- lane) when this ligase is expected to selectively ligate cohesive overhangs.

- We appreciate the reviewer’s astute question. T7 DNA ligase does favor cohesive overhangs but under certain reaction conditions, blunt-end ligation product can also be formed.
- Specifically, the use of crowding agents (high MW PEG) in the buffer leads to more favorable conditions for blunt-end ligation. Our buffer contains 7.5% (w/v) PEG 6000. The extended reaction time further makes it more likely to form blunt product (if blunt ends are available to ligate).
- This phenomenon is known from literature. Use of crowding agents such as 20-30% PEG 6000, has been shown to make T7 DNA ligase activity comparable to T4 DNA ligase for blunt-end ligation (Doherty et al, *J. Biol. Chem.* **1996**, <https://doi.org/10.1074/jbc.271.19.11083>).

Also, please clarify the expected observations for G+/- ligations.

- The G+/- conditions (now amended to G+/[•]) is a self-ligation negative control. Under certain conditions mismatch ligation products can arise. For similar reasons why we encounter blunt-end ligation using T7 ligase, running ligation reactions at high hairpin concentrations, long incubation times, high ligase concentrations, or use of crowding agents.

- For example, it has previously been shown that T3 DNA ligase is more tolerant of standard purine:purine mismatches compared to T4 and T7 DNA ligases (Bilotti et al, *Nucleic Acids Res.* **2022**, <https://doi.org/10.1093/nar/gkac241>).
- We consider mismatch ligation possibilities in this work and run these negative controls. Importantly, mismatch ligation sets a constraint for ligation yield. Increasing time/hairpin concentration/ligase can lead to higher yield at the cost of lower purity. Reaction optimization was done to maximize purity, and we looked for XNA tailing and XNA ligations conditions where blunt end or mismatch ligation product is minimized.

Half the DNA in these reactions (-) is capable of self (blunt) ligation, so ligation is expected as for the -/controls, just less. This comment assumes that "-" represents a blunt hairpin oligo, explaining why -/- gives so much ligation product. If this interpretation is incorrect, please clarify the legend.

- o This is a poor choice on our end for legend and we apologize for the confusion here due to our lack of clear labeling. Due to an inconsistency in how we previously lanes, the (-) represented two different things:
 1. For (-/-): - meant blunt end DNA. This should be a major product.
 2. For (N/-): - meant only one hairpin was included in the reaction (N+1 tailed). Any product in N/- conditions is from blunt-end ligation or mismatch ligation (both minor)
- o We have fixed this in the figure caption and figure legend by updating all the gels for consistency.
 - has replaced (-) to indicate absence of the hairpin tailed with the complementary base.

Figure S12;

For part (a), there seem to be two bands corresponding to higher-molecular weight ligation products for the J/V ligation, please comment.

- Thank you for this astute observation. For background: this experiment was designed to distinguish between blunt-end ligation from mismatch self-ligation (as observed in **Supplementary Figure 11**). This was accomplished by utilizing hairpins that form the NdeI restriction site when blunt-end ligated, such that a subsequent combination of exonuclease and NdeI treatment would lead to complete degradation if the product were blunt-end ligated, and no degradation if there was a mismatch ligation.
- Regarding the reviewer's comment on the higher MW species: we do not have a definitive answer as to what that species was. Since the presence of the two bands was not important for our conclusion for this particular experiment, we never followed up on it. We had performed this experiment early on in development when screening conditions and had initially attributed it to a gel artifact.
- The ladder shown in the gel is a 50 bp NEB (the bottom three bands are 50bp, 100bp, 150bp respectively). We could not come up with a combination of products that would give us two bands that resolve as they did, given that we were using two hairpins with similar MW. These reactions did have NdeI one in them and were run on a gel without purification after NdeI + Exo treatment.
- Since we could not draw up a simple explanation for two bands, we decided to repeat ligation of the NdeI-containing hairpins for the JV condition.
- The samples shown in the gel were all prepared in parallel using the optimized conditions.
- **Result:** We do not see two bands even though there is clearly higher resolution in this gel (as measured by inter-ladder spacing) than in the original gel. The V-self ligation lane (Lane 2) for this

experiment does look like it elutes slightly higher, but harder to tell. The same ladder is used here as was used in the original figure.

- **Note 1:** Here we load the same amount of initial DNA in each lane, rather than the same amount of reaction volume such that comparisons can be performed quantitatively across the board. For example, Lane 5 and Lane 6 started with the same total amount of hairpin, but Lane 6 clearly has more product.
- **Note 2:** This means we are loading less single hairpin conditions (Lanes 1, 2, 4, 5, 9) than we typically would. We show a longer exposure image here to more clearly see any faint bands from mismatch ligation that might be present.

RESPONSE FIGURE – Repeat of experiment (Not included in manuscript)

T3 ligase for JV XNA ligation with and without NdeI-HP using optimized conditions. We repeated ligation of J/V hairpins that contain NdeI cut site as shown in **Supplementary Figure 12a**, and included another control hairpin that does not form an NdeI restriction site. Lanes labeled with single letter abbreviation of nucleotide tailed onto 3'-end of hairpin. In all gels, • indicates absence of the hairpin tailed with the complementary base. All hairpins with an (*) indicate control hairpins that do not form an NdeI restriction site upon blunt-end ligation. Pre-tailed negative control (G⁺) shows lack of ligation and subsequent digestion for a hairpin with a 3'. Blunt end negative control of a reaction ligation, (-/-) condition, containing 5'-Phos-NdeI-HP-1 and 5'-Phos-NdeI-HP-2 shows digestion by NdeI, and subsequent digestion by exonucleases. Blunt end positive control of a reaction ligation, (* / *) condition, containing 5' Phos-HP11 which does not form an NdeI restriction site when it undergoes blunt-end ligation; the ligated product will not be cleaved by NdeI, and is therefore resistant to exonucleases.

- *Why is self-ligation observed for the G+/- condition (expected because of self-ligation of the blunt “- component) but not for (J/-, X/-, K/-, and most importantly, -/- reactions?*
- We thank the reviewer for their thoughtful question. This result is expected but for a different reason. We had a poor choice of lane labels here that we have clarified. Specifically, some reactions only contained one hairpin (e.g., J/-) which is different from reactions that contained blunt-end hairpins (-/-).
 - Generally, products in J/-, X/-, K/-, and G+/- conditions should not be expected since these should not have blunt end ligation. However, mismatch ligation is known to be possible (and an undesired product).
 - It has previously been observed that DNA ligases are capable of mismatch ligation under certain conditions and using certain ligases (Bilotti et al, *Nucleic Acids Res.* **2022**, <https://doi.org/10.1093/nar/gkac241>).
 - We optimize reactions to minimize mismatch ligation while also making sure to observe correct ligation product.
 - The (-/-) condition is blunt end ligation condition. Since we are treating with NdeI, we should not observe product as blunt-end ligation results in NdeI cut site formation.
 - In order to clarify this point, this sentence has been added to the figure caption “*This ensures that after XNA tailing, XNA ligation and NdeI/exonuclease treatment, the only products left are ligation products from properly tailed material, which prohibits the formation of the NdeI restriction site.*”
 - Additionally, we have included discussion in the main text that mentions how these controls were used to optimize reactions around minimizing both blunt-end ligation and mismatch ligation.
 - These experiments were used in reaction development to help us think about and optimize reactions. As mentioned previously with respect to Figure S11, we always run single hairpin ligation controls. And finally, as mentioned in an earlier comment, we have since updated lanes that contain one hairpin only as (N/●) instead of (N/-) to clarify that these lanes only contain a singular tailed hairpin and no blunt-end hairpins.

Figure S14; Line 1324; ‘various XNAs base pairs’ > ‘various XNA base pairs’

- The “s” from “XNAs” has been removed.

Figure S16;

- *‘outlined in previously’ > ‘outlined previously’.*
 - *The “in” has been removed.*

For part (c), please comment on the observation that J and V seem to self-ligate more than the other bases.

Absolutely and thank you for this suggestion. In the text we have made the following caption edit:

EDITED TEXT: Supplementary information

Minimal self-ligation is observed for XNA-tailed NNN-pools, with J and V-tailed NNN-pools showing the most self-ligation compared to all the other XNA-tailed NNN-pools. These data suggest that the major product of XNA ligation with complementary sets is desired heteroligation products. After sequencing, self-ligation products are identified by their pool barcodes and are removed prior to model building.

As a response to the reviewer: The observation that J and V seem to self-ligate more than other bases is very likely either coincidental or a result of both using a T3 ligase and amenable to self-ligation. Two possible sources of these products:

1. Mismatch ligation: JV ligation uses T3 ligase which is more likely to allow for mismatch ligation. Because our XNA tailing and XNA ligation is optimized on for one sequence context, we do not know how much J and V self-ligation is favored under all possible sequence contexts.
2. Blunt end ligation as resulting from incomplete tailing: It is also possible that the library's expanded sequence contexts led to incomplete tailing for specific sequences, and therefore blunt end ligation.

In the library tailing and ligation it is not possible to rule out either of these possibilities. Since the destination of these products was for sequencing, both of these self-ligation products could be easily identified from pool barcodes and removed *in silico*.

Figure S18; 'sequences that contains' > 'sequences that contain'

- The "s" from "contains" has been removed.

Figure S19; 'Boxplot show showing' > 'Boxplot showing'

- "Show" has been removed from this sentence."

Figure S25;

- *This is a table(s), not a figure.*

- o This figure has been changed into a table.

- *In the statement "'N vs §' comparisons, '§'" the § symbol has more than one appearance.*

- o **We apologize for the oversight; this is due to differences in font. The § symbol here has been turned to §.**

- *Please rephrase 'were used classification.'*

- o This phrase has been changed to "were used for the base classification."

- *There is a '0.92' in Table (a) that is not correctly formatted.*

- o This formatting has been corrected. Apologies.

- *There is one value in Table (c) that is much higher than the neighbouring numbers (0.34, intersection of row A with column Xt).*

- o This formatting error has been fixed. We thank the reviewer for picking it up.

REVIEWERS' COMMENTS

Reviewer #1 (Remarks to the Author):

The authors have responded to my previous review comments clearly. At this stage, this work includes a substantial amount of work, which is respected and appreciated. The core innovation of this work is actually more about the enzymatic synthesis of sequencing libraries including xenonucleobases. However, this enzymatic synthesis method doesn't seem to be efficient at all.

The technical advancement in the development of nanopore sequencing technique is rather limited. I admit that a custom algorithm was included. Though it is useful for extremely specialized cases of sequence combinations to show the technical feasibility, it is far from completeness.

Innovation wise, the 2023 JACS paper already showed nanopore sequencing of DNA with single base substitution of xenonucleobases, though only 4 exonucleobases instead of 8 were used in that paper. However, the test of feasibility is already demonstrated. The use of a different platform, a different pore type or a different protocol in this manuscript are not counted as the innovation, as long as the same technical principle is used. They are just technical differences. The authors also failed to show a complete Kmer database ($12^4 = 20736$), which is necessary for de novo sequencing of any base combinations of all 12 possible xenonucleobases. I understand that it is a lot of workload but at this stage it means that the nanopore sequencing model is not yet complete and we don't need two high impact papers to demonstrate the feasibility of nanopore sequencing of DNA containing Supernumerary DNAs again and again.

The journal of Nature Communications has a transparent peer review policy, meaning that all review comments will be published along with an accepted paper. At this stage, I would not recommend or deny the publication of this manuscript but to show my honest opinion on this work. I also hate to permit publishing of a work which contains clear overstatements of technical advancements. Since a paper of this kind would discourage publishing of future works which truly solved the problem.

Thus, I have two further suggestions for the authors:

1. Change the manuscript title to: Enzymatic synthesis of 12-letter Supernumerary DNA and nanopore sequencing of DNA with single xenonucleotide substitutions. Feel free to change the writing of the title as you wish but make sure to clearly state that your model can only handle nanopore sequencing of DNA with single xenonucleotide substitutions in the title.

2. In the abstract and the conclusion, clearly state that the current sequencing model fails to cover a complete Kmer database of $12^4=20736$.

Eventually, this manuscript demonstrates a lot of workload and it is respected. However, I still consider it an incomplete work. The innovation is also damaged by a prior work with a few coauthors in this manuscript as well. If this paper is still published, please make sure to follow my above suggestions to leave the publishing opportunity for you or others in the future when nanopore sequencing of 12-letter Supernumerary DNA is truly achieved.

Reviewer #2 (Remarks to the Author):

The authors have addressed all comments from the previous round of review

Reviewer #3 (Remarks to the Author):

I thank the authors for their very detailed response to my comments. I was particularly satisfied that they performed additional experiments that I only alluded to, no requests for additional experimental work were made. In particular, I think the experiment reported in the new SI Figure S25 (PCR amplification and sequencing of a DNA template with a P≡Z base pair) is very valuable, demonstrating how the reported XNA synthesis and sequencing technology could be used to study e.g., XNA replication conditions. Great care seems to have been taken in addressing my comments, and this applies to the other two reviewer's comments as well. On the one hand one can understand the more critical nature of some of the other reviewer's comments, since for now each X needs to be in the context of natural bases to be sequenced using the kmer models reported in the paper. As reviewer 1 points out, this is only a small subset of the total combinatorial space where all bases are taken from the 12 letter alphabet (i.e., NXN vs XXX). However, looking at the impressive amount of work it took to train a base caller for identifying X bases in the context of natural bases, it is clear that the extra effort required for sequencing X in the context of other unnatural bases is not something one can currently rationally expect of the authors. Indeed, more breakthroughs will be necessary. Importantly, the authors have provided us with tools that place XNA synthesis and sequencing in the hands of the general scientific public, which is exactly what is needed for these new breakthroughs to be achieved.

Please find attached our revision and comments on the version of the manuscript titled originally titled “Synthesis and Sequencing of 12-letter Supernumerary DNA” (Manuscript ID: NCOMMS-23-17508B). At the outset, Review #1 suggests that we expand the title, which we have. It is now: “Enzymatic Synthesis and Nanopore Sequencing of 12-letter Supernumerary DNA”. We greatly appreciate the feedback of editors and reviewers. With the changes made, we are now confident the manuscript has been appropriately prepared for publication. Please see below for reviewer point-by-point responses.

The following text formatting was used in this response to help improve readability:

- *Original reviewer comment = blue italics*
- Authors response = black
- Edited manuscript text excerpts = green

Reviewer #1 (Remarks to the Author):

The authors have responded to my previous review comments clearly. At this stage, this work includes a substantial amount of work, which is respected and appreciated. The core innovation of this work is actually more about the enzymatic synthesis of sequencing libraries including xenonucleobases. However, this enzymatic synthesis method doesn't seem to be efficient at all.

We appreciate the respect and appreciation.

For BSPZ, the alternative is phosphoramidite synthesis that costs \$100-400 per nucleotide, or \$200-800 per base pair, with a lead time of 2-4 weeks, without the possibility of making libraries. Our enzymatic synthesis is cheap (~\$2 per base pair), generates products in one day, and is compatible with library preparation. For other bases (JVXK), commercial chemical synthesis alternatives do not exist. In our view, which NC's transparency makes available to its readership, one does not criticize a method that does things that can be done with 0.5% of the cost, or a method that does things that otherwise not done at all, because it does not "seem" to be efficient.

The technical advancement in the development of nanopore sequencing technique is rather limited. I admit that a custom algorithm was included. Though it is useful for extremely specialized cases of sequence combinations to show the technical feasibility, it is far from completeness.

We have addressed how this work opens access to new areas of investigation for the field for both low cost and accessible synthesis, and next generation sequencing. The “extremely specialized cases” you allude to are the main focus of the scientific field. Examples we have previously highlighted – aptazymes and xnazymes are sparse in XNA density; expanded genetic codes can use a single XNA pair in the anticodon

of tRNA and codon of mRNA to create hundreds of new codons with what we described (addressing an existing problem in codon-anticodon orthogonality).

Innovation wise, the 2023 JACS paper already showed nanopore sequencing of DNA with single base substitution of xenonucleobases, though only 4 exonucleobases instead of 8 were used in that paper. However, the test of feasibility is already demonstrated. The use of a different platform, a different pore type or a different protocol in this manuscript are not counted as the innovation, as long as the same technical principle is used. They are just technical differences. The authors also failed to show a complete Kmer database (12 to the 4=20736), which is necessary for de novo sequencing of any base combinations of all 12 possible xenonucleobases. I understand that it is a lot of workload but at this stage it means that the nanopore sequencing model is not yet complete and we don't need two high impact papers to demonstrate the feasibility of nanopore sequencing of DNA containing Supernumerary DNAs again and again.

These comments were already addressed extensively in the first round of reviews but we can summarize key points. The 2023 paper shows nanopore sequencing with only 4 of the added nucleotides, because it could not do the others. Certainly, it did not demonstrate the "feasibility of the others". This work has more building blocks (and different ones even for BSPZ set).

Regarding the "use of a different platform" - These are the differences between millions of reads per day versus tens of reads per day; between a capital investment of \$1000 versus \$100,000; and between sequencing an instrument that is commercially available versus one that is not. Sanger DNA sequencing and Illumina DNA sequencing also use "the same technical principle". Does the Reviewer conclude that the second offers no innovation over the first? We argue our improvements are one that would interest and captivate a NC reader.

Reviewer might have one concept of "completeness"; but we have another. For us, we have **completed** the 12 (4 + 8) alphabet. The 2023 paper does not. We have completed it on a publicly accessible platform. The 2023 platform does not. The 2023 paper does not demonstrate feasibility for more than half of the expansion. Ours demonstrates feasibility for the **complete** expansion. In any case, before this paper, the state of the art for XNA sequencing was gel shift assays. These are non-multiplexable, non-single molecule, and low throughput. We see progress. Review 1 sees little. Let us let the NC readers decide. This work is not a feasibility test, it is technology development and deployment.

For BSPZ, the alternative is phosphoramidite synthesis that costs \$100-400 per nucleotide, or \$200-800 per base pair, with a lead time of 2-4 weeks, without the possibility of making libraries. Our enzymatic synthesis is cheap (~\$2 per base pair), generates products in one day, and is compatible with library preparation. For other bases (JVXK), commercial chemical synthesis alternatives do not exist. In our view, which NC's transparency makes available to its readership, one does not criticize a method that does things that can be done with 0.5% of the cost, or a method that does things that otherwise not done at all, because it does not "seem" to be efficient.

Regarding – “authors failed to show complete kmer database...” - Since this paper demonstrates, **for the first time**, the feasibility of sequencing a **complete** 12-letter alphabet, we believe that it will **encourage**, not discourage, future work to address the fully factorial sequencing problems. The NC transparency policy will let the readers decide.

We have actually done more than the review is giving us credit for. This work does more than show feasibility of sequencing 8 letters like JACS papers. Countless papers sequence just 4 letters so surely we can be less reductionist here. We have made a good-faith effort to elaborate on innovations presented in this work (contextualized to the state of the field) in the first round of reviews, which seem to not have been acknowledged. As mentioned before, if your problem in this respect is with the authors of the JACS paper, please note as mentioned previously that this work was led (and performed, and funded) by a completely independent research group. Please see author contributions to see how those Co-authors contributed to this independent project.

The journal of Nature Communications has a transparent peer review policy, meaning that all review comments will be published along with an accepted paper. At this stage, I would not recommend or deny the publication of this manuscript but to show my honest opinion on this work. I also hate to permit publishing of a work which contains clear overstatements of technical advancements. Since a paper of this kind would discourage publishing of future works which truly solved the problem.

We value the transparent peer review policy of Nature Communications, appreciate the willingness of Reviewer 1 to not "deny publication of this manuscript", and suggest that we simply proceed to publication to test whether we, or the reviewer, are correctly evaluating value and impact of this paper.

Thus, I have two further suggestions for the authors:

1. Change the manuscript title to: Enzymatic synthesis of 12-letter Supernumerary DNA and nanopore sequencing of DNA with single xenonucleotide substitutions. Feel free to change the writing of the title as you wish but make sure to clearly state that your model can only handle nanopore sequencing of DNA with single xenonucleotide substitutions in the title.

We have made changes to the title to specify enzymatic synthesis and nanopore sequencing. Given that we show both first time enzymatic synthesis and nanopore sequencing of DNA with a full set of 12-letters, we believe the title accurately conveys the message we wanted to deliver and is clearly shown in the work.

2. In the abstract and the conclusion, clearly state that the current sequencing model fails to cover a complete Kmer database of $12^4=20736$.

We have stated this throughout the text as per suggestions from the first round of revisions.

Eventually, this manuscript demonstrates a lot of workload and it is respected. However, I still consider it an incomplete work. The innovation is also damaged by a prior work with a few coauthors in this manuscript as well. If this paper is still published, please make sure to follow my above suggestions to leave the publishing opportunity for you or others in the future when nanopore sequencing of 12-letter Supernumerary DNA is truly achieved.

We have done this throughout the text. Please keep in mind, before we can sequence all 12 letters in supernumerary DNA covering all possible kmers, the synthetic chemistry will need to be improved. If you want progress in this field, we need to inspire it and invite others to take on challenges. We, and others in the community, eagerly anticipate that improvements at commercial scales will become available in the next decade that will open up even more possibilities.

Reviewer #2 (Remarks to the Author):

The authors have addressed all comments from the previous round of review

- We thank Reviewer #2 for taking the time to look through our revisions, and are grateful that our comments were able to properly address their concerns.

Reviewer #3 (Remarks to the Author):

I thank the authors for their very detailed response to my comments. I was particularly satisfied that they performed additional experiments that I only alluded to, no requests for additional experimental work were made. In particular, I think the experiment reported in the new SI Figure S25 (PCR amplification and sequencing of a DNA template with a P≡Z base pair) is very valuable, demonstrating how the reported XNA synthesis and sequencing technology could be used to study e.g., XNA replication conditions. Great care seems to have been taken in addressing my comments, and this applies to the other two reviewer's comments as well. On the one hand one can understand the more critical nature of some of the other reviewer's comments, since for now each X needs to be in the context of natural bases to be sequenced using the kmer models reported in the paper. As reviewer 1 points out, this is only a small subset of the total combinatorial space where all bases are taken from the 12 letter alphabet (i.e., NXN vs XXX). However, looking at the impressive amount of work it took to train a base caller for identifying X bases in the context of natural bases, it is clear that the extra effort required for sequencing X in the context of other unnatural bases is not something one can currently rationally expect of the authors. Indeed, more breakthroughs will be necessary. Importantly, the authors have provided us with tools that place XNA synthesis and sequencing in the hands of the general scientific public, which is exactly what is needed for these new breakthroughs to be achieved.

- We thank Reviewer #3 for their kind words, and also once again for the amount of time and effort placed into reviewing our manuscript. We agree that our work is currently limited by a small subset of the full 12-letter combinatorial space, and simultaneously that public accessibility is required for additional breakthroughs. We are grateful that our revisions were able to fully address Reviewer #3's comments.